# Deep-sea-floor diversity in Asteroidea is shaped by competing processes across different latitudes and oceans

H. F. Carter [1,2] ✉, G. Bribiesca-Contreras [1,3] & S. T. Williams [1]

The occurrence, shape and drivers of global distributional trends in species richness throughout the deep sea are poorly explored. Here we present a spatial description of the global, bathymetric and taxonomic extent of the benthic marine class Asteroidea using a compiled dataset of ~200,000 species-level occurrence records. We used these data to produce comparisons of sea-floor richness between hemispheres and oceans. We show that species richness is significantly correlated with temperature and nutrient flux despite markedly different distributional patterns across oceans and latitudes that suggest further influence from a combination of additional geographic, taxonomic and environmental factors. The relative importance of temperature and nutrient levels also varies greatly with depth. Species richness peaks in the shallow-water tropics, closely matching sea-floor temperature variation, but at bathyal and abyssal depths it is higher at temperate latitudes, where nutrient flux levels are of greater importance. We show that richness in the deep benthos is restricted below ~1.5 °C, with this strong thermal threshold consistent among oceans irrespective of other factors.

The deep benthos is by far the largest single habitable space on Earth, comprising around 90% of the sea floor—itself 70% of the global surface[1]—yet despite an increasing number of important regional and broad-scale studies[2,3], the drivers shaping large-scale patterns of diversity and species richness remain poorly understood[4–6]. Many previous studies have focused on limited geographic regions[7], use very broad-resolution data or risk possible bias towards common or cosmopolitan species through reliance on modelled results rather than collection data. Despite the increasing size of many available datasets, taxonomic or habitat incompleteness have also hindered such analyses[8]. For instance, most putatively global studies have focused only on the Atlantic (although see ref. 9), which should not necessarily be treated as representative of other oceans that have their own geographic and evolutionary histories impacting faunal compositions[10–12].

The class Asteroidea (starfish) as an entirely marine, globally distributed, benthic clade are an excellent model group with which to examine patterns of sea-floor richness over oceanic, hemispheric and worldwide scales. Here we present global patterns of asteroid species richness derived from a large, near taxonomically comprehensive database representing 91.4% of described asteroid species (1,751 species) and 98.3% of genera, spanning 160° of latitude from the Arctic basin to the Antarctic coastline across the full extent of longitude, and a bathymetric range of more than 7,500 m (Figs. 1 and 2b). We used both raw and interpolated species ranges based on the known latitudinal limits for each species within their individual bathymetric ranges to produce values for maximum potential species richness for each degree of latitude. We used these data to produce a uniquely comprehensive overview of starfish biodiversity and to depict the changing shape of benthic diversity in this clade across three major depth strata in the ocean (shallow, 0–200 m; bathyal, 200–2,000 m; lower bathyal and abyssal, 2,000–6,000 m), over differing spatial scales (global, hemispheres and ocean basins) and accounting for taxonomic relationships.

[1]Department of Life Sciences, Natural History Museum, London, UK. [2]Department of Genetics, Evolution and Environment, University College London, London, UK. [3]Ocean BioGeosciences Group, National Oceanography Centre, Southampton, UK. ✉e-mail: h.carter@nhm.ac.uk

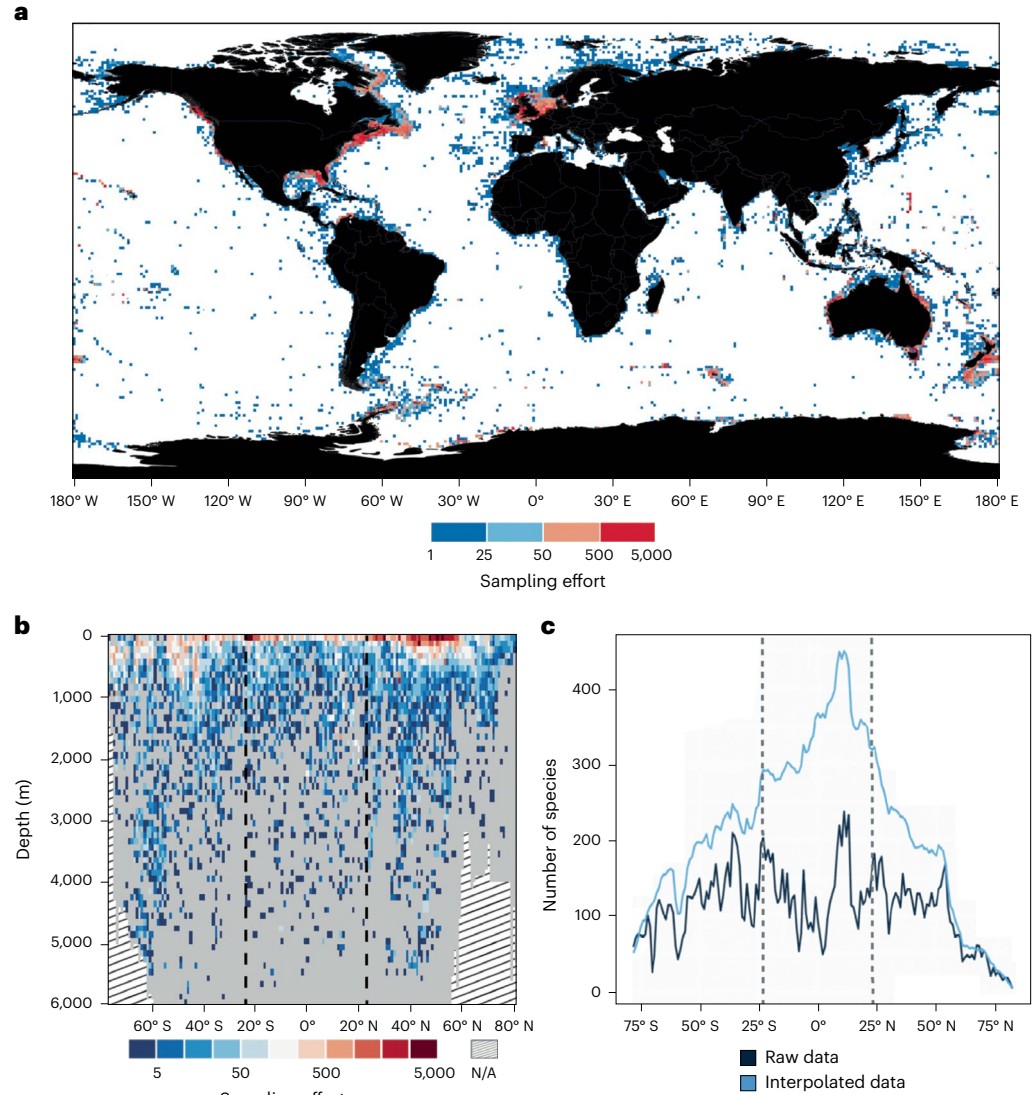

**Fig. 1 | Asteroid sampling effort and latitudinal diversity. a**, Map of sampling effort at 1 × 1° resolution across the entire dataset used here. Blue shades indicate lower levels of sampling effort while red shades indicate higher sampling effort. Note that sampling is concentrated around the continental margins with few collections from the deep continental basins. **b**, Collection effort plotted against latitude and depth. Sampling is highest in shallow and upper bathyal waters, peaking at high latitudes. Sampling of the deep sea below 3,000 m is relatively scarce, but in our analyses, this is largely mitigated by the larger range size of abyssal species. Graphical areas with no data because the graph depth exceeds the deepest ocean depth at that latitude are indicated by sloped lines. Vertical dashed lines represent the northern and southern extent of the tropics. **c**, Latitudinal species diversity of asteroids. The raw sampling data (dark blue) show a general increase towards the tropics but no trend within the tropics, while interpolated species richness (light blue) shows a pronounced peak in the tropics, just north of the Equator. Vertical dashed lines represent the northern and southern extent of the tropics. Map shapefile obtained from Natural Earth.

## Results and discussion
### Global patterns of species richness
At a global scale, species richness peaks ~10° N of the Equator in shallow tropical waters (Fig. 2a,c) with a strong decline in species number throughout temperate waters on both sides of the Equator, culminating in low numbers of species at high latitudes, particularly in the Arctic. Species richness at 60° S/N is ~65% and ~85% lower, respectively, than that at the tropical peak (Fig. 2c). Grouping shallow species ranges by marine eco-province[13] identified the central Indo-West Pacific (IWP) and to a lesser extent the Atlantic Caribbean Basin as hotspots of tropical diversity (Fig. 3a). While this global pattern is similar to that seen in other terrestrial and shallow marine groups[14–16], this is in contrast to a previous study of shallow-water asteroids that proposed low tropical species richness, on the basis of data from reef and rocky substrate species representing only ~10% of the taxa included in the present study[8]. Richness is not uniformly or exclusively high throughout the tropics

and is notably lower in the eastern and southern Atlantic and around the central Pacific islands, while being relatively elevated along the more temperate northern Pacific coastlines. This uneven distribution of shallow tropical richness probably reflects regional variation in habitat availability, particularly where shallow habitats are reduced around islands and steeply sloping continental margins, as has been found in other studies[17,18]. However, the elevated North Pacific peak in species richness is better explained as reflecting a regional radiation of a small number of genera with range-restricting life history strategies[19,20] and is not maintained at higher taxonomic levels (Extended Data Fig. 1 and Supplementary Table 1).

At temperate latitudes, conversely, species richness is highest in deeper water, generally below the first 50 m of the water column (Fig. 2a). Tracing species richness across latitude with increasing depth (Fig. 2b) reveals that the narrow peak of richness observed in the shallow tropics graduates into broader bimodal, but uneven, peaks

of bathyal temperate richness (35–50° N, 40–60° S) that are maintained into the lower bathyal and abyssal depth zones. Such bimodal temperate peaks of deep-water richness have also been observed in other taxa[9,21]. Furthermore, the number of species present in the tropics declines more rapidly with increasing depth (shallow-to-bathyal decline in species richness: tropical, ~40%; temperate, ~15%; polar, ~20%) than at higher latitudes, particularly in the southern hemisphere (Fig. 2c–e). It should be noted, however, that the greatest overall rate of species richness decline observed in this study was between northern temperate latitudes and the high Arctic (~75%; Fig. 2c).

Species richness in the lower bathyal and abyssal depth zone remains highest outside of the tropics in both hemispheres. This unimodal per hemisphere distributional pattern has been noted in other taxa[9,22], although we note that the magnitudes of these peaks are much lower than those seen in shallow waters (Fig. 2e). The relative evenness of species richness across the deep sea when compared with that across shallow waters reflects both the decrease in sampling effort and observed species numbers at these extreme depths (Fig. 1c) but also supports the apparent tendency for deep-water clades to occupy more cosmopolitan ranges across larger and less variable habitats (Fig. 4f) at these depths[23,24].

## Patterns of species richness by hemisphere and ocean

While the overall transition from a unimodal tropical peak in species richness to bimodal temperate peaks with increasing depth occurs in both hemispheres, we show a striking divergence in the depth at which relative temperate richness exceeds that of the tropics on either side of the Equator (Fig. 2b–e). Below ~100 m species richness is higher in southern temperate waters than in southern tropical waters. This is not true in the northern hemisphere, where species richness is higher in northern temperate waters than in northern tropical waters only below ~750 m (Fig. 2).

At all depths, species richness is also notably higher north of the Equator, except for polar waters, where, by contrast, the Antarctic appears to be far more species rich than the Arctic (Fig. 2). At abyssal depths the difference between polar regions is partly due to the Arctic being shallower (Fig. 2a,b), exceeding 4,000 m only in the Amundsen Basin and in a small region known as the Molloy Hole, and thus having less available deep-water sea-floor area than the much more extensive deep water in the Southern Ocean[25]. This does not, however, explain the observed patterns in shallower depths or outside of polar waters. A similar hemispheric asymmetry in the strength of a latitudinal pattern on either side of the Equator has been noted previously in benthic molluscs and crustaceans[26–28], and a tendency for studies to find higher species richness in the northern hemisphere is well documented[29]. This pattern in shallower waters may reflect the relative lack of continental shelf area in the tropical and temperate zones south of the Equator[30], although this may be a non-trivial relationship to untangle requiring further investigation and probably plays a larger role when considering diversity between genera or families (Fig. 4). Further potential explanations include this being an artefact of uneven sampling and taxonomic effort across latitudes[29,31]. Lower diversity in the Arctic may further be expected given the younger age and relative isolation of the Arctic Basin fauna compared with the Southern Ocean fauna[32].

Patterns of species richness across the same range of latitudes are also notably different between the Atlantic (439 species, 352 endemics) and the 'extended' IWP (encompassing temperate Australia and New Zealand north to Japan; 1,029 species, 944 endemics), which each have highly distinctive faunal compositions (Figs. 3b and 5). Atlantic species richness (Fig. 5a) is greatest in the northern tropics, peaking around 25° N in upper bathyal water and largely reflecting high Caribbean diversity. This tropico-temperate peak persists into the bathyal and abyssal zone, gradually becoming more northern temperate with increasing depth in a similar, but less pronounced, trend to that observed in the global dataset (Fig. 2). In the southern tropical and southern temperate Atlantic, species richness is much lower than north of the Equator at all depths except for a shallow/upper bathyal peak encompassing southern South America ~40–50° S. Contrastingly, the extended IWP is dominated by a shallow tropical fauna concentrated south of the Equator around the Celebes Sea and northern Australia with a species count that declines rapidly with increasing depth below ~100 m throughout the region (Fig. 5b). Unlike in the Atlantic, temperate and tropical species richness is relatively even in both bathyal and bathyal and abyssal extended IWP waters, although species richness is lower north of the Equator. Lower levels of species richness across most of the northern temperate extended IWP are partially explained by there being less available sea-floor area, since more of the total area at these latitudes is land than in the southern temperate regions.

Although we are unable to fully exclude the impacts of historic taxonomic expertise, the size and taxonomic completeness of our dataset allow us to suggest that sampling bias is unlikely to entirely explain our observed hemispheric asymmetry and pronounced differences between oceanic realms. The raw dataset used in this study (Fig. 1a,b) has greater overall coverage at temperate latitudes, but across the tropics where the major hemispheric asymmetry in species richness shown in this study occurs (Fig. 2b), there is no substantive variation in sampling effort, in contrast to higher latitudes, where sampling effort is heavily weighted towards the northern hemisphere, as has been found in other studies[24,29]. Rather, we suggest that these patterns are at least partially linked to differences in the extent of the topographic heterogeneity of coastline and sea floor that exist between these regions[33] and to differences in the relative impacts of important environmental variables. Similar patterns have also been observed for another class of echinoderms, the Ophiuroidea[9,34]. Ophiuroidea differ, however, because studies have shown a southern overall peak in richness in contrast to the northern peak in asteroids. This may reflect the effects of evolutionary history, possible niche competition or differential habitat preference, but this requires further investigation[35].

## Geographic and environmental factors driving marine diversification

Species diversity and habitat diversity are closely linked[36], and geographically complex regions with greater habitat heterogeneity and more barriers to dispersal might be expected to harbour greater species richness[24,37]. Marine habitat spatial heterogeneity has rarely been investigated at scale[36,38,39] but is here roughly estimated as a function of the length of available coastline. In the Atlantic, spatial heterogeneity and coastline length in the northern tropics (48,831 km) is more

---

**Fig. 2 | Global bathymetric and latitudinal species gradients for Asteroidea. a**, Global bathymetric plot of species diversity at depth intervals spanning the intertidal to lower abyssal zone (0–6,000 m) for latitudes between 78° S and 82° N and the full extent of longitude. Species richness is binned at intervals of 10 m (shallow), 50 m (bathyal) and 200 m (abyssal). Graphical areas with no data because the graph depth exceeds the deepest ocean depth at that latitude are indicated by sloped lines. Blue colours indicate fewer species, and red colours indicate more species; the colour scale gives the total number of species present in a given cell. Vertical dashed lines indicate the Equator and approximate extent of the tropics. **b**, Bathyal sections with independently scaled species gradients.

Diversity peaks in shallow tropical waters (0–200 m), with relative diversity gradually becoming bimodal and more temperate in the bathyal zone. Overall diversity is highest in the northern hemisphere across all depths. **c**, Species diversity curve for global diversity at 100 m. Diversity peaks strongly in the tropics. **d**, Species diversity curve for global diversity at 1,000 m. Diversity peaks bimodally on either side of the Equator with a pronounced southern tropical minimum. **e**, Species diversity curve for global diversity at 3,000 m. Diversity is highest at the northern tropico-temperate transition with a smaller southern peak and distinct southern tropical minima. The red vertical dashed lines indicate the approximate extent of the tropics.

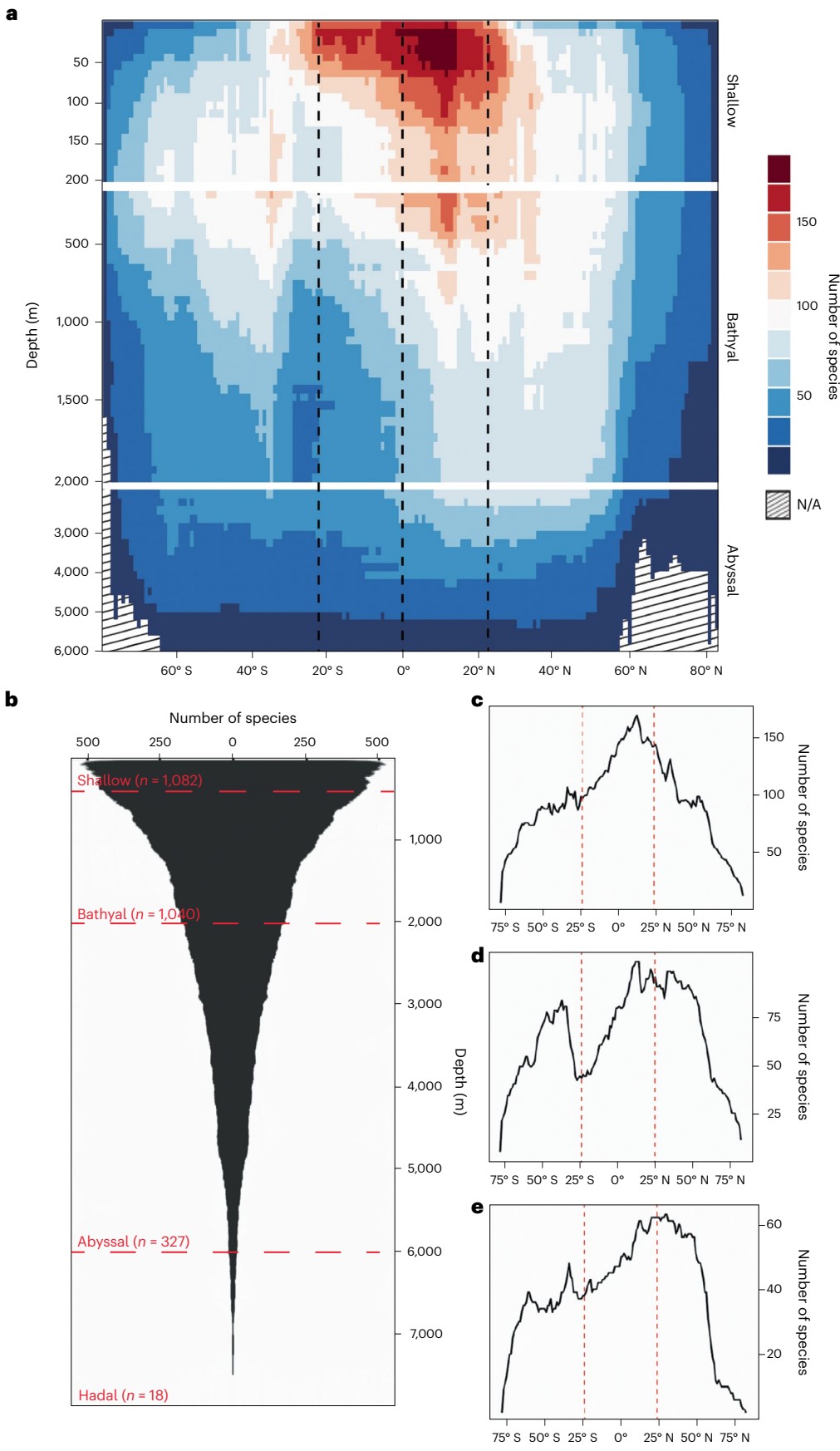

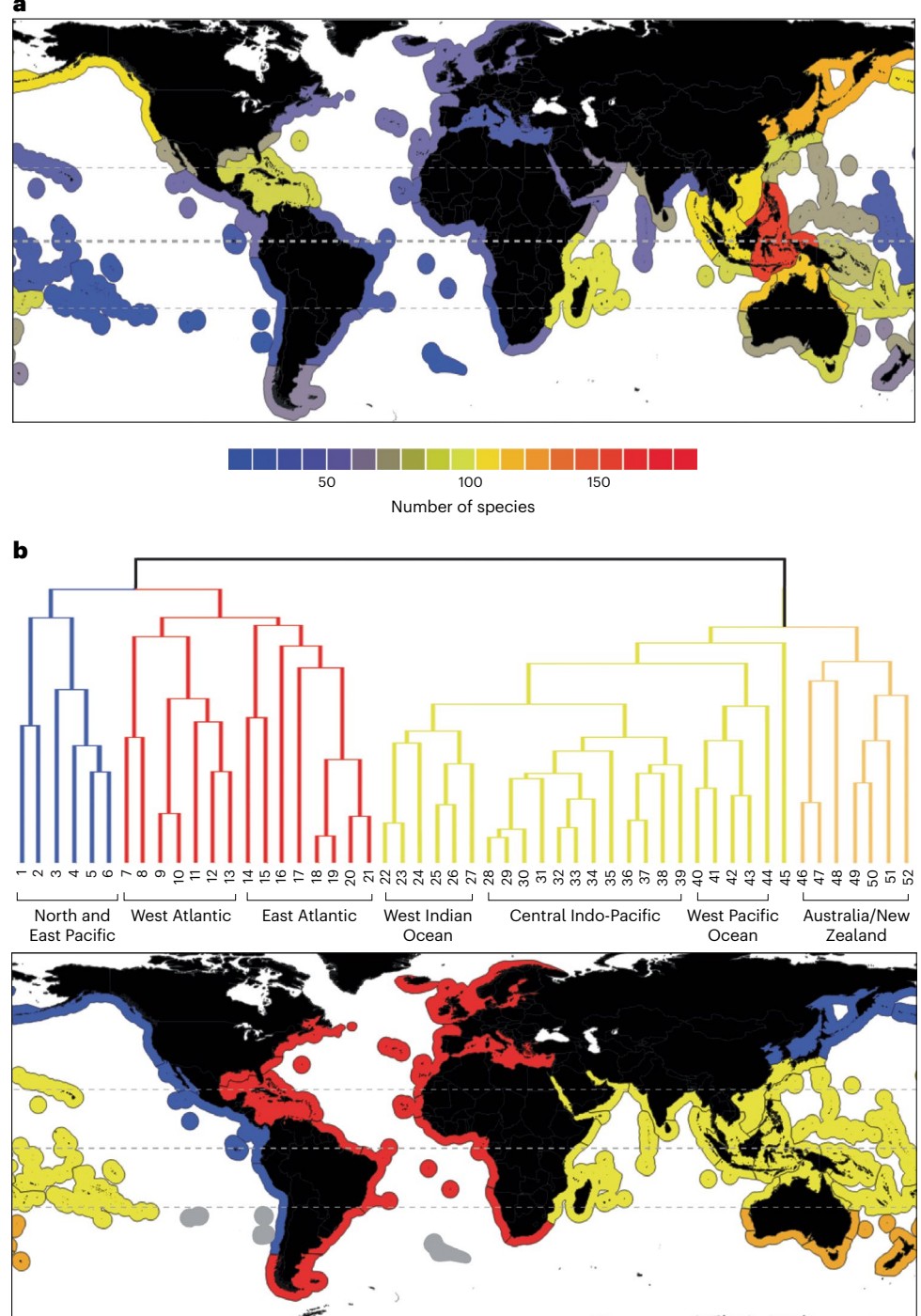

**Fig. 3 | Global shallow-water (0–200 m) temperate and tropical species diversity for the Asteroidea (excludes polar regions). a**, Species-level diversity per marine eco-province[12]. Tropical diversity (within the outer horizontal dashed lines) is by far the highest in the central IWP, with the next most diverse region containing less than two thirds the number of species. Temperate diversity is also surprisingly high in the North Pacific. Diversity is lowest around Rapa Nui and the Juan Fernández Islands. **b**, Unweighted pair group method with arithmetic mean tree of beta diversity showing relationships among eco-provinces and map showing the distribution of major clades. The tip numbers refer to the eco-provinces listed in Supplementary Table 1. Three major biogeographic clusters were identified: Atlantic (red), north and east Pacific (blue) and IWP (yellow indicates tropical; pale orange indicates temperate) (the colours match the locations in the map below). The relationships of faunas associated with three small island groups with high endemicity (grey) were not clearly resolved. Map shapefile obtained from Natural Earth.

than 2.5 times greater than in the southern tropics (18,185 km), driven principally by the complex basin and island systems in the Caribbean[40]. This matches patterns of higher asteroid richness north of the Equator (Fig. 5), although additional factors such as the major freshwater discharges from the Amazon and Congo Rivers in the southern tropics, which vastly increase turbidity and sea-floor sediment deposition, are likely to contribute to the lower faunal count observed south of the Equator[41,42]. The tropical IWP is contrastingly far more even in spatial complexity and coastline extent, with the northern tropical coastline (122,730 km) only 1.25 times longer than the southern tropical coastline

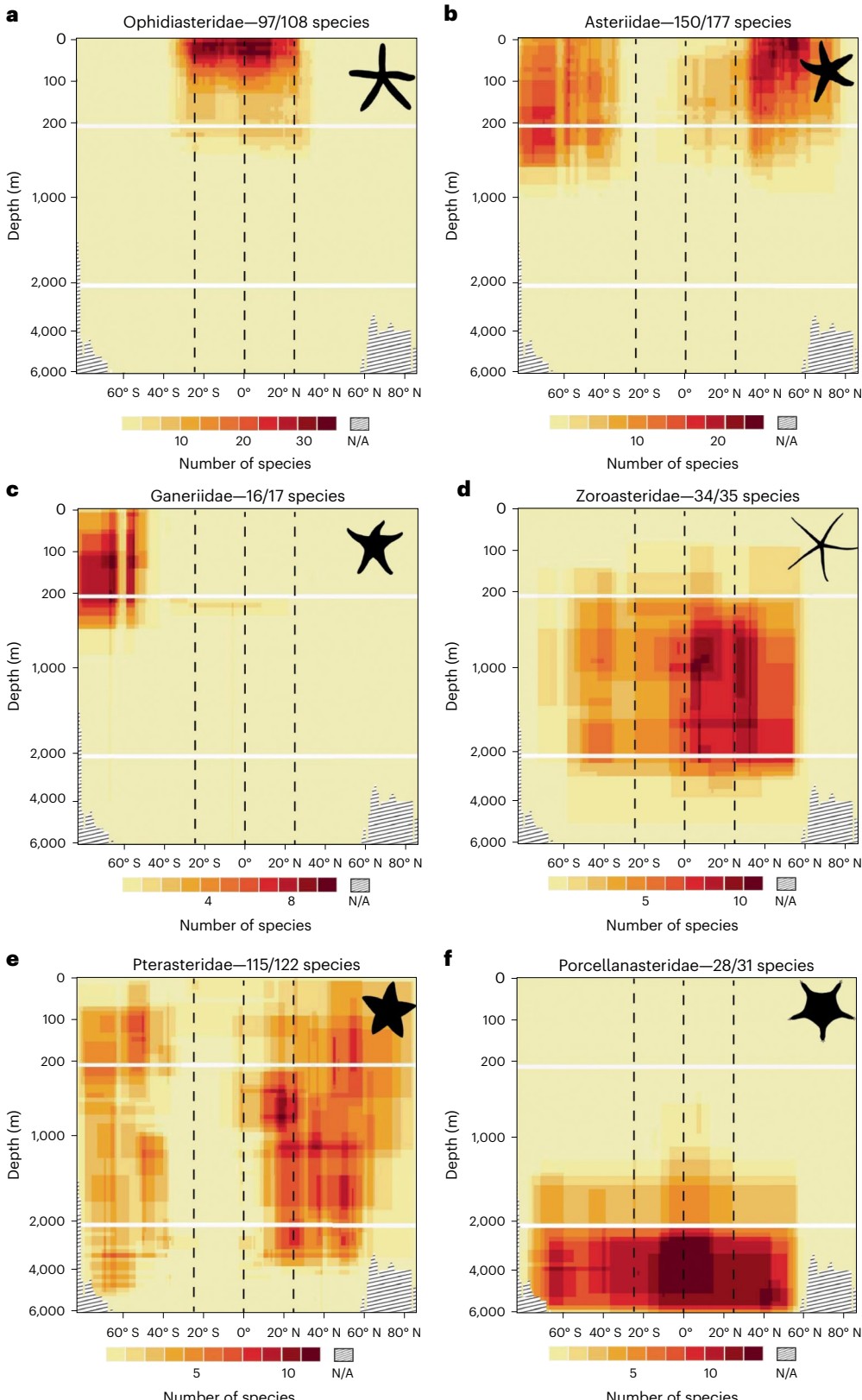

**Fig. 4 | Global bathymetric diversity plots for six exemplar asteroid families.**
**a**–**f**, The majority of shallow-water families are tropical specialists (for example, **a**), although some are bimodally distributed (for example, **b**) in temperate waters or restricted to the Antarctic (for example, **c**). No families are solely restricted to Arctic waters. Primarily bathyal families occupy greater latitudinal ranges than shallow families, with most centred in the tropics (for example, **d**), or bimodal on either side of the Equator (for example, **e**). Abyssal families tend to be largely cosmopolitan (for example, **f**), centred in or just north of the tropics. Graphical areas with no data because the graph depth exceeds the deepest ocean depth at that latitude are indicated by sloped lines. Vertical dashed lines indicate the Equator and approximate extent of the tropics. See Supplementary Fig. 2 for all families.

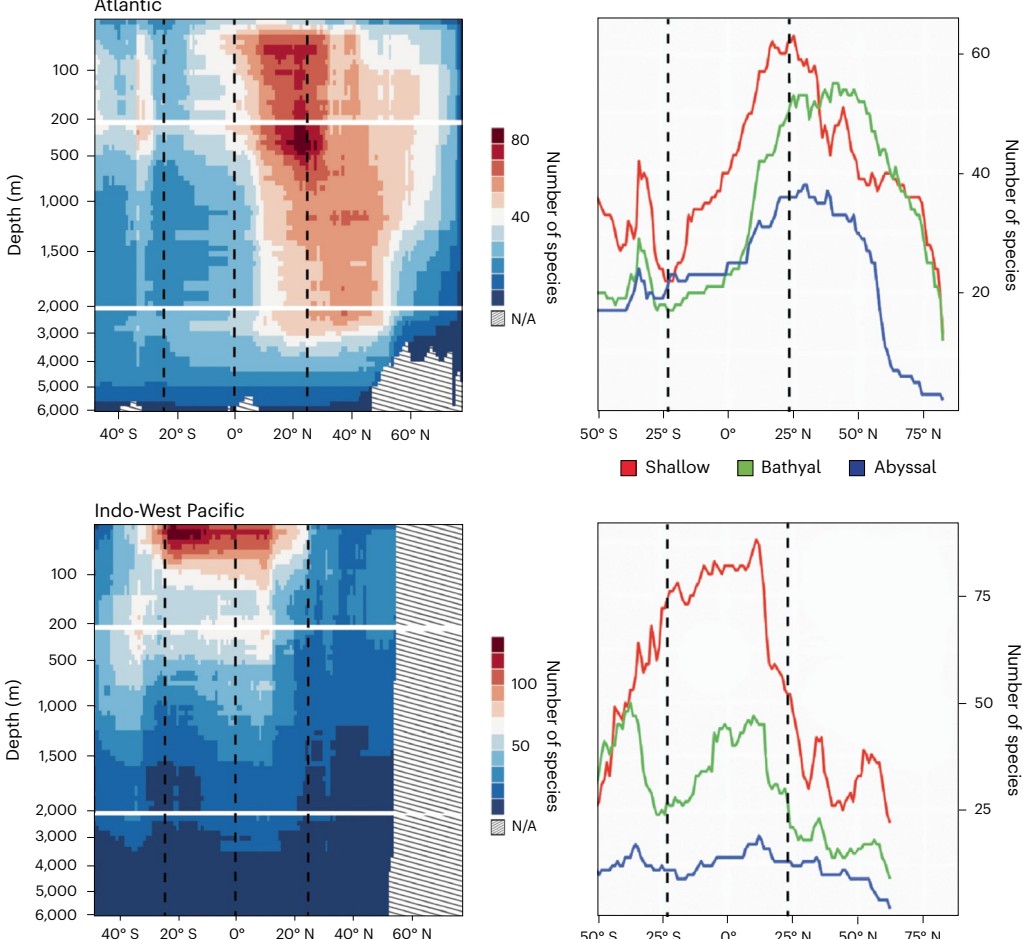

**Fig. 5 | Ocean-specific species gradients.** Patterns of diversity between the Atlantic and the extended IWP are markedly different, with diversity concentrated north of the Equator in the Atlantic and south of the Equator in the IWP. Overall diversity is lower and peaks at greater depths in the Atlantic. Atlantic diversity is particularly low in the southern tropics, and IWP diversity is low in northern temperate waters. There is a pronounced shift in the peak of diversity towards southern temperate waters in the IWP and towards northern temperate waters in the Atlantic. The line graphs show sections at 100 m (red), 1,000 m (green) and 4,000 m (blue) for each ocean. Graphical areas with no data because the graph depth exceeds the deepest ocean depth at that latitude are indicated by sloped lines. Vertical dashed lines indicate the Equator and approximate extent of the tropics.

(97,727 km). This evenness is reflected in the more equal distribution of species across the tropical IWP (Fig. 5), and these patterns in both oceans support an important role for spatial factors in the promotion of species richness.

Although such geographic factors are clearly important in promoting species richness, this is also shaped by interactions with environmental variables, and these are believed to have increasing importance with depth[4]. At a global scale, species richness across the sea floor appears to be strongly linked to variation in energy availability (the species–energy hypothesis)[43], and our results broadly support this theory. However, we show that the principal sources of energy at the sea floor—thermal (temperature) and chemical (here represented as particulate organic carbon (POC) flux)—vary in relative importance between the shallow, bathyal and abyssal zones. Higher temperatures at lower latitudes correlate with increased tropical richness in global shallow waters (Extended Data Tables 1–3; $P < 0.001$), while in bathyal waters, the relative shift towards higher species numbers at temperate latitudes is best characterized by the cumulative interaction of both increased temperature and POC flux in concert (Extended Data Table 2; for temperature, $P < 0.001$; for POC, $P < 0.001$). In lower bathyal and abyssal waters, nutrient availability has previously been considered the principal driver of diversity in what is broadly a cold and highly thermally homogenous environment[9,43,44]. Conversely, the results of

this study suggest a more important role for temperature over POC flux in this bathymetric region (Fig. 6, Extended Data Tables 1–3 and Extended Data Fig. 2; for temperature, $P < 0.001$). These results were generated using two different statistical model approaches, stationary linear models (SLMs) and random forest regression analyses, and are robust across multiple iterations accounting for both the effects of sampling effort and potential overestimation through our approach of using informed interpolation (Extended Data Tables 1–3).

The Atlantic and extended IWP also show significant correlations with the same environmental factors despite differing species richness patterns, although the relative importance of these factors across bathymetry varies between regions. In the extended IWP, temperature is the most significant correlate across all depth zones ($P < 0.001$; Extended Data Tables 1–3 and Extended Data Fig. 3), with POC flux showing a significant relationship only at lower bathyal and abyssal depths ($P < 0.001$). This increased role for temperature suggests that the rapid drop-off in sea-floor temperature with increasing depth in the extended IWP is a major driver for the pronounced decline in species richness below the shallows (Fig. 5, Extended Data Fig. 3 and Supplementary Fig. 1). The environmental variables tested in this study show weaker correlations with Atlantic species richness patterns, perhaps suggesting that other variables, not tested here, also play an important role. This is particularly evident in the shallows, where spatial

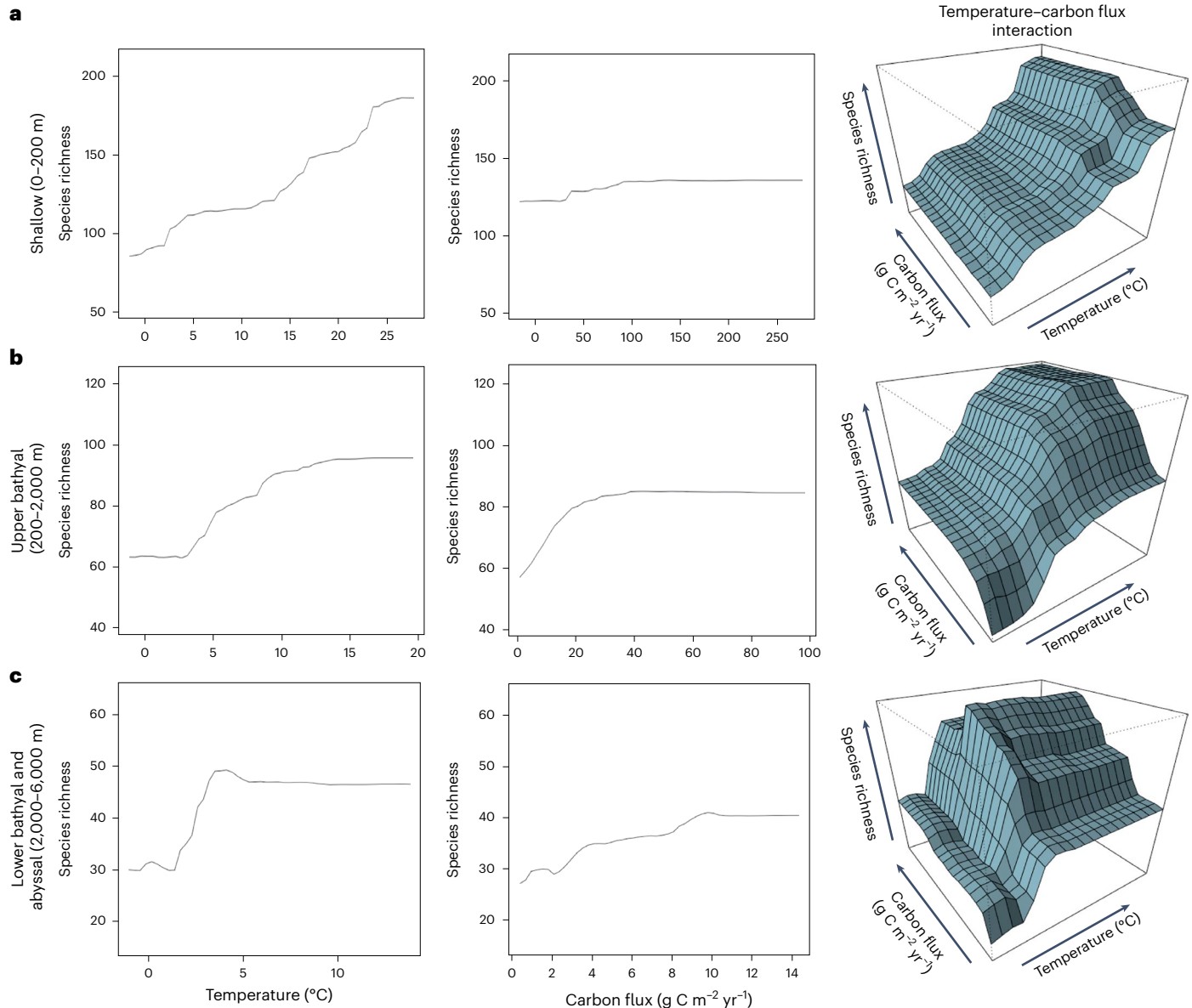

**Fig. 6 | Global pairwise interaction plots for temperature and POC flux.**
**a**, Pairwise interaction plot for the shallow depth zone (0–200 m). Increasing temperature correlates with increased species richness, but no real interaction was observed with POC flux. **b**, Pairwise interaction plot for the upper bathyal depth zone (200–2,000 m). A combination of increased POC flux and temperature correlates with increasing species richness such that species count is lowest where both are low and highest where both are high. **c**, Pairwise interaction plot for the lower bathyal and abyssal depth zone (2,000–6,000 m). Low temperatures correspond to low species richness regardless of POC flux levels below a thermal threshold. Above this threshold, increased richness correlates closely with increased POC flux, but further increasing temperature has no notable effect.

and habitat heterogeneity may be of greater importance, but POC flux and temperature become significantly correlated with richness with increasing depth ($P < 0.001$; Extended Data Tables 1–3 and Extended Data Fig. 4). Within these regional analyses, we also found significant correlations with salinity at bathyal and abyssal depths (in the Atlantic and extended IWP, $P < 0.001$; Extended Data Tables 1–3), an environmental correlate that is not significant in the global analysis. Although some marine regions, such as major river estuaries or semi-enclosed basins, do show pronounced salinity differences when compared with most of the global oceans, salinity does not vary significantly across the majority of the sea floor (Extended Data Figs. 2–4) and not in a way that is likely to be biologically meaningful in restricting asteroid distributions[45]. Rather, we interpret this apparent relationship in terms of salinity acting as a proxy for major water mass boundaries, which may be important delimiters of species ranges within basins[46].

## The role of temperature in deep ocean richness patterns

A significant, positive relationship between benthic species richness and temperature has been suggested for only the shallow and upper bathyal portions of the water column, with warmer water generally associated with higher species richness. Conversely, the lower bathyal and abyssal oceans are generally considered to be largely thermally homogenous and show no significant relationship between species richness and temperature[9,21,23]. However, at polar latitudes, these depths are markedly colder than average for this depth zone, and species richness is low, despite receiving a contrastingly large proportion of total annual POC flux (Fig. 2 and Extended Data Figs. 2–4).

A pairwise comparison of the impacts of temperature and POC flux on diversity in this region (Fig. 6 and Supplementary Fig. 1) suggests that low species richness is linked not to a linear interaction between the two variables but to the presence of a lower thermal

threshold around 1–1.5 °C below which high species numbers cannot be maintained. Below this threshold, increasing levels of POC flux correspond with only negligible increases in species richness, but above this threshold they display a strong positive interaction that appears to be independent of additional thermal effects (Fig. 6c). This threshold effect may not be solely linked to thermal effects alone. The seasonality and increasing diatom content of POC flux at temperate and polar latitudes[47] compared with the tropics, and the reduction in the calcium-carbonate-based fraction of marine snow at the poles and below the carbonate compensation depth[48] when compared with the rest of the global oceans, may act as further stressors and impose additional ecological constraints on species richness and diversity at polar latitudes and bathyal and abyssal depths, as has recently been shown in the Clarion–Clipperton Zone[49]. The abyssal echinoderm fauna is also probably relatively recently derived from shallower taxa[50] (but see ref. [51]) and so may have had less time to develop the complex community structures characterizing older assemblages.

Nevertheless, this pattern of richness reduced below a thermal limit between -1 and 1.5 °C is replicated at varying magnitudes across every lower bathyal and abyssal oceanic division studied in these analyses (Fig. 6 and Supplementary Data 2). The same reduction in species richness is not shown in any of our analyses at lower latitudes, where nutrient seasonality is also high, but temperatures are above this thermal threshold (Fig. 6 and Supplementary Fig. 1). In the southern hemisphere, lower bathyal and abyssal POC flux peaks at ~40° S and is more significantly correlated with species richness ($P < 0.05$) than in the northern hemisphere ($P > 0.05$), where POC flux peaks in colder polar waters (~65° N), but in both instances, we show the presence of the same thermal threshold (Supplementary Fig. 1). This thermal threshold effect may operate in similar way to the mechanisms proposed in the gill-oxygen limitation[52] and oxygen- and capacity-limited thermal tolerance[53] (theories whereby body size is limited at very low temperatures). These theories have received recent empirical evidence in polar fish[21], but any such physiological mechanism remains to be identified in echinoderms. While seasonal POC flux variability may play a role in shaping deep-sea species richness, with the magnitude of this yet to be tested, this process appears to be principally governed by this thermal threshold effect. Low sea-floor temperatures thus prevent the establishment of species-rich asteroid communities regardless of the levels of chemical energy entering the system[44].

### Phylogenetic effects

A robust, complete phylogeny is lacking for the Asteroidea, but the assignment of extant starfish genera to families is generally robust[54–56], although a small number of clades are too taxon poor or too infrequently collected to draw firm conclusions. Examination of species richness at the family level provides the opportunity to investigate the effects of phylogeny on bathymetric diversity and to investigate the impacts of taxonomic scale. A large proportion (11/38) of families, including several of the most species rich, are restricted primarily to the shallow tropics (see Fig. 4a for one example and Supplementary Fig. 2 for more examples), contributing to the high diversity observed in this region, whereas only a small number of shallow families are anti-tropical (for example, Fig. 4b and Supplementary Fig. 2). Bathyal families tend to encompass broader latitudinal ranges as environmental conditions become less variable and tend to either be centred on the tropics (for example, Fig. 4d and Supplementary Fig. 2) or have distinct southern tropical minima (for example, Fig. 4e and Supplementary Fig. 2). Primarily abyssal families occupy practically all available deep-sea latitudes (Fig. 4f and Supplementary Fig. 2), absent only from sub-zero polar waters. Conversely, we identified a small number of shallow and upper bathyal families endemic to Antarctica (for example, Fig. 4c and Supplementary Fig. 2) and note that overall taxonomic diversity at all levels is much higher at high southern latitudes than the equivalent northern latitudes, in line with previous findings for echinoderms[52]. It is

clear, therefore, that the strength, direction and presence of latitudinal gradients vary greatly with taxonomic scale, often becoming apparent only when aggregated to the class level. Although the contribution of each family to the overall richness pattern of the class can, in this study, be relatively readily untangled, our results show the crucial importance of selecting the correct taxonomic scale for analyses of this kind.

## Conclusions

We have shown that benthic species richness is strongly limited by a minimum temperature threshold, around 1.5 °C, observed across all bathyal and abyssal regions in this study. Overall, the bathymetric profile of starfish biodiversity supports a species–energy framework, driven at all depths above the thermal threshold by variation in thermal energy, with nutrient availability (POC flux) of increased importance in deeper, more thermally homogenous waters. The bimodal peaks of temperate richness apparent at bathyal depths are closely linked to elevated POC flux on either side of the tropics, but at abyssal depths, the relatively even distribution of species at lower latitudes is restricted near the poles by a deep-water thermal threshold. The magnitude of this hard thermal boundary suggests that such threshold effects may play a greater role in determining deep-sea richness patterns than has been previously recognized. Our results, encompassing sampling across all major basins and bathymetric and phylogenetic levels for a single class of marine invertebrates, also show that while broad patterns of diversity are shaped by widely recognized general trends, regionally specific evolutionary factors (including geographic complexity and clade niche specificity) are of substantive importance at different scales.

Despite 250 years of collection effort, our understanding of how deep-sea faunas are established, are maintained and evolve over time and how the interconnections among bathymetric zones shape these processes remains poor[23,57], and we note the critical need for robust, taxonomically complete phylogenies against which evolutionary and historical biogeographic hypotheses can be tested alongside the vital importance of continued ecological sampling at all depths. In particular, we note the importance of sampling throughout the understudied tropics to determine the accuracy of our estimates of species numbers in the face of new discoveries, and at the poles to establish the true extents of species ranges in marginal habitats. We suggest that both regions should be primary targets for future scientific expeditions. Large-scale spatial and bathymetric descriptions of global and regional diversity for an entire class of benthic marine species are a rare resource and provide a framework against which conservation measures can be targeted across this unique realm. The need for such measures is increasingly critical in the face of the accelerating pressures of deep-sea exploitation and climate change[58].

## Methods

### Datasets

A composite dataset of species-level occurrence records was compiled from three sources: the Ocean Biodiversity Information System (OBIS) database[59], the Global Biodiversity Information Facility (GBIF)[60] and an additional set of manually curated literature resources. The final occurrence dataset including all sources (available at https://data.nhm.ac.uk/dataset/) comprises 256,861 species-level coordinate records spanning a bathymetric range of 0–7,956 m and representing 91.4% (1,751/1,916) of the currently recognized species, 98.3% (345/351) of all genera and all 38 families (Fig. 1).

The OBIS contribution comprises 268,643 georeferenced records from 517 institutional or museum-based datasets, of which 196,876 are taxonomically identified to the species level (accessed in October 2021). Prior to the analyses, names were matched to the World Register of Marine Species and manually inspected for taxonomic errors[61]. All subspecies were assigned to the appropriate parent taxon, and where taxa have been synonymized, records were assigned to the current species name. Post-validation, the OBIS dataset represents 76.6%

(1,467/1,916) of the extant described taxa and includes representatives of 93.4% (328/351) of extant genera and all 38 extant families.

The GBIF contribution (see Supplementary Data 1 for the full list of contributing datasets) comprises 476,834 records assigned to 2,779 names, of which 370,208 represent georeferenced, species-level observations (accessed September 2022) and include 52 species not present in the OBIS dataset. The GBIF and OBIS datasets share large numbers of records, although at broad scales OBIS has better coverage along the North Atlantic and North Indo-Pacific coastlines, and the GBIF has better coverage along the South African and the western North and South American coastlines. The two datasets are complementary over large portions of infrequently sampled deep ocean, particularly in the Pacific, where many records are present in only one dataset. Validation was performed in the same way as for the OBIS dataset except that records flagged as derived from the iNaturalist citizen science repository were removed. This additional step was taken to reduce misidentification issues and to ensure that the final dataset comprises only expert-identified records, primarily from research cruises and museum collections. The post-validated GBIF dataset represents 240,182 records comprising 1,574/1,916 (82.2%) species, 331/352 (94.3%) genera and all 38 families (Supplementary Data 2).

An additional 9,602 coordinate occurrence records covering 1,159 species were derived from 259 literature sources, primarily taxonomic or biogeographic monographs, along with records transcribed from collection labels at the Natural History Museum, UK. These records span ~150 years of collecting effort and include data from prominent research cruises not previously digitized or represented in OBIS or GBIF, including Challenger, Discovery, the British, Australian and New Zealand Antarctic Research Expedition, John Murray, and Investigator, and were targeted towards geographic and taxonomic gaps in the GBIF and OBIS datasets. Coordinate data were standardized to decimal degrees and bathymetric data were standardized to metres prior to incorporation into the dataset. Where location data were reported as trawled extent, centroid points were derived to assign each record a single coordinate. In a small number of cases where no coordinate data were given but the exact locality could be derived from locality descriptions, GIS software[62] was used to assign records an estimated decimal coordinate.

| Dataset | Total records | Cleaned records | Species |
|---|---|---|---|
| GBIF | 476,834 | 240,182 | 1,574 |
| OBIS | 268,643 | 196,876 | 1,467 |
| Literature compilation | 9,602 | 9,602 | 1,159 |
| Total (duplicates removed) | | 206,861 | 1,751 |

A separate catalogue of descriptive geographic and bathymetric ranges for 99.3% (1,903/1,916) and 91.7% (1,760/1,916) of extant species, respectively, was additionally compiled for use in data validation and faunal assemblage analyses (Supplementary Data 2). This catalogue was collated manually from ~750 literature resources spanning 275 years of global research, largely consisting of original descriptions, cruise reports and regional faunal monographs. This catalogue comprises full taxonomic hierarchy, bathymetric limits and descriptive geographic ranges for all included species. Species for which the bathymetric range was given as 'Intertidal' or 'Shallow' were assigned the generic depth range of 0–10 m for inclusion in Fig. 2b and included in the 0–100 m depth cell for all further analyses. Species were further categorized as belonging to the Atlantic, IWP, north/east Pacific, Arctic or Antarctic faunas, or to cross-regional faunas, to produce broad-scale assemblages prior to further analysis. This catalogue was used to compare our collated coordinate records with species ranges derived directly from peer-reviewed literature reports for every species analysed in this study. It controls for identification and georeferencing

errors that otherwise may place species records outside of their actual range and greatly strengthens our confidence in the data underlying this study. The geographically validated dataset used in our analysis comprises a total of 176,315 occurrence records for 1,751 extant species.

### Data validation and processing

Data hosted by OBIS and GBIF primarily derive from museum and institutional databases or the results of ecological surveys[59], and they are are prone to misidentification or contain erroneous coordinate placement outside of the known geographic range of a species. Such errors tend to arise through either automated georeferencing from imprecise locality data or translocation of coordinates[63]. We analysed records for each species individually, manually comparing these against species range descriptions present in our catalogue and removing all records outside of known species ranges or those placed on land. A small number of literature-derived records were also excluded due to inconsistencies between coordinate placement and the published descriptions of the collection locality.

Bathymetric data were available for only 66.7% of locality records. Missing depths were estimated from a global bathymetric grid derived from the National Ocean and Atmospheric Administration bathymetry dataset at a resolution of 1 arcmin using the R package marmap (v. 1.0.6)[64]. Records from hadal depths (>6,000 m) were removed because taxonomic diversity was low, the data were sparsely distributed and the total number of records low, and because the trenches that characterize this depth zone have been extremely variably sampled[65].

### Marine ecoregions

To investigate shallow-water faunal compositions, we allocated each species to one or more of 232 marine ecoregions of the world[13]. Species were allocated either through a coordinate record within the boundary of the ecoregion or where descriptive ranges in our catalogue indicated their occurrence. Coordinate records were allocated using the R package meowR (v.0.6.2)[13] and manually verified using GIS software. We ultimately grouped the results by marine eco-province, 62 regions hierarchically encompassing one or more ecoregions, and used these groupings as a basis for comparisons of faunal composition and patterns of beta diversity. This maximized faunal distinctiveness between regions and minimized variability based on inconsistent sampling effort (Figs. 1 and 3).

Sorenson's coefficient[66] was used to produce pairwise comparison matrices of shared species between each eco-province as a proportion of the total and unique number of species in each area using the formula 100 × {(2 × number of species in both regions)/[(2 × number of species in both regions) + (number of species in region A only) + (number of species in region B only)]}. The resulting coefficients were clustered using an unweighted pair group method with arithmetic mean algorithm, and the results were plotted on a dendrogram (Fig. 3b). Three island groups (Juan Fernández and Desventuradas, Rapa Nui and Tristan de Cunha), dominated by a small number of endemics, could not be directly resolved within this dendrogram and were excluded from this analysis. The full faunal compositions of each eco-province are available in Supplementary Table 1.

### Latitudinal diversity gradient and bathymetry plots

The ocean, particularly the deep sea, is vast and incompletely sampled. The interpolated curve of species richness we use in this study accounts for sampling discrepancies by collapsing all records for a species within a degree of latitude to a single data point and then expanding ranges between occurrences where there are gaps. This approach greatly reduces sampling bias but may overestimate diversity by overfitting species ranges where habitats or conditions are not continuously present. To account for this, we modified the usual approach to interpolation, using additional data from our catalogue for informed interpolation. This is a bespoke approach considering

each species independently. Primary literature for each species was used to bound depth, latitudinal ranges and latitudinal/regional range disjunctions (see our catalogue).

We analysed our data at one-degree latitudinal intervals and treated our results as estimated maximum richness rather than absolute richness for each degree of latitude. Approximate species ranges were created by interpolating between minimum and maximum latitudinal, longitudinal and bathymetric extents following the interpolation methods in ref. 23 with the additional step of using the catalogue data to refine ranges. The sea-floor depth range for each degree of latitude was estimated by applying a grid of 100 equally spaced sampling points to each longitudinal degree grid square for the full extent of each degree of latitude and using this to generate maximum and minimum depths for the sea floor enclosed by each degree square. Our catalogue was used to ensure ranges were not interpolated across impossible or improbable latitudes and depths, minimizing interpolatory overfitting. Ranges were not extended across latitudes where the maximum sea-floor depth for that latitude was outside the bathymetric limits of a species and where continuous sea floor of the correct depth within a species range was not present, or where there were substantive coastline disjunctions within a species range.

Species known to be anti-tropical (for example, Fig. 4b) or to have substantial disjunct portions to their ranges from the published literature were flagged during data validation, and their interpolated ranges were manually adjusted to encompass the regions of known absence. This prevents artificial inflation of our estimates of species count across the tropics from these taxa. For analysis of the Atlantic and IWP as separate regions, the ranges of species present in both oceanic regions were manually adjusted to reflect only their range limits within each region.

Through this process, simple species ranges are transformed into complex polygons more reflective of their true regions of potential occupancy. This approach is ultimately similar to those employed by range estimation techniques that create species ranges using convex hulls[67], but it allows for the inclusion of a greater number of rare taxa, which otherwise have too few occurrence records for such hull-based techniques to adequately capture their true ranges. The informed interpolation technique used here maximizes taxonomic completeness while allowing for expert-informed manual control across known biogeographic breaks, but it remains susceptible to overestimation of species range size for some taxa, particularly if cryptic species have not been recognized or where widespread species extend over multiple small-scale disjunctions. For latitudinal analyses, however, this overestimation is greatly reduced by collapsing these data to two dimensions.

More complex species distribution models were not used in this analysis due to concerns about potential overfitting of environmental data where occurrence records are sparsely distributed, to include the maximum amount of taxonomic diversity and to avoid circularity when determining environmental drivers. Species range models require a minimum number of records to adequately capture species range niches and are considered robust only at large sample sizes, with most previous work having been done on well-documented terrestrial and not marine fauna[68–71]. A total of 22.6% of species in our dataset have fewer than ten records, and this contribution to latitudinal diversity would be lost using these models.

Species ranges based on informed interpolation were summed per degree of latitude and then split bathymetrically into three major depth strata based on those defined in ref. 9 (shallow, 0–200 m; upper bathyal, 200–2,000 m; lower bathyal and abyssal, 2,000–6,000 m). Species were further binned within each region to finer depth strata of varying resolutions (shallow, 10 m; bathyal, 50 m; lower bathyal and abyssal, 200 m). These data were used to produce bathymetric diversity plots across the global extent of latitude (Fig. 2a), for the Atlantic and extended IWP assemblages separately (Fig. 5) and for each family of the Asteroidea (Fig. 4 and Supplementary Fig. 2). Each plot spans 160° of latitude from −78° S to 82° N, encompassing the entire geographic

limits of starfish in the full dataset. The spatial extent of survey effort for each latitudinal/bathymetric cell was plotted to visualize potential global collection bias (Fig. 1a,b).

## Uncertainty

To illustrate where in our analyses a greater reliance has been placed on interpolation, we compared raw species richness across the global dataset to the interpolated values. Species were considered continuous within their known bathymetric range at each latitude but were not considered continuous across latitudes. We calculated the percentage difference in richness between our raw and interpolated data to give a measure of how much each latitude and depth cell is produced through interpolation (Extended Data Fig. 5). We additionally produced a measure of how individual species ranges rely on interpolation by looking at the disjunction of the raw data across the latitudinal range of a species. These data were separately plotted for each of the three major depth zones considered in this analysis (Extended Data Fig. 5c–e.)

## Environmental data

To investigate the role of different environmental variables in shaping global patterns of sea-floor diversity among bathymetric realms, we used linear and random forest models to compare the importance of seven environmental factors (Extended Data Figs. 2–4 and Extended Data Tables 1–3). Models were fitted independently for the global, hemisphere and ocean datasets. Average sea-floor temperature (°C), dissolved oxygen levels (ml l$^{-1}$), salinity (practical salinity units) and two proxies for water column productivity (nitrates and dissolved silicates (µmol l$^{-1}$)) were derived from the sea-floor values in the CARS 2009 dataset[72]. Phosphates were excluded from all analyses due to co-linearity issues with nitrates. Silicates were not included in our separate analyses of the Atlantic and extended IWP due to co-linearity issues with nitrates. Net primary productivity was derived from a vertically generalized production model as a function of satellite-derived chlorophyll levels[73] averaged across the years 2010–2020 (http://sites.science.oregonstate.edu/ocean.productivity/index.php). POC flux (g C m$^{-2}$ yr$^{-1}$) to the sea floor for each 100-m depth level was estimated from net primary productivity data using a productivity export model[74]. All environmental data were averaged for each latitudinal and 100-m bathymetric cell across 150° of latitude from 70° S to 80° N, spatially covering ~98.3% of records in the asteroid dataset. Environmental data for regional analyses were separately averaged within the boundaries of each oceanic realm. Oceanic realm boundaries were based on the shallow water clusters identified in Fig. 3b but limited to 50° S as an approximate margin for the Southern Ocean, which has a distinct and complex fauna largely shared across its Atlantic and IWP sectors. Although the temperate regions of Australia and New Zealand and the northwest Pacific form clusters distinct from the remainder of the IWP, we have included these within the extended IWP basin analyses for simplicity and to allow these basins to cover an equivalent latitudinal extent to the Atlantic. The Atlantic was extended into the adjacent Arctic waters as the fauna here exclusively represents the northern range limits of some temperate Atlantic species. Values for the Mediterranean were removed from the Atlantic analysis because this region is environmentally extremely distinct[75,76]. The extended IWP was extended northwards to incorporate the Cold Temperate Northwest Pacific, as the Bering Sea provided a more structured geographic boundary and allowed for direct latitudinal comparisons with the Atlantic.

We further used coastline length to establish a measure of habitat complexity for the tropics across both the extended IWP and the Atlantic. This was calculated in GIS software using the Natural Earth 10-m coastline shapefile trimmed to each region of interest.

## Statistical analysis

For statistical analyses, environmental data were subdivided across the three major depth strata (shallow, 0–200 m; bathyal, 200–2,000 m;

lower bathyal and abyssal, 2,000–6,000 m) for each of the global, hemisphere and basin datasets, with values within each bathymetric zone treated independently of depth. All analyses presented were performed using the interpolated data unless otherwise stated. The robustness of these analyses, observed trends, statistical correlates and the informed interpolation approach used were further tested by repeating every analysis across both the interpolated and raw datasets while accounting for a sampling offset (Extended Data Tables 1–3 and Supplementary Fig. 1). Sampling offset values were produced by taking a logged value of the total number of unique occurrences of all echinoderm samples per 1° latitudinal/100-m depth cell in the OBIS dataset (~1,350,000 total occurrence records). All statistical analyses were performed in R (v.3.6.3)[77].

All environmental variables were normalized (mean, 0) for each depth section for SLM-based analysis but retained as absolute values for random forest analyses. All analyses were performed on binned latitudinal data at a resolution of 1° and for depth sections of 100 m. Maximum latitudinal species richness was used as the response variable for SLMs to test hypotheses about the relationship between diversity and environmental predictors at each depth level. Environmental variables were not used to produce any of the interpolated or raw occurrence records, which eliminated circular reasoning in our analysis.

We applied an all-model selection method using the R package slm (v.1.2.0)[78] to determine the best models on the basis of Akaike information criterion scores fitting variables to a Poisson distribution. SLMs correct for spatial autocorrelation between time or geographical series and so eliminate the possible confounding impact of non-independence of latitudinal faunal counts. The top-performing model for each depth zone and the relative importance of each predictor were analysed using $z$-tests and multiple-$r^2$ values used to assess model fit (Extended Data Tables 1–3). Silicate was removed from most SLM analyses to avoid co-linearity issues due to its strong correlation with nitrate. Random forest analyses were performed using the R package randomforest (v. 4.7-1.1)[79]. Each model was run over 1,000 tree iterations using a mtry value (the number of values to sample at each node) of three splits at each node with latitudinal species richness used as the response variable to determine the importance of environmental variables in shaping latitudinal diversity. So that results were comparable between regions and bathyal zones, random forest variable importance was derived from the %IncMSE, the decrease in model accuracy if the variable in question is excluded (Extended Data Tables 1–3). Partial dependence plots for each variable across each geographic and bathymetric split were produced using the R package plotmo (v. 3.6.2)[80] and used to determine whether the importance of a variable was due more to a continual directional impact over the range of values included or to the variable functioning more as an environmental switch with diversity limited above or below a certain value. Pairwise comparisons of variable importance were produced to visualize the impact on diversity of interactions between the most important variables for each geographic and bathymetric split (Fig. 6 and Supplementary Fig. 1). Environmental variables were treated independently for each major depth stratum, and a 70:30 test:train split was used to determine the best-performing environmental predictors across each analysis.

#### Reporting summary

Further information on research design is available in the Nature Portfolio Reporting Summary linked to this article.

## Data availability

The full OBIS and GBIF datasets used to generate coordinate records for this study are available online via OBIS at https://obis.org/taxon/123080 and via GBIF at https://www.gbif.org/species/214. The fully curated species occurrence dataset and catalogue of descriptive Asteroidea ranges are all derived from publicly available sources. The geographically cleaned species occurrence dataset is available from the NHM data portal[81]. The catalogue of descriptive ranges is available in Supplementary Data 1. Environmental data excluding POC flux are available from the CARS 2009 dataset (https://www.marine.csiro.au/~dunn/cars2009); POC flux data are derived from data available at http://sites.science.oregonstate.edu/ocean.productivity/index.php. Bathymetric data were sourced from the National Ocean and Atmospheric Administration.

## Code availability

All code used in this study relied on publicly available R packages whose usage is fully defined in the published literature as referenced in the methodology. All parameters used in the SLM and random forest model set-up are described in the methodology.

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

## Acknowledgements

We thank N. Knowlton for helpful criticisms of an earlier draft and P. Oliveri and J. Thompson for useful discussion. This work was funded by London Natural England Research Council Doctoral Training Programme grant no. NE/L002485/1 (H.F.C.).

## Author contributions

Conceptualization: H.F.C. and S.T.W. Methodology: H.F.C., G.B.-C. and S.T.W. Formal analysis: H.F.C. and G.B.-C. Investigation: H.F.C. Data curation: H.F.C. Writing—original draft: H.F.C. and S.T.W. Writing—review and editing: H.F.C., S.T.W. and G.B.-C. Visualization: H.F.C. and S.T.W. Supervision: S.T.W.

## Competing interests

The authors declare no competing interests.

## Additional information

**Extended data** is available for this paper at https://doi.org/10.1038/s41559-025-02808-2.

**Correspondence and requests for materials** should be addressed to H. F. Carter.

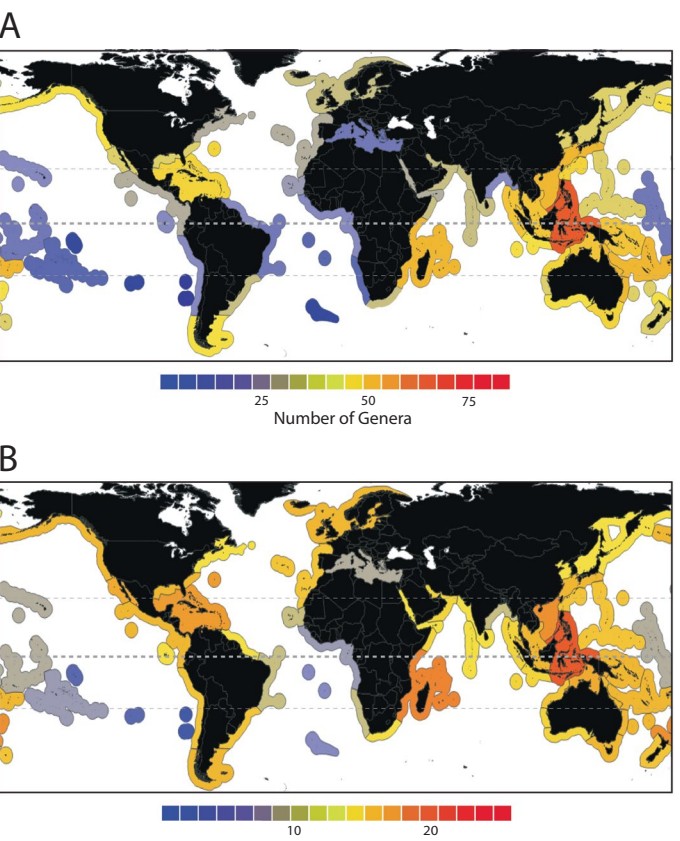

**Extended Data Fig. 1 | Global shallow water (0–200 m) temperate and tropical generic and familial richness for the Asteroidea. a)** Genus level richness per marine eco-province[18]. Tropical diversity (within outer horizontal dashed lines) is highest in the central Indo-West Pacific (IWP) with the next genus rich region containing less than 75% of the total IWP genera. Richness is lowest around Rapa-Nui and the Juan Fernandez Group. **b)** Family level richness per marine eco-province. Richness is highest in the IWP but not substantively greater than elsewhere in the tropics. Richness is notably low only within the Gulf of Guinea and around some south Pacific islands. Base map made with Natural Earth.

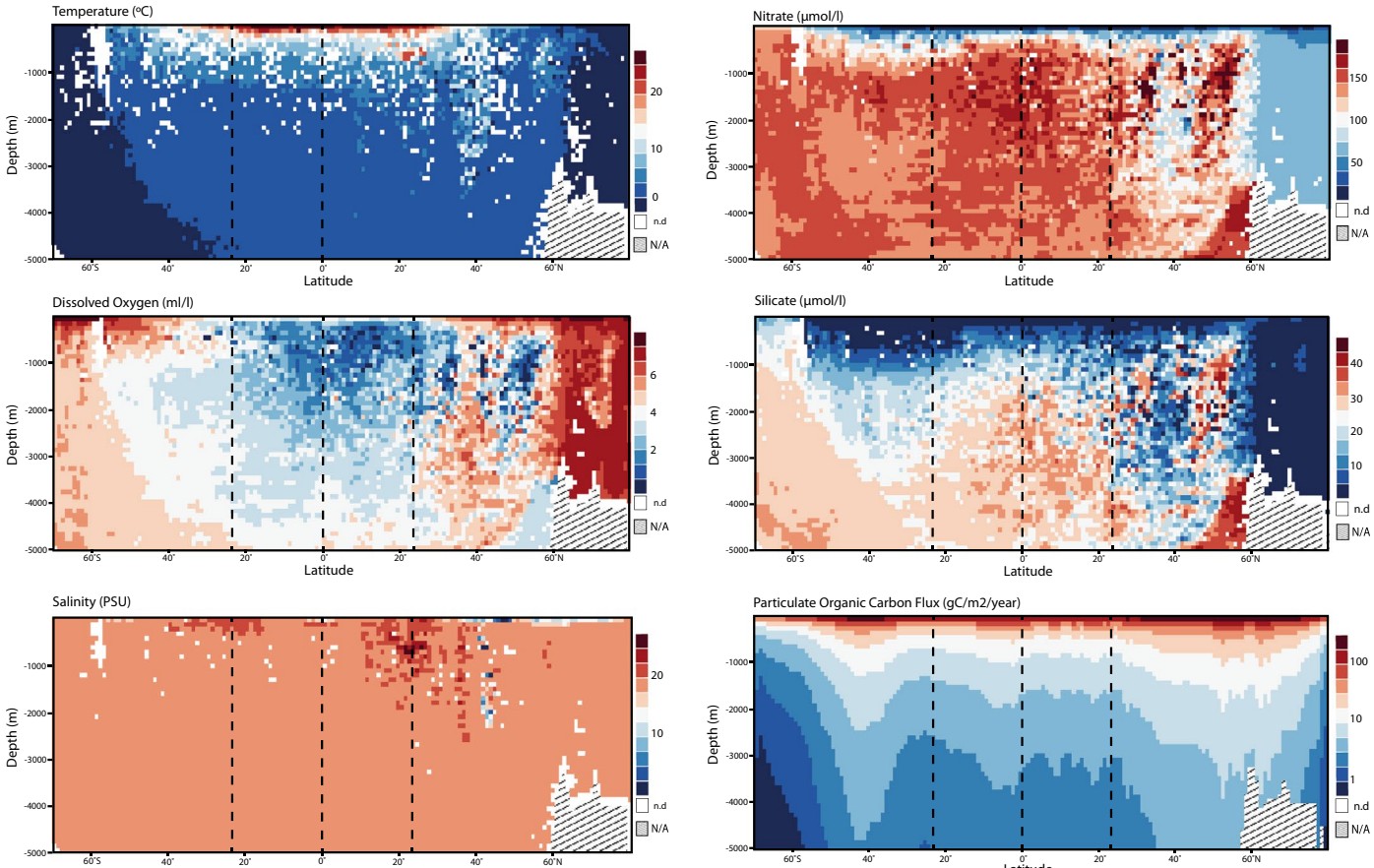

**Extended Data Fig. 2 | Global environmental plots of mean annual sea floor data at 100 m intervals from 0–5,000 m.** Raw data for all variables except particulate organic carbon flux were downloaded from CSIRO[73]. Organic carbon flux values were ultimately derived from VGPM data hosted on the Ocean Productivity database (See methods). Graphical areas with no data because graph depth exceeds the deepest ocean depth at that latitude are indicated by grey and white sloped lines. Temperature: Temperature peaks in shallow tropical and subtropical waters (to ~40° latitude) with sub-zero water at each pole. Dissolved Oxygen: Levels are low throughout tropical waters with a bathyal plume of deoxygenated water rising at the poles where it mixes with descending cold, dense, oxygen-rich water derived from the polar regions and the Greenland ice sheet. Oxygen-rich abyssal polar water spreads equatorially to lower latitudes. Salinity: Highest in warm tropical and subtropical surface waters either side of the equator with relatively fresh, low salinity surface water

spreading out from the polar ice caps. Nitrate: High in shallow waters only along the Antarctic margin, nitrate levels peak in bathyal northern temperate waters, although not uniformly so, and are generally high in all deep waters except the Arctic, where they are low at all depths. Silicate: Silicate levels are low throughout shallow water environments, bathyal temperate waters and the Arctic basin, but are generally higher in bathyal tropical and Antarctic waters and variably so in abyssal temperate waters. Particulate Organic Carbon Flux (POC flux): POC flux is high throughout shallow waters, but peaks in temperate/polar waters through bathyal and abyssal depths. POC flux peaks over a wider but more northerly latitudinal extent in the northern hemisphere than in the southern hemisphere with a substantial proportion of POC accumulating in the arctic basin. Southern deep water POC flux peaks around 40° S and is low in the Antarctic. Vertical dashed lines indicate the equator and approximate extent of the tropics.

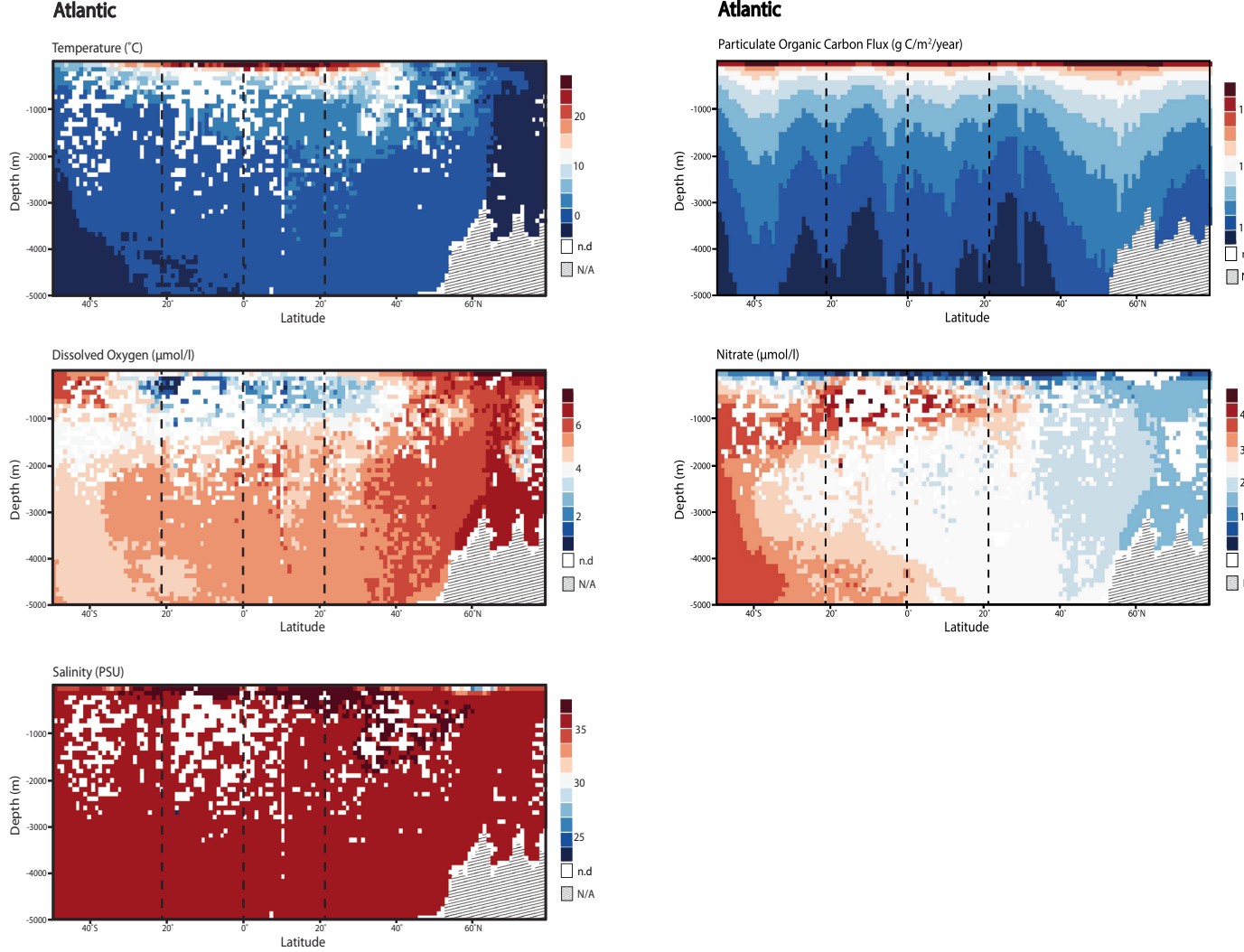

**Extended Data Fig. 3 | Atlantic environmental plots of mean annual sea floor data at 100 m intervals from 0–5,000 m.** Raw data for all variables except particulate organic carbon flux were downloaded from CSIRO[73]. Organic carbon flux values were ultimately derived from VGPM data hosted on the Ocean Productivity database (see methods). Graphical areas with no data because graph depth exceeds the deepest ocean depth at that latitude are indicated by grey and white sloped lines. Temperature: Temperature is highest in shallow tropical and temperate waters and lowest around polar waters. Bathyal and upper abyssal water north of the equator is notably warmer than the equivalent water south of the equator. Dissolved Oxygen: Dissolved Oxygen is low in shallow and upper bathyal tropical waters and highest in deeper waters and around the polar margins. Salinity: Salinity is fairly uniform throughout the Atlantic but is highest in the shallow tropics and in bathyal water around 35–40ºN corresponding with Mediterranean overflow water. Particulate Organic Carbon Flux (POC flux): POC flux is highest in shallow waters but is relatively even throughout the bathyal and abyssal zones, reaching notable minima around 10º S and 25º N, and with only a relatively small increase at temperate latitudes. Nitrate: Nitrate levels are low in shallow waters and in bathyal and abyssal waters in the northern polar and temperate regions. Nitrate levels are higher in southern tropical and temperate bathyal waters and at depths below 4,000 m, but are relatively reduced in upper abyssal tropical waters. Vertical dashed lines indicate the equator and approximate extent of the tropics.

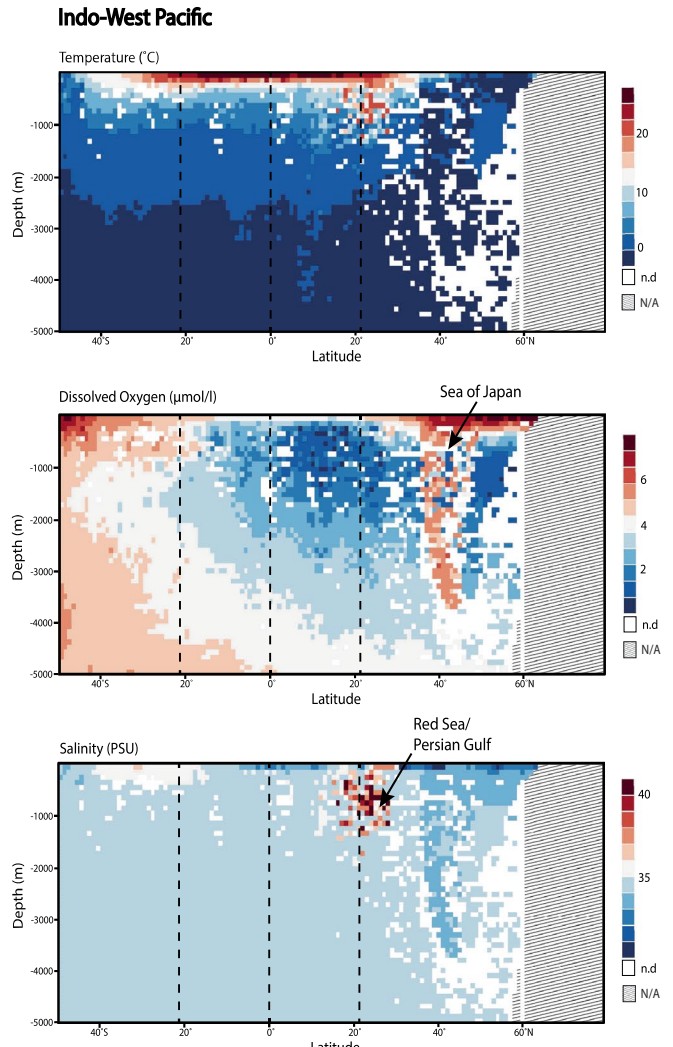

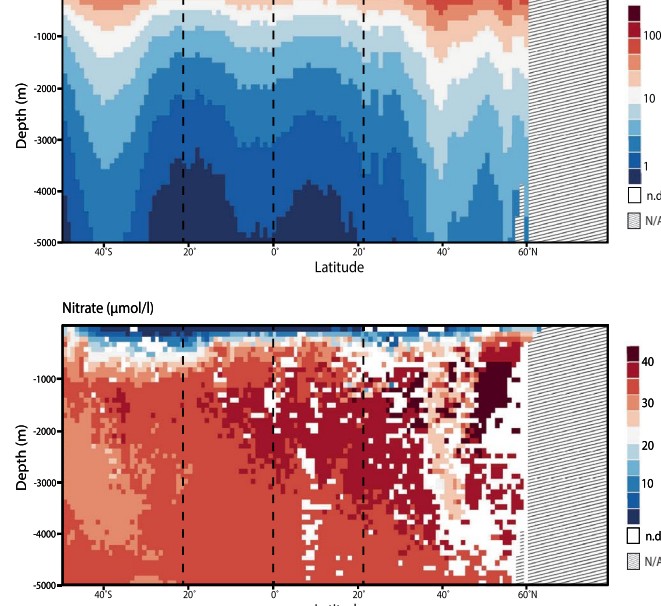

**Extended Data Fig. 4 | 'Extended' Indo-West Pacific (IWP) environmental plots of mean annual sea floor data at 100 m intervals from 0–5,000 m.** Raw data for all variables except particulate organic carbon flux were downloaded from CSIRO[73]. Organic carbon flux values were ultimately derived from VGPM data hosted on the Ocean Productivity database (see methods). Graphical areas with no data because graph depth exceeds the deepest ocean depth at that latitude are indicated by grey and white sloped lines. Temperature: Temperature is highest in shallow tropical and southern temperate waters but is low around the northern margin of the plot, corresponding to the seas of Okhotsk and Japan. Temperature drops off far more rapidly with increasing depth than in the global ocean as a whole. Dissolved Oxygen: Dissolved Oxygen levels are lowest in shallow and bathyal tropical waters but are relatively high in shallow temperate waters

reflecting the higher oxygen saturation levels of colder waters and are high at depth at mid northern latitudes. Salinity: Salinity is fairly uniform throughout all depths in the IWP but is lowest at mid northern latitudes (Sea of Japan) and highest around the northern margins of the tropics, partially reflecting the extremely levels of the Red Sea and Persian Gulf. Particulate Organic Carbon Flux (POC flux): POC flux is highest throughout shallow waters but peaks in two plumes in temperate waters north and south of the equator, both around 40° N/S. Nitrate: Nitrate levels are low in shallow waters, but much higher in bathyal and abyssal waters, particularly those north of the equator, except at mid latitudes where they are notably reduced. Vertical dashed lines indicate the equator and approximate extent of the tropics.

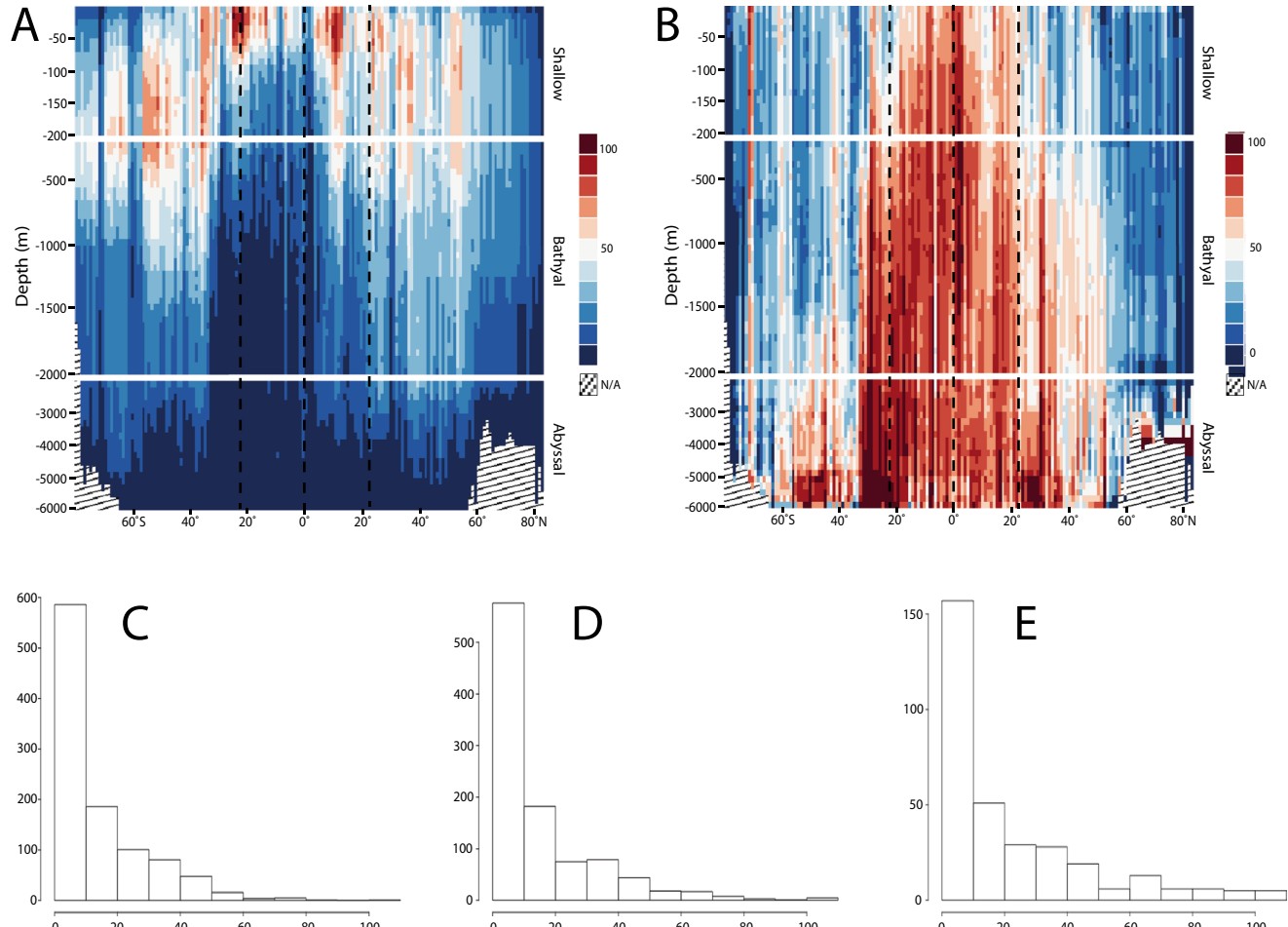

**Extended Data Fig. 5 | Global bathy-diversity plots illustrating uncertainty in interpolation effort. a)** Global bathymetric plot of raw species diversity interpolated only across the bathymetric range of each species included. Diversity given for depth intervals spanning the intertidal to lower abyssal zone (0–6,000 m) for latitudes between 78° S and 82° N and the full extent of longitude. Species richness is binned at intervals of 10 m (shallow), 50 m (bathyal) and 200 m (abyssal). **b)** Global bathymetric plot of the percentage difference between the raw and interpolated species diversity patterns (Fig. 3). Diversity given for depth intervals spanning the intertidal to lower abyssal zone (0–6,000 m) for latitudes between78° S and 82° N and the full extent of longitude. Species richness is binned at intervals of 10 m (shallow), 50 m (bathyal) and 200 m (abyssal). Regions of the dataset rely more heavily on interpolation in red, less so in blue. **c-e)** Histograms of the number of species for each depth section by the number of degrees of latitude for each individual range that have been interpolated (C: Shallow (0–200 m), D: Bathyal (200–2,000 m), E: Lower Bathyal and Abyssal (2,000–6,000 m). Vertical dashed lines indicate the equator and approximate extent of the tropics.

**Extended Data Table 1 | Spatial linear model (SLM) and Random Forest regression results for the interpolated species richness of three major bathomes**

| | Interpolated Data, SLM's | | | Interpolated Data, Random Forests | | |
|---|---|---|---|---|---|---|
| | Shallow | Bathyal | Lower Bathyal/ Abyssal | Shallow | Bathyal | Lower Bathyal/ Abyssal |
| **Global** | | | | | | |
| Max Species | 281 | 176 | 79 | 281 | 176 | 79 |
| Min Species | 24 | 10 | 1 | 24 | 10 | 1 |
| Seafloor Temperature | z=9.073*** | z=3.972*** | z=6.098*** | **57.30755** | 113.25034 | **120.54758** |
| Dissolved Oxygen | z=1.137 | z=-1.288 | z=-1.382 | 27.72139 | 49.60687 | 83.09766 |
| Salinity | z=0.164 | z=-0.357 | z=0.566 | 20.19912 | 104.18000 | 78.36120 |
| Silicate | z=1.644 | z=-0.642 | - | 24.41689 | 36.27889 | 53.83967 |
| Nitrate | z=1.150 | z=0.840 | z=0.096 | 29.41851 | 65.56463 | 39.64005 |
| POC Flux | z=1.416 | z=4.577*** | z=1.416 | 26.61111 | **133.80623** | 83.61285 |
| Multiple $R^2$ | 0.8870 | 0.6672 | 0.5354 | 150.6293 | 56.83681 | 22.77481 |
| Mean Sq Resid | | | | 95.98 | 93.31 | 90.25 |
| % Var explained | | | | 281 | 176 | 79 |
| **Southern Hemisphere** | | | | | | |
| Max Species | 257 | 142 | 61 | 257 | 142 | 61 |
| Min Species | 55 | 30 | 10 | 55 | 30 | 10 |
| Seafloor Temperature | z=6.234*** | z=1.422 | z=-0.064 | **38.05896** | 39.43038 | 76.83430 |
| Dissolved Oxygen | z=1.662 | z=1.403 | z=-0.840 | 34.21720 | 84.02310 | 73.77897 |
| Salinity | z=-0.395 | z=-1.458 | z=-0.397 | 18.80590 | 36.92266 | 67.99913 |
| Nitrate | z=1.343 | z=0.910 | z=-0.034 | 30.09758 | 22.41606 | 31.50330 |
| Silicate | - | - | - | 15.24728 | 29.86442 | 47.08117 |
| POC Flux | z=0.502 | z=7.087*** | z=2.460* | 22.64510 | **104.46844** | **111.19269** |
| Multiple $R^2$ | 0.8451 | 0.7989 | 0.4843 | | | |
| Mean Sq Resid | | | | 84.20096 | 22.87922 | 33.22013 |
| % Var explained | | | | 96.58 | 95.66 | 91.51 |
| **Northern Hemisphere** | | | | | | |
| Max Species | 281 | 176 | 79 | 281 | 176 | 79 |
| Min Species | 24 | 10 | 1 | 24 | 10 | 1 |
| Seafloor Temperature | z=7.350*** | z=6.357*** | z= 5.585*** | **51.99973** | 85.51660 | 78.28992 |
| Dissolved Oxygen | z=0.708 | z=-3.905*** | z= -7.768*** | 27.71591 | 45.18085 | 46.15036 |
| Salinity | z=0.087 | 0.010 | z=6.246*** | 14.19280 | 47.18447 | 44.86376 |
| Silicate | - | - | - | 18.71158 | 29.81318 | 86.96191 |
| Nitrate | z=1.873 | z=1.124 | z=-5.659*** | 18.36836 | 42.42297 | 56.63816 |
| POC Flux | z=1.436 | z=5.351*** | z=0.733 | 18.64478 | **104.28641** | **102.17163** |
| Multiple $R^2$ | 0.9339 | 0.8058 | 0.6443 | | | |
| Mean Sq Resid | | | | 120.7536 | 56.44982 | 7.571894 |
| % Var explained | | | | 97.48 | 94.67 | 89.01 |
| **Atlantic** | | | | | | |
| Max Species | 78 | 78 | 54 | 78 | 78 | 54 |
| Min Species | 24 | 16 | 3 | 24 | 16 | 3 |
| Seafloor Temperature | z=0.613 | z=4.725*** | z= 15.157*** | **58.19724** | 78.05699 | 66.23481 |
| Dissolved Oxygen | z=-0.241 | z=-0.795 | z= 6.315*** | 25.91574 | 60.71769 | 67.51979 |
| Salinity | - | z=3.109** | z=-9.227*** | 53.93365 | **122.50105** | 74.43238 |
| Nitrate | z=-1.437 | z=-6.220*** | z=6.481*** | 46.01554 | 100.85837 | 70.67706 |
| POC Flux | z=-1.314 | z=0.143 | z=4.701*** | 40.57453 | 69.31335 | **112.70479** |
| Multiple $R^2$ | 0.3783 | 0.5109 | 0.6149 | | | |
| Mean Sq Resid | | | | 61.65609 | 16.72757 | 7.456373 |
| % Var explained | | | | 71.28 | 92.28 | 91.61 |
| **Indo-West Pacific** | | | | | | |
| Max Species | 177 | 95 | 27 | 177 | 95 | 27 |
| Min Species | 23 | 9 | 2 | 23 | 9 | 2 |
| Seafloor Temperature | z=5.100*** | z=17.728*** | z=12.470*** | **45.58353** | **108.79467** | **156.82556** |
| Dissolved Oxygen | z=2.573* | z=3.632*** | z=0.459 | 27.90714 | 83.98331 | 78.60132 |
| Salinity | - | z=- | z=-1.951 | 42.75714 | 86.72118 | 99.62713 |
| Nitrate | z=2.090* | z=3.198** | z=-2.489* | 20.29570 | 45.73316 | 65.31943 |
| POC Flux | z=-0.580 | z=0.020 | z=4.598*** | 38.02151 | 66.87571 | 124.47171 |
| Multiple $R^2$ | 0.7954 | 0.6002 | 0.5506 | | | |
| Mean Sq Resid | | | | 93.30233 | 26.10883 | 1.556258 |
| % Var explained | | | | 95.72 | 91.91 | 87.45 |

Spatial linear model (SLM) and Random Forest regression results for the species richness of three major depth zones: Shallow (0–200m), Bathyal (200–2,000m), Lower Bathyal and Abyssal (2000–6,000m) across the global dataset, northern and southern hemispheres and the Atlantic and Indo-West Pacific oceanic realms using interpolated species richness values. Maximum and minimum richness are the highest and lowest values for each depth section respectively. For SLM's, model results are for the best performing SLM as determined by Akaike information criteria. Statistical results are given as two-tailed Z values: *$P < 0.05$, **$P < 0.01$, ***$P < 0.001$. Multiple $R^2$ values give the proportion of variance in species richness that can be explained by the tested environmental factors. For Random Forest regression result, model outputs are given as the relative importance of each tested variable (% Increase in Mean Squared Error - %IncMSE) which represents the percentage decrease in model performance for iterations where the variable is not tested. Summary statistics for full model performance are given as the mean of squared residuals and percentage of variance that can be explained by the tested environmental factors. The most important variables for each geographic and bathymetric subdivision after each random forest was run across 1,000 iterations are highlighted in bold.

**Extended Data Table 2 | Spatial linear model (SLM) and Random Forest regression results for the raw and interpolated, sampling effort corrected species richness of three major bathomes across the global, northern and southern hemisphere datasets**

| | Interpolated Data, Sampling Correction SLM's | | | Interpolated Data, Sampling Correction Random Forest | | | Raw Data, Sampling Correction Random Forest | | | Raw Data, Sampling Correction Random Forest | | |
|---|---|---|---|---|---|---|---|---|---|---|---|---|
| | Shallow | Bathyal | Lower Bathyal/Abyssal | Shallow | Bathyal | Lower Bathyal/Abyssal | Shallow | Bathyal | Lower Bathyal/Abyssal | Shallow | Bathyal | Lower Bathyal/Abyssal |
| **Global** | | | | | | | | | | | | |
| Max Species | 281 | 176 | 79 | 281 | 176 | 79 | 133 | 90 | 40 | 133 | 90 | 40 |
| Min Species | 24 | 10 | 1 | 24 | 10 | 1 | 11 | 1 | 1 | 11 | 1 | 1 |
| Seafloor Temperature | z=8.983*** | z=3.935*** | z=6.046*** | **54.30647** | 108.95675 | **108.6458** | z=3.063** | z=2.519* | z=3.271** | **47.44072** | 67.31853 | 87.10524 |
| Dissolved Oxygen | z=1.124 | z=-1.293 | z=-1.399 | 29.26082 | 49.34926 | 89.69277 | z=2.098* | z=1.868 | z=2.395 | 37.22579 | 87.73667 | 65.15769 |
| Salinity | z=0.148 | z=-0.292 | z=0.644 | 17.86046 | 95.76012 | 72.15951 | z=1.988* | z=-0.674 | z=-2.373* | 21.03018 | 94.65119 | 79.74837 |
| Silicate | z=1.618 | z=-0.579 | - | 25.40769 | 33.75551 | 53.68384 | z=-0.906 | z=0.590 | - | 28.83699 | 37.50255 | 62.27224 |
| Nitrate | z=1.149 | z=0.734 | z=0.078 | 29.19042 | 62.75989 | 38.75920 | z=1.732 | z=1.135 | z=2.382* | 41.24431 | 53.99263 | 59.80344 |
| POC Flux | z=1.334 | z=4.498*** | z=1.335 | 25.17782 | **140.22129** | 78.81505 | z=2.010* | z=4.847*** | z=3.509*** | 27.67891 | **161.14978** | **138.43425** |
| Multiple R² | 0.8866 | 0.6721 | 0.5423 | | | | 0.2841 | 0.2999 | 0.2628 | | | |
| Mean Sq Resid | | | | 150.8357 | 59.11824 | 26.33178 | | | | 267.8167 | 72.34434 | 16.64584 |
| % Var explained | | | | 95.97 | 93.04 | 88.72 | | | | 52.19 | 72.96 | 65.54 |
| **Southern Hemisphere** | | | | | | | | | | | | |
| Max Species | 257 | 142 | 61 | 257 | 142 | 61 | 133 | 90 | 29 | 133 | 90 | 29 |
| Min Species | 55 | 30 | 12 | 55 | 30 | 12 | 11 | 1 | 1 | 11 | 1 | 1 |
| Seafloor Temperature | z=6.171*** | z=2.322* | z=-0.048 | **39.84385** | 44.53019 | 74.11451 | z=0.227 | z=1.181 | z=-2.333* | 14.11292 | 35.65438 | 50.47599 |
| Dissolved Oxygen | z=1.625 | z=0.943 | z=-0.864 | 34.94283 | 70.46255 | 46.63607 | z=3.350*** | z=5.813*** | z=1.043 | 20.74720 | **107.86269** | 42.42964 |
| Salinity | z=-0.440 | z=-2.860** | z=-0.401 | 17.42765 | 33.13498 | 42.00006 | z=2.341* | z=2.012* | z=-0.989 | 21.62584 | 39.54080 | 45.53312 |
| Nitrate | z=1.324 | z=0.334 | z=-0.043 | 27.81813 | 22.71151 | 50.71918 | z=-1.291 | z=1.815 | z=-0.183 | 33.27008 | 36.50186 | 55.72706 |
| Silicate | - | - | - | 16.66546 | 31.76342 | 84.77161 | - | - | - | 12.65540 | 35.47928 | **92.62619** |
| POC Flux | z=0.457 | z=8.301*** | z=2.459* | 22.45492 | **111.04902** | **101.0382** | z=-0.417 | z=5.083*** | z=4.224*** | **35.41285** | 84.12676 | 76.07532 |
| Multiple R² | 0.8461 | 0.7966 | 0.4911 | | | | 0.3780 | 0.6601 | 0.3846 | | | |
| Mean Sq Resid | | | | 82.7488 | 23.73135 | 8.450308 | | | | 223.831 | 69.45109 | 10.72822 |
| % Var explained | | | | 96.64 | 95.49 | 87.73 | | | | 70.37 | 79.01 | 68.75 |
| **Northern Hemisphere** | | | | | | | | | | | | |
| Max Species | 281 | 176 | 79 | 281 | 176 | 79 | 127 | 77 | 40 | 127 | 77 | 40 |
| Min Species | 24 | 10 | 1 | 24 | 10 | 1 | 14 | 3 | 1 | 14 | 3 | 1 |
| Seafloor Temperature | z=7.223*** | z=6.500*** | z=5.477*** | **50.56231** | 79.85440 | 79.93092 | z=2.330* | z=2.546* | z=6.828*** | **39.78924** | 57.49800 | 81.06120 |
| Dissolved Oxygen | z=0.719 | z=-3.884*** | z=-7.564*** | 26.90077 | 33.85779 | **91.26112** | z=0.591 | z=-0.110 | z=-4.820*** | 28.56007 | 40.51154 | 50.45429 |
| Salinity | z=0.068 | z=0.008 | z=6.337*** | 14.40164 | 48.04474 | 62.50212 | z=1.438 | z=-0.573 | z=3.860*** | 21.23006 | 55.35593 | 51.44487 |
| Silicate | - | - | - | 16.22445 | 32.90558 | 64.02971 | - | - | - | 12.54331 | 47.85905 | 39.42599 |
| Nitrate | z=1.852 | z=1.195 | z=-5.432*** | 18.97245 | 38.29043 | 33.96154 | z=1.719 | z=0.837 | z=3.428*** | 26.07093 | 40.32222 | 35.29192 |
| POC Flux | z=1.290 | z=5.099*** | z=-0.669 | 16.65322 | **111.48756** | 81.00610 | z=2.458* | z=7.158*** | z=5.365*** | 27.86188 | **159.86086** | **148.52290** |
| Multiple R² | 0.9323 | 0.809 | 0.6498 | | | | 0.4623 | 0.4608 | 0.5531 | | | |
| Mean Sq Resid | | | | 119.3856 | 61.91751 | 36.16981 | | | | 189.7072 | 68.78223 | 19.54047 |
| % Var explained | | | | 97.74 | 94.12 | 90.67 | | | | 65.12 | 65.09 | 65.68 |

Spatial linear model (SLM) and Random Forest regression results for the species richness of three major depth zones: Shallow (0–200m), Bathyal (200–2,000m), Lower Bathyal and Abyssal (2000–6,000m) across the global dataset and northern and southern hemispheres for interpolated species richness values with a sampling offset and raw species richness values with a sampling offset. Maximum and minimum richness are the highest and lowest values for each depth section respectively. For SLMs, model results are for the best performing SLMs as determined by Akaike information criteria. Statistical results are given as two-tailed Z values: *$P < 0.05$, **$P < 0.01$, ***$P < 0.001$. Multiple $R^2$ values give the proportion of variance in species richness that can be explained by the tested environmental factors. For Random Forest regression result, model outputs are given as the relative importance of each tested variable (% Increase in Mean Squared Error - %IncMSE) which represents the percentage decrease in model performance for iterations where the variable is not tested. Summary statistics for full model performance are given as the mean of squared residuals and percentage of variance that can be explained by the tested environmental factors. The most important variables for each geographic and bathymetric subdivision after each random forest was run across 1,000 iterations are highlighted in bold.

**Extended Data Table 3 | Spatial linear model (SLM) and Random Forest regression results for the raw and interpolated, sampling effort corrected species richness of three major bathomes across the global, Atlantic and Indo-west Pacific datasets**

| | Interpolated Data, Sampling Correction SLM's | | | Interpolated Data, Sampling Correction Random Forest | | | Raw Data, Sampling Correction Random Forest | | | Raw Data, Sampling Correction Random Forest | | |
|---|---|---|---|---|---|---|---|---|---|---|---|---|
| | Shallow | Bathyal | Lower Bathyal/Abyssal | Shallow | Bathyal | Lower Bathyal/Abyssal | Shallow | Bathyal | Lower Bathyal/Abyssal | Shallow | Bathyal | Lower Bathyal/Abyssal |
| **Global** | | | | | | | | | | | | |
| Max Species | 281 | 176 | 79 | 281 | 176 | 79 | 133 | 90 | 40 | 133 | 90 | 40 |
| Min Species | 24 | 10 | 1 | 24 | 10 | 1 | 11 | 1 | 1 | 11 | 1 | 1 |
| Seafloor Temperature | z=8.983*** | z=3.935*** | z=6.046*** | 54.30647 | 108.95675 | **108.6458** | z=3.063** | z=2.519* | z=3.271** | 47.44072 | 67.31853 | 87.10524 |
| Dissolved Oxygen | z=1.124 | z=-1.293 | z=-1.399 | 29.26082 | 49.34926 | 89.69277 | z=2.098* | z=1.868 | z=2.395 | 37.22579 | 87.73667 | 65.15769 |
| Salinity | z=0.148 | z=-0.292 | z=0.644 | 17.86046 | 95.76012 | 72.15951 | z=1.988* | z=-0.674 | z=-2.373* | 21.03018 | 94.65119 | 79.74837 |
| Silicate | z=1.618 | z=-0.579 | - | 25.40769 | 33.75551 | 53.68384 | z=-0.906 | z=0.590 | - | 28.83699 | 37.50255 | 62.27224 |
| Nitrate | z=1.149 | z=0.734 | z=0.078 | 29.19042 | 62.75989 | 38.75920 | z=1.732 | z=1.135 | z=2.382* | 41.24431 | 53.99263 | 59.80344 |
| POC Flux | z=1.334 | z=4.498*** | z=1.335 | 25.17782 | **140.22129** | 78.81505 | z=2.010* | z=4.847*** | z=3.509*** | 27.67891 | **161.14978** | **138.43425** |
| Multiple R² | 0.8866 | 0.6721 | 0.5423 | | | | 0.2841 | 0.2999 | 0.2628 | | | |
| Mean Sq Resid | | | | 150.8357 | 59.11824 | 26.33178 | | | | 267.8167 | 72.34434 | 16.64584 |
| % Var explained | | | | 95.97 | 93.04 | 88.72 | | | | 52.19 | 72.96 | 65.54 |
| **Atlantic** | | | | | | | | | | | | |
| Max Species | 78 | 78 | 54 | 78 | 78 | 54 | 59 | 57 | 35 | 59 | 57 | 35 |
| Min Species | 27 | 16 | 3 | 27 | 16 | 3 | 1 | 1 | 1 | 1 | 1 | 1 |
| Seafloor Temperature | z=0.695 | z=2.588** | z=7.138*** | 53.64683 | 79.07021 | 81.10140 | z=-1.906 | z=0.301 | z=2.935** | 31.27605 | 62.90625 | 61.84061 |
| Dissolved Oxygen | z=-0.354 | z=-0.315 | z=3.166** | 21.52550 | 36.38349 | 72.47783 | z=-1.422 | z=-0.567 | z=1.975* | 27.79711 | 44.43608 | 56.53338 |
| Salinity | z=-0.848 | z=-2.571* | z=-4.585*** | 48.30194 | **96.48346** | 60.63813 | z=-0.496 | z=-0.811 | z=-1.211 | 32.24261 | 85.25554 | 66.04785 |
| Nitrate | z=-1.179 | z=-1.613 | z=3.152** | 31.38444 | 92.31092 | 75.33500 | z=-2.867** | z=-3.133*** | z=0.597 | 43.74317 | **102.40558** | 73.76073 |
| POC Flux | z=2.728** | z=-0.155 | z=2.158* | 29.17578 | 59.12827 | **125.3075** | z=2.148* | z=2.501* | z=3.172** | 26.70344 | 59.18157 | **114.89418** |
| Multiple R² | 0.4012 | 0.5226 | 0.617 | | | | 0.3267 | 0.4681 | 0.477 | | | |
| Mean Sq Resid | | | | 69.31439 | 21.9895 | 8.648602 | | | | 113.3803 | 35.03947 | 12.07454 |
| % Var explained | | | | 67.71 | 89.85 | 90.25 | | | | 47.2 | 75.67 | 71.88 |
| **Indo-West Pacific** | | | | | | | | | | | | |
| Max Species | 177 | 95 | 27 | 177 | 95 | 27 | 115 | 69 | 22 | 115 | 69 | 22 |
| Min Species | 28 | 9 | 2 | 28 | 9 | 2 | 1 | 1 | 1 | 1 | 1 | 1 |
| Seafloor Temperature | z=4.891*** | z=5.780*** | z=7.054*** | 49.60165 | 96.91025 | 147.7645 | z=3.410*** | z=3.911*** | z=4.118*** | 35.67738 | 43.28008 | 58.58469 |
| Dissolved Oxygen | z=2.181* | z=0.908 | z=0.341 | 32.28798 | 67.76277 | 73.82147 | z=4.022*** | z=3.884*** | z=3.073** | 36.30096 | **96.97106** | 42.35445 |
| Salinity | z=-0.530 | z=-7.732*** | z=-1.104 | 47.86372 | 49.45267 | 94.26833 | z=3.717*** | z=-0.646 | z=1.732 | 46.50942 | 41.79564 | 42.58813 |
| Nitrate | z=1.915 | z=1.202 | z=-1.446 | 20.26849 | 43.67829 | 56.95848 | z=1.359 | z=3.224** | z=0.239 | 36.07971 | 39.04137 | 37.18748 |
| POC Flux | z=-0.458 | z=0.067 | z=1.536 | 39.41511 | 46.04500 | 101.73817 | z=-1.392 | z=0.779 | z=4.161*** | **53.03815** | 58.93143 | **82.01852** |
| Multiple R² | 0.7975 | 0.7106 | 0.5657 | | | | 0.5386 | 0.425 | 0.4482 | | | |
| Mean Sq Resid | | | | 93.44809 | 30.6048 | 1.736431 | | | | 167.2532 | 37.95761 | 4.350153 |
| % Var explained | | | | 95.64 | 90.52 | 86.00 | | | | 74.38 | 78.56 | 62.59 |

Spatial linear model (SLM) and Random Forest regression results for the species richness of three major depth zones: Shallow (0–200m), Bathyal (200–2,000m), Lower Bathyal and Abyssal (2000–6,000m) across the global dataset and Atlantic and Indo-west Pacific oceanic realms for interpolated species richness values with a sampling offset and raw species richness values with a sampling offset. Maximum and minimum richness are the highest and lowest values for each depth section respectively. For SLMs, model results are for the best performing SLMs as determined by Akaike information criteria. Statistical results are given as two-tailed Z values: *$P < 0.05$, **$P < 0.01$, ***$P < 0.001$. Multiple $R^2$ values give the proportion of variance in species richness that can be explained by the tested environmental factors. For Random Forest regression result, model outputs are given as the relative importance of each tested variable (% Increase in Mean Squared Error - %IncMSE) which represents the percentage decrease in model performance for iterations where the variable is not tested. Summary statistics for full model performance are given as the mean of squared residuals and percentage of variance that can be explained by the tested environmental factors. The most important variables for each geographic and bathymetric subdivision after each random forest was run across 1,000 iterations are highlighted in bold.

# Reporting Summary

Please do not complete any field with "not applicable" or n/a.  Refer to the help text for what text to use if an item is not relevant to your study.
For final submission: please carefully check your responses for accuracy; you will not be able to make changes later.

## Statistics

For all statistical analyses, confirm that the following items are present in the figure legend, table legend, main text, or Methods section.

| n/a | Confirmed | |
|---|---|---|
| ☐ | ☑ | The exact sample size ($n$) for each experimental group/condition, given as a discrete number and unit of measurement |
| ☐ | ☑ | A statement on whether measurements were taken from distinct samples or whether the same sample was measured repeatedly |
| ☐ | ☑ | The statistical test(s) used AND whether they are one- or two-sided *Only common tests should be described solely by name; describe more complex techniques in the Methods section.* |
| ☐ | ☑ | A description of all covariates tested |
| ☐ | ☑ | A description of any assumptions or corrections, such as tests of normality and adjustment for multiple comparisons |
| ☐ | ☑ | A full description of the statistical parameters including central tendency (e.g. means) or other basic estimates (e.g. regression coefficient) AND variation (e.g. standard deviation) or associated estimates of uncertainty (e.g. confidence intervals) |
| ☐ | ☑ | For null hypothesis testing, the test statistic (e.g. $F$, $t$, $r$) with confidence intervals, effect sizes, degrees of freedom and $P$ value noted *Give P values as exact values whenever suitable.* |
| ☑ | ☐ | For Bayesian analysis, information on the choice of priors and Markov chain Monte Carlo settings |
| ☑ | ☐ | For hierarchical and complex designs, identification of the appropriate level for tests and full reporting of outcomes |
| ☑ | ☐ | Estimates of effect sizes (e.g. Cohen's $d$, Pearson's $r$), indicating how they were calculated |

*Our web collection on statistics for biologists contains articles on many of the points above.*

## Software and code

Policy information about availability of computer code

| Data collection | **No Code was used for Data Collection** |
|---|---|
| Data analysis | **All code used for analysis is detailed in the methodology. A full statement is attached to the manuscript.** |

For manuscripts utilizing custom algorithms or software that are central to the research but not yet described in published literature, software must be made available to editors and reviewers. We strongly encourage code deposition in a community repository (e.g. GitHub). See the Nature Portfolio guidelines for submitting code & software for further information.

R packages used: marmap 1.0.10; meowR 0.6.2; slm 1.2.0; randomforest 4.7-1.1, plotmo 3.6.2
Software used: R 4.2.3; QGIS 3.0.0 Girona

## Data

Policy information about availability of data

All manuscripts must include a data availability statement. This statement should provide the following information, where applicable:
- Accession codes, unique identifiers, or web links for publicly available datasets
- A description of any restrictions on data availability
- For clinical datasets or third party data, please ensure that the statement adheres to our policy

**A Full Data Availability Statement has been appended to the manuscript**

The full OBIS and GBIF datasets used to generate coordinate records for this study are available online at OBIS: https://obis.org/taxon/123080; GBIF: https://www.gbif.org/species/214. The fully curated species occurrence dataset and Catalogue of descriptive Asteroidea ranges are all derived from publicly available sources. The species occurrence dataset will be hosted at: https://data.nhm.ac.uk/dataset/. The Catalogue of descriptive ranges is available under Supplementary Data 1. Environmental Data excluding POC flux are available from the CARS 2009 dataset (https://www.marine.csiro.au/~dunn/cars2009), POC flux data are derived from data in (http://..sites.science.oregonstate.edu/ocean/productivity/index.php). Bathymetric data were sourced from NOAA

## Research involving human participants, their data, or biological material

Policy information about studies with human participants or human data. See also policy information about sex, gender (identity/presentation), and sexual orientation and race, ethnicity and racism.

| | |
|---|---|
| Reporting on sex and gender | N/A |
| Reporting on race, ethnicity, or other socially relevant groupings | N/A |
| Population characteristics | N/A |
| Recruitment | N/A |
| Ethics oversight | N/A |

Note that full information on the approval of the study protocol must also be provided in the manuscript.

# Field-specific reporting

Please select the one below that is the best fit for your research. If you are not sure, read the appropriate sections before making your selection.

☐ Life sciences ☐ Behavioural & social sciences ☑ Ecological, evolutionary & environmental sciences

For a reference copy of the document with all sections, see [nature.com/documents/nr-reporting-summary-flat.pdf](http://nature.com/documents/nr-reporting-summary-flat.pdf)

# Life sciences study design

All studies must disclose on these points even when the disclosure is negative.

| | |
|---|---|
| Sample size | |
| Data exclusions | |
| Replication | |
| Randomization | |
| Blinding | |

# Behavioural & social sciences study design

All studies must disclose on these points even when the disclosure is negative.

| | |
|---|---|
| Study description | |
| Research sample | |
| Sampling strategy | |
| Data collection | |
| Timing | |
| Data exclusions | |
| Non-participation | |
| Randomization | |

# Ecological, evolutionary & environmental sciences study design

All studies must disclose on these points even when the disclosure is negative.

| | |
|---|---|
| Study description | A global and bathymetric analysis of distribution patterns in the class Asteroidea |
| Research sample | 256,861 species level coordinate records for 1,751 species of Asteroid |
| Sampling strategy | All data points counted as single occurrence records, summed by degree of latitude within the study region |
| Data collection | Data was compiled from the online repositories OBIS and GBIF, the published literature and museum specimens |
| Timing and spatial scale | c.1760-2022 |
| Data exclusions | Point occurrence records outside of the known range for each species, Mediterranean Data from global analyses |
| Reproducibility | Full details to enable reproduction of study in the methods |
| Randomization | Data randomly drawn to train Random Forest models and for stationary linear models |
| Blinding | N/A |

Did the study involve field work?　☐ Yes　☑ No

## Field work, collection and transport

| | |
|---|---|
| Field conditions | |
| Location | |
| Access & import/export | |
| Disturbance | |

# Reporting for specific materials, systems and methods

We require information from authors about some types of materials, experimental systems and methods used in many studies. Here, indicate whether each material, system or method listed is relevant to your study. If you are not sure if a list item applies to your research, read the appropriate section before selecting a response.

### Materials & experimental systems

| n/a | Involved in the study |
|---|---|
| ☑ | ☐ Antibodies |
| ☑ | ☐ Eukaryotic cell lines |
| ☑ | ☐ Palaeontology and archaeology |
| ☑ | ☐ Animals and other organisms |
| ☑ | ☐ Clinical data |
| ☑ | ☐ Dual use research of concern |
| ☑ | ☐ Plants |

### Methods

| n/a | Involved in the study |
|---|---|
| ☑ | ☐ ChIP-seq |
| ☑ | ☐ Flow cytometry |
| ☑ | ☐ MRI-based neuroimaging |

## Antibodies

| | |
|---|---|
| Antibodies used | |
| Validation | |

# Eukaryotic cell lines

Policy information about cell lines and Sex and Gender in Research

| | |
|---|---|
| Cell line source(s) | |
| Authentication | |
| Mycoplasma contamination | |
| Commonly misidentified lines (See ICLAC register) | |

# Palaeontology and Archaeology

| | |
|---|---|
| Specimen provenance | |
| Specimen deposition | |
| Dating methods | |

☐ Tick this box to confirm that the raw and calibrated dates are available in the paper or in Supplementary Information.

| | |
|---|---|
| Ethics oversight | |

Note that full information on the approval of the study protocol must also be provided in the manuscript.

# Animals and other research organisms

Policy information about studies involving animals; ARRIVE guidelines recommended for reporting animal research, and Sex and Gender in Research

| | |
|---|---|
| Laboratory animals | |
| Wild animals | |
| Reporting on sex | |
| Field-collected samples | |
| Ethics oversight | |

Note that full information on the approval of the study protocol must also be provided in the manuscript.

# Clinical data

Policy information about clinical studies

All manuscripts should comply with the ICMJE guidelines for publication of clinical research and a completed CONSORT checklist must be included with all submissions.

| | |
|---|---|
| Clinical trial registration | |
| Study protocol | |
| Data collection | |
| Outcomes | |

# Dual use research of concern

Policy information about dual use research of concern

## Hazards

Could the accidental, deliberate or reckless misuse of agents or technologies generated in the work, or the application of information presented in the manuscript, pose a threat to:

| No | Yes | |
|----|-----|--|
| ☐ | ☐ | Public health |
| ☐ | ☐ | National security |
| ☐ | ☐ | Crops and/or livestock |
| ☐ | ☐ | Ecosystems |
| ☐ | ☐ | Any other significant area |

## Experiments of concern

Does the work involve any of these experiments of concern:

| No | Yes | |
|----|-----|--|
| ☐ | ☐ | Demonstrate how to render a vaccine ineffective |
| ☐ | ☐ | Confer resistance to therapeutically useful antibiotics or antiviral agents |
| ☐ | ☐ | Enhance the virulence of a pathogen or render a nonpathogen virulent |
| ☐ | ☐ | Increase transmissibility of a pathogen |
| ☐ | ☐ | Alter the host range of a pathogen |
| ☐ | ☐ | Enable evasion of diagnostic/detection modalities |
| ☐ | ☐ | Enable the weaponization of a biological agent or toxin |
| ☐ | ☐ | Any other potentially harmful combination of experiments and agents |

# Plants

| Seed stocks | |
|-------------|--|
| Novel plant genotypes | |
| Authentication | |

# ChIP-seq

## Data deposition

☐ Confirm that both raw and final processed data have been deposited in a public database such as GEO.

☐ Confirm that you have deposited or provided access to graph files (e.g. BED files) for the called peaks.

| Data access links
May remain private before publication. | |
|---|---|
| Files in database submission | |
| Genome browser session
(e.g. UCSC) | |

## Methodology

| Replicates | |
|------------|--|
| Sequencing depth | |
| Antibodies | |
| Peak calling parameters | |
| Data quality | |
| Software | |

# Flow Cytometry

## Plots

Confirm that:

- ☐ The axis labels state the marker and fluorochrome used (e.g. CD4-FITC).
- ☐ The axis scales are clearly visible. Include numbers along axes only for bottom left plot of group (a 'group' is an analysis of identical markers).
- ☐ All plots are contour plots with outliers or pseudocolor plots.
- ☐ A numerical value for number of cells or percentage (with statistics) is provided.

## Methodology

| | |
|---|---|
| Sample preparation | |
| Instrument | |
| Software | |
| Cell population abundance | |
| Gating strategy | |

☐ Tick this box to confirm that a figure exemplifying the gating strategy is provided in the Supplementary Information.

# Magnetic resonance imaging

## Experimental design

| | |
|---|---|
| Design type | |
| Design specifications | |
| Behavioral performance measures | |

| | |
|---|---|
| Imaging type(s) | |
| Field strength | |
| Sequence & imaging parameters | |
| Area of acquisition | |

Diffusion MRI        ☐ Used        ☐ Not used

## Preprocessing

| | |
|---|---|
| Preprocessing software | |
| Normalization | |
| Normalization template | |
| Noise and artifact removal | |
| Volume censoring | |

## Statistical modeling & inference

| | |
|---|---|
| Model type and settings | |
| Effect(s) tested | |

Specify type of analysis:        ☐ Whole brain        ☐ ROI-based        ☐ Both

Statistic type for inference

(See Eklund et al. 2016)

Correction

## Models & analysis

| n/a | Involved in the study |
|---|---|
| ☐ ☐ | Functional and/or effective connectivity |
| ☐ ☐ | Graph analysis |
| ☐ ☐ | Multivariate modeling or predictive analysis |

Functional and/or effective connectivity

Graph analysis

Multivariate modeling and predictive analysis

