## [Peer Review File · Nature Ecology & Evolution]

Deep seafloor diversity in Asteroidea is shaped by competing processes across different latitudes and oceans

Corresponding Author: Mr Hugh Carter

Version 0:

Decision Letter:

24th August 2022

Dear Mr Carter,

Your Article, "Competing processes shape variable deep seafloor diversity between hemispheres" has now been seen by two reviewers. I apologize for the long duration of the peer review process; we were waiting for a report from a third reviewer, but they never delivered. Based on the reports we have received, and after careful consideration, we have decided to invite a major revision of the manuscript.

As you will see from the reports copied below, the reviewers raise several important concerns. We find that these concerns limit the strength of the study, and therefore we ask you to address them with additional work. Without substantial revisions, we will be unlikely to send the paper back to review. In particular, both reviewers suggest reconsidering the included datasets (i.e. adding GBIF but dropping iNaturalist), accounting for spatial sampling biases, conducting sensitivity analyses, etc.

If you feel that you are able to comprehensively address the reviewers' concerns, please provide a point-by-point response to these comments along with your revision. Please show all changes in the manuscript text file with track changes or colour highlighting. If you are unable to address specific reviewer requests or find any points invalid, please explain why in the point-by-point response.

If you have not done so already we suggest that you begin to revise your manuscript so that it conforms to our Article format instructions at <http://www.nature.com/natecolevol/info/final-submission>. Refer also to any guidelines provided in this letter.

Include a revised version of any required reporting checklist. It will be available to reviewers (and, potentially, statisticians) to aid in their evaluation if the manuscript goes back for peer review. A revised checklist is essential for re-review of the paper.

Link Redacted

If you wish to submit a suitably revised manuscript we would hope to receive it within 6 months. If you cannot send it within this time, please let us know. We will be happy to consider your revision so long as nothing similar has been accepted for publication at Nature Ecology & Evolution or published elsewhere.

Nature Ecology & Evolution is committed to improving transparency in authorship. As part of our efforts in this direction, we are now requesting that all authors identified as 'corresponding author' on published papers create and link their Open Researcher and Contributor Identifier (ORCID) with their account on the Manuscript Tracking System (MTS), prior to acceptance. This applies to primary research papers only. ORCID helps the scientific community achieve unambiguous attribution of all scholarly contributions. You can create and link your ORCID from the home page of the MTS by clicking on 'Modify my Springer Nature account'. For more information please visit please visit <a

href="http://www.springernature.com/orcid">www.springernature.com/orcid.

Thank you for the opportunity to review your work.

[redacted]

Reviewer expertise:

Reviewer #1: Marine biogeography

Reviewer #2: Deep sea ecology, OBIS data

Reviewers' comments:

Reviewer #1 (Remarks to the Author):

Major comments:

Overall I think this is a great study trying to hypothesise drivers of patterns and processes that shape global patterns of marine biodiversity for one highly novel dataset. I think Nature Ecology and Evolution should consider publishing this paper.

This is an interesting study done on a highly novel global dataset of Asteroids. There clearly has been a huge amount of work put into the gathering of data, developing the manuscript and undertaking analyses. The figures are comprehensive and well done.

My only major concern is the role of sampling effort and the effect that creating range/interpolated maps from biases/patchy data might have on results. I think these concerns could be easily addressed with a few more figures and some more analyses. I realise it is very difficult to disentangle observational processes from biodiversity ones. But I would appreciate a little more clarity on some of the datasets, how bias/sampling effort is accounted for in the study and also some reporting of the trends of uncertainty for interpolated/predicted species richness.

I like the addition of figure one to describe sampling effort, but the mapped figure you cannot see the density of samples per a cell or bioregion. I think it would be reasonable to present this rather than just the spatial locations of Asteroids for shallow, bathyal and abyssal regions. Like Fig. 1B, it would be great to see the sampling effort per cell across latitude/longitude. In fact, it is surprising this has been omitted considering how diligent the authors are on providing latitude/depth maps of sampling effort. The map of spatial occurrences (Fig. 1A) could be moved to the SI. I also think effort needs to be reported on the log scale. This is how it would be used in a model (linear predictor scale) and it would be interesting to see if the sampling effort matches the diversity peak in Fig. 3a. Currently hard to see on the reported count scale because effort is so high in the shallow regions (which is to be expected).

I am wary of the interpolated ranges approach. I feel it smears much of the nuance when it comes to describing species distributions. I understand that it is commonly used, but if you are then using interpolated ranges to make inference on environmental drivers it seems to be that this will introduce biases into the inferences, as you are potentially inflating richness where it is low, or unintentionally creating correlations/confounding effects in the interpolated data that did not exist in the raw data. I think this could be easily fixed by using the raw richness and correcting for effort an offset in a statistical model. Sometime like this using a GLM in R (or your software of choice):

```
species_richness_per_cell ~ 1 + covariates + offset(log(samples_per_cell))
```

This will effectively correct for richness in areas that are sampled more intensely and remove the need to build metrics/inference on interpolated ranges. Of course this assumes a poisson (log link function) to account for effort in cell/site richness.

Uncertainty is missing from this analysis. I think it would be prudent to report uncertainty in maps and predictions. As the data is not even sampled everywhere I would expect there to be regions which are highly uncertain for the richness estimates where sampling is low or outside of the environmental values observed at survey (raw data) sites. These uncertainties could then be used to identify priorities for future surveys or looking for alternative taxa which could fill these voids.

Minor comments:

Line 50: "or risk bias towards common or cosmopolitan species by relying on models" is really an inaccurate comment. This work also included analyses using species richness models that included > 2000 unique species. In addition, the multiple-

species Bayesian model was developed to account for sampling bias in the data, but comes at the cost of not being able to include the very rare species in the analysis. This is true for any SDM (species interpolation) that is based on a statistical model.

Line 61: abyssal: 2,000-6,000m is not really abyss, more like lower bathyal (2000-3500m) and abyss, yes other authors have used this standard. But many bioregionalisation schemes make this differentiation.

Lines 158-163: Looks like you have a richness peak in Fig. 3B which is actually a lower bathyal pattern in the northern hemisphere, looks to be correlated with the temperature spike around the same depth and latitude for ED Fig. 4. In addition, some of the richness peaks for truly abyssal species (Fig. 4F) appear to be in areas that contain very few samples (Fig1B). So it is hard to know if this pattern is real or an artefact of using ranges. So I wonder if this equator pattern is biasing the inference on environmental drivers of abyssal diversity.

Line 210: It looks like the authors of this paper have closely copied many of the methods used in that paper, but have applied the methods (albeit slightly different) to a new novel dataset. Do you think that this could be a regional bias in taxonomy/sampling? I know that the Woolley et. al., paper was built largely on work done by taxonomists in the southern hemisphere. Perhaps we are seeing a slight bias towards the northern hemisphere?

Line 214: I know the use of emotive language is a winner for Nature, but I wouldn't say these are dramatically different, we are in fact describing pretty similar patterns. If you look at the raw data Fig1C, the patterns are not that different, and it is the choice to use interpolated data which makes the contrast so stark.

Methodology:

Datasets:

In addition, the current approach does not account for the fact that the data are largely presence-only observations, therefore it is hard to know if species are absent or just not recorded. This is a common problem in many broadscale biodiversity databases and considerable effort is being undertaken in statistical ecology to account for these biases. I realise the authors have tried to account for this via interpolation of ranges. But it just seems at the very least some kind of effort/sampling offset is needed to correct for the interpolated ranges.

The use of iNaturalist also raises a red flag, as it is largely citizen science data. Though very useful under certain circumstances, it really is a different type of data than that is typically vouchered in museum records, and feels like these two types of data are not really compatible. There at least needs to be some depiction of the effort between these two data types and also the species, regions and depths they target.

The authors need to go into more detail about the "second dataset", were these records geo-referenced? Seems like a very different data type, given a broad regional distribution. I also imagine many of the OBIS records might be duplicated in these literature sources (Assuming OBIS data comes from museums/natural history institutes that voucher many specimens used in historical studies/research). There is also a well known publishing bias in the northern hemisphere, it really does make you wonder if this data is adding in undue bias into the richness patterns.

I would remove these records and the iNaturalist data and focus just on the OBIS data. Which turns out to be an impressive dataset (196,876 occurrences). Unless a really good reason can be presented why they are complementary.

Line 529: This seems like a very coarse bathymetric map. GEBCO is now available at 15 arc seconds: https://www.gebco.net/data_and_products/gridded_bathymetry_data/#a1. If you are getting the bathymetric location of the missing 33.3% of occurrences wrong, this is going to strongly bias your results. It also requires that the occurrence geocoordinate is accurate. For example, if you sample a seamount, but take a course (average depth) at the geolocation. You end up having a shallower water species appearing to be occupying deeper water.

Environmental Data:

As CARS2009 isn't supported anymore, I'd be using something like WOA13v2. Is more up to date, and is publicly available: <https://www.nodc.noaa.gov/OC5/woa13/>

Statistical analyses:

Lines 621-643. Did you use a Poisson or Negative binomial model here? As you have a count (richness will be an integer ≥ 0). If you used something like CPUE (catch per unit effort) richness/sampling effort you could maybe get away with a Gaussian model.

Lines 632. I disagree with this statement. By interpolating species ranges you are introducing a very naive/basic species distribution model. So unintentionally you m (unknowingly) and then are potentially capturing these patterns as effects in SLMs and RF. I think the interpolated ranges approach works well as a pattern describing mechanism, making inference on those patterns becomes more tricky, as you are not modelling observations, but rather inferred data patterns.

Reviewer #2 (Remarks to the Author):

General: The manuscript uses open-access data and literature data to uncover the patterns of biodiversity in Asteroidea globally and identify the environmental drivers of biodiversity patterns in this group. I acknowledge the effort in mining the

data from about 700 papers, as it is very important. I also appreciate the partial data quality control and diverse statistical analyses which were done in the study. However, I highly recommend the authors acknowledge other studies which have been done during the last years on discovering the patterns of species richness against latitude and depth around the world, especially in the NW Pacific area. Against what authors claim is novel " We also show for the first time that diversity differs enormously between hemispheres and Atlantic and Indo-West Pacific realms and find the temperature, carbon flux, and geographic complexity as main drivers for these differences. We, therefore, present a rare and valuable spatial description of biodiversity across the global oceans and describe patterns unlike those previously reported for shallow-water or terrestrial environments."; there are a lot of papers published in the last few years uncovering biodiversity patterns in all marine species and across taxa in both shallow and deep-waters, even included even more environmental variables and open database sources such as GBIF which is missing from this paper. They found temperature, carbon flux, and current as important drivers of benthic diversity worldwide, or at a regional scale, as well as reported that the global species richness is bimodal in most taxa, and there are different peaks in bathymetrical species richness which is taxa specific. Some examples are

Jawin, E. R., Walsh, K. J., Barnouin, O. S., McCoy, T. J., Ballouz, R.-L., DellaGiustina, D. N., et al. (2020). Global patterns of recent mass movement on asteroid (101955) Bennu. *Journal of Geophysical Research: Planets*, 125, e2020JE006475. <https://doi.org/10.1029/2020JE006475>

Arfianti T, Costello MJ (2020) Global biogeography of marine amphipod crustaceans: latitude, regionalization, and beta diversity. *Mar Ecol Prog Ser* 638:83-94. <https://doi.org/10.3354/meps13272>

Zhao, Q, Basher, Z, Costello, MJ. Mapping near surface global marine ecosystems through cluster analysis of environmental data. *Ecological Research*. 2020; 35: 327– 342. <https://doi.org/10.1111/1440-1703.12060>

Saeedi, H., Costello, M.J., Warren, D. et al. Latitudinal and bathymetrical species richness patterns in the NW Pacific and the adjacent Arctic Ocean. *Sci Rep* 9, 9303 (2019). <https://doi.org/10.1038/s41598-019-45813-9>

Talmage, S. C. & Gobler, C. J. Effects of Elevated Temperature and Carbon Dioxide on the Growth and Survival of Larvae and Juveniles of Three Species of Northwest Atlantic Bivalves. *Plos One* 6, <https://doi.org/10.1371/journal.pone.0026941> (2011).

As the dataset used here lacks a lot of digital data from OBIS, I would recommend rejecting the paper as stands now and giving the possibility of resubmission if the missing data are integrated and all the analyses are redone. The missing data can definitely change the patterns of diversity across the globe. Also, the language of the manuscript needs substantial revisions. I also could not find any patterns or structure in writing the manuscript. What are the knowledge gaps, how do your results address those gaps, what are the highlights of your paper, how the findings could be applied to other disciplines, etc? Abstract: The keywords are missing. Please revise the abstract, the grammar and structure of the sentences are very confusing, some examples, lines 20, 31, 32, ... The methods used, the aim, the gaps, and the highlight of the paper are missing or now well integrated into the Abstract.

Results and Discussion: There is no clear cut between Introduction, Results, and Discussion. Even if this is a format of the journal, the authors should categorize the text and give a pattern to their findings, also drafting the background studies, the gaps, and how their findings addressed those gaps. After reading the whole text, I was still puzzled about what is new about this manuscript, what were the aims, what were the gaps, and the highlights of the paper.

Methodology: I see major concerns in the data collection and I believe that the dataset could be majorly improved and the analyses should be done based on the new dataset. I highlight shortly my concerns regarding the dataset. However, I highly respect reviewing about 750 literature by the authors and this is very important. I was not able to see if the data mined from the literature are digitized and will be submitted open access or not.

- The dataset only includes data from OBIS, not the Global Biodiversity Information Facility (GBIF). There are a tremendous amount of data in GBIF which has not been submitted and mobilized to OBIS, thus, the data must be completed by integrating the data from both OBIS and GBIF.

- What is the resolution of the environmental layers here used? It seems many of the grids have NAs. I recommend extracting the environmental data from BioOracle <https://www.bio-oracle.org/> as the resolution is 5 arcmin.

- The data quality control and data cleaning procedure to me lacks fundamental checks. I recommend that authors visit https://manual.obis.org/data_qc.html and reclean third data based on the quality control flags recommended by OBIS to remove the data on lands, out of the bathymetry boundary, data related to fossils, etc.

- I am not sure, why the authors used an average for environmental variables per latitude, but not per records per latitude. If you have distribution patches in some areas in one latitude but use an average on 1degree latitude to correlate the species richness with latitude, the results will not be so reliable. One should extract the environmental values from the distribution records of the species, and then make an average of each latitude.

I stopped reviewing the figures and the corresponding legends, as the methods require substantial revisions and the structure of the paper itself. And other important concerns that I had regarding this paper mentioned above.

Version 1:

Decision Letter:

6th October 2023

Dear Mr Carter,

I sincerely apologize for the delay in reaching a decision on your manuscript entitled "Competing processes shape variable deep seafloor diversity between hemispheres". The paper has now been seen by two new reviewers; as I explained

previously, neither reviewer from the first round of review were available to re-review the paper. As you will see from the comments copied below, Reviewer 3 finds the paper to be acceptable for publication aside from some minor points. However, Reviewer 4 raises more substantial concerns, largely related to the presentation and clarity of the paper. It seems that the main issue is that the paper is still formatted in Nature's Letter style, rather than our journal's Article style. Therefore, we ask you to now conform to our formatting as described below. We will therefore need to see your responses to the concerns, along with a revised manuscript, before we can reach a final decision regarding publication.

Please follow the specific instructions for the Article format (<https://www.nature.com/natecolevol/content>) and also see our more detailed formatting guidelines (<https://www.nature.com/natecolevol/submission-guidelines/aip-and-formatting>). In particular, we highlight these items that need revision:

- The abstract should be reduced to 200 words.
- The article should be structured as follows: Introduction (brief, around 500 words), Results, Discussion, Methods. If you wish, you are allowed to have a combined Results & Discussion section, but the section should be structured to provide more clear definition between what you found and what others have found (as Reviewer 4 noted).
- We further recommend that you divide the Results section into subsections. Each subsection should provide a brief description of the method underlying the result; this type of signposting can help guide the reader. This approach should also address Reviewer 4's concerns about clarity.
- Our current estimate of your word count is about 2700 words, but you have substantial room to expand the word count to provide clarity in response to Reviewer 4's concerns.
- Please make use of the additional display item limits (you can add one to the main text and 3 to the Extended Data) to surface important supporting results out of the Supplementary Information.
- Please address Reviewer 4's concerns by ensuring that a result described verbally in the main text of the Results is represented in a Figure with statistical evidence.

* If you have not done so already please begin to revise your manuscript so that it conforms to our Article format instructions at <http://www.nature.com/natecolevol/info/final-submission>. Refer also to any guidelines provided in this letter.

Link Redacted

Nature Ecology & Evolution is committed to improving transparency in authorship. As part of our efforts in this direction, we are now requesting that all authors identified as 'corresponding author' on published papers create and link their Open Researcher and Contributor Identifier (ORCID) with their account on the Manuscript Tracking System (MTS), prior to acceptance. ORCID helps the scientific community achieve unambiguous attribution of all scholarly contributions. You can create and link your ORCID from the home page of the MTS by clicking on 'Modify my Springer Nature account'. For more information please visit www.springernature.com/orcid.

[redacted]

Reviewers' comments:

Reviewer #3 (Remarks to the Author):

I am a second-round reviewer of this manuscript and have been tasked with determining whether the authors have satisfied the first round of reviews.

In general, I find the authors have addressed the points raised by the earlier reviewers, including refining the dataset, interpolation issues, data resolution issues, and introducing a second series of models (SLMs) that have a sampling-intensity offset. The manuscript issues listed by reviewer 2 would appear to be due to the format required by this journal.

The cars2009 datasets are now closed, however, they are highly correlated with more modern datasets so I don't see that changing them would influence the findings of this study. If anything, they are more temporally aligned with the sampling than more modern oceanographic datasets (e.g. Argo float data) which are becoming influenced by climate change. The pattern described by the authors is a strong one, robust to various methods, and largely observable even in the raw data.

I did notice some smaller issues that need correction:

You use richness and diversity as synonyms throughout the text, but typically in macro-ecological literature 'diversity' differs from 'richness' by including evenness (e.g. Shannon Weiner diversity). So the two are not interchangeable. What you have is richness (ie a simple count of species).

Line 65. Change family to relationships?

Line 82. You don't cite Fig. 2b which appears to justify your division of the global into Atlantic and Indo-Pacific realms.

Lines 101-2. I don't fully understand what you are trying to say here. You say an increase in temperate richness is "accompanied" by a decrease in deeper tropical richness. But surely temperate deep-sea richness is what has declined. Maybe you mean that deeper richness declines from the tropics. This needs rewording.

Lines 145-151. Richness around the equator is complex, and in the Atlantic it is affected by the discharge of major sources of fresh water (eg Amazon, Congo, Niger).

Line 161. I think it would be good to specify you are using Random Forest and SLM models (with citations) at this point, to understand what has generated the p values.

Line 162. Is it temperature 'gradients' (ie spatial changes in temperature) that promote tropical diversity? The latitudinal diversity gradient is often attributed to the higher temperatures alone (e.g. by enhancing mutation) or simply the age of the fauna – stretching back to the Mesozoic (ie less extinction than temperate or polar faunas, see O'Hara et al 2019 Nature).

Lines 188-193. Polar POC is highly seasonal in nature (summer only) and, thus the annual POC may not be fully available to the fauna.

Line 202, 206. I am not sure if I understand this. What is a "thermal cliff"? Do you mean a steep gradient? Are you saying this correlate "drives" the faunal richness models?

Line 214. I don't have this reference (34) at hand, but I suspect they are studying asteroids in estuaries where there are massive changes in salinity. The salinity variation in the oceans is not nearly as strong and I don't know of a reference that shows echinoderms are biologically affected by these small changes. For me salinity changes across the oceans are a proxy for water mass, a habitat variable that can be important (also incorporating elements of water flow).

Line 263-5. It is also important to sample at the poles. Interpolation means that ranges are extended throughout the tropics. But this doesn't happen outside the known edge of species ranges. So marginal habitats need more intensive sampling. There was a lot of discussion about the mid-domain artefacts in early diversity studies.

Fig. 2 & 3. Add that these use the interpolated dataset.

Line 549. Change "abyss" to "hadal".

Line 656-7. Can you discuss this in relation to Figure 2b, which shows separate NW Pacific and Aust/NZ clusters?

Line 681. Need a reference for the SLM method.

Extended Fig. 1. Change "Abyssal" on the figure to "Lower Bathyal and Abyssal".

Extended Fig. 3. The legend for the salinity plot is incorrect (it shows the temperature gradient).

Line 868. Do you mean "lower bathyal" rather than "upper abyssal"?

Reviewer #4 (Remarks to the Author):

My apologies for the delay in providing my review.

Like the previous referees, I see merit in this work. I also agree with the findings in principle in that there are no surprises, and the data may provide rare support for several biogeographic hypotheses. The compilation of data is commendable.

A large part of the delay is that I found the paper heavy reading and hard to align the declared findings with the data presented. The mixing of Results and Discussion and use of numbering for references was partly to blame (and partly the fault of the journal style). The SM lacked sufficient legends for me to understand what all the figures were.

The main conclusions relate to the role of temperature and particle flux to the deep sea. However, no data, graphs or maps of these variables and their relationship to asteroid species composition (regions of endemism) or richness are provided. This means I cannot trace the main findings to the data. The text muddles these findings by citing other studies so I am not sure whether the statements refer to the present or other studies. It does quote some statistics but these are not convincing without some visualization of the results.

It is commonplace in biogeography to interpolate between species' reported locations to estimate their range. However, the authors should include other ways to estimate richness that account for sampling effort, such as ES50, Hill Numbers (sense Chao), and fitting GAM models (as in the paper by Chaudhary et al cited). They should also distinguish alpha and gamma richness because these can be quite different. For example, the total number of species in a latitudinal band (gamma) may be higher in the southern hemisphere if there are many endemic species in Southern America, South Africa, Australia and New Zealand, even though the number of species in each may be similar to that in equivalent northern latitudes.

I note previous referees also recommend better ways to estimate richness and the authors say they did it but buried the results in SM. They also refer to the poor structure and writing of the paper, specifically the blending of methods, results and discussion. This is still problematic.

It is true that the iNaturalist data are collected by citizens but only verified research-grade data are submitted to GBIF. These are at least as good as scientist identified (many scientists are not good at taxonomy).

Minor comments

Line 20 - this line could be omitted as it is circular. Depending on how one defines an ecosystem it can be true or false.

Line 47 - now the deep sea is no longer an ecosystem but called a habitat. I suggest it is neither but an environment defined by depth. To be a habitat one needs to specify for what species or assemblage. To be an ecosystem it should have defined boundaries of energy exchange (maybe research will find it does, eg a net sink).

Line 49 - A primary reference for the 90% and 70% would be desirable.

Line 55 and throughout - the word diversity is used which can mean many different things (richness, relative abundance indices, beta - alpha - gamma diversity). I recommend being more specific. e.g, say richness if that is what is meant.

Line 79 - here again, I am not sure how a paper on limpets on one island can support the prior general statement.

Line 90 - please replace ambiguous / with whatever is meant in words (and., or, and or, etc.). Check elsewhere in MS for this. See line 236.

Line 84. this is a new paragraph. What part of the previous paragraph does "this" refer to?

Line 96-97 - this is a key finding, supporting the hypothesis proposed in Costello and Chaudhary 2017 (ref 26). But Fig 4 does not show this as explicitly as a graph of a simpler plot of species range against depth. I recommend such an x-y plot is added to or replaces Figure 4.

Line 105, very recently some papers by Lin et al. in PeerJ also show geographic patterns with depth in global oceans for all fish and could add support to the Discussion.

Line 123 why "intriguingly" - the difference between the Atlantic and IWP is well known and to be expected.

Lines 134, 135. The term "shift" is not appropriate here as the species or richness has not moved (as they may for example due to climate change).

Lines 236-238, this is a truism as depending on what one means by "patterns of diversity" it is always or never true. I suggest deleting the sentence.

Line 145 - where is the evidence or data that "In the Atlantic, spatial heterogeneity is much higher north of the equator"?

Line 149 - consequently this paper does not show what it says it does, no habitat complexity data are provided or correlated.

Line 158_ - It says "we show that the principal sources of energy at the seafloor" but I cannot find these data in the paper. Maybe something is in the SM but it must be in the main paper as by definition SM are supplementary.

Line 161 - how were point data (for fluxes) correlated with species range data in this paper? Where and what are the models referred to here?

Line 174-5. Again where is the data to show these differences in temperature?

Line 177-8. This is very interesting but again, where is the data showing this 1.5oC threshold?

Line 182-3. Why is it assumed that deep-sea cosmopolitan taxa are species complexes when the data here support wider ranges for deep-sea species? There seems no need to perpetuate thinking that there are no such thing as a cosmopolitan species' because they clearly exist.

Line 189 - did the cited paper just speculate or did they show deep-sea species are more sensitive to environmental variation than shallow-water species?

Line 197. What realms - those defined by Spalding et al. (MEOW) or Costello et al 2017 biogeographic realms?

Line 213 - asteroids should be lowercase; only formal names are capitalised.

Lines 215-8 - this sentence is always true and can be omitted.

Line 220. See also recent paper by Vitoreo et al. 2023 in *Ecography* on ophiuroid biogeography.

Line 244 - there is no comparison with terrestrial environments here to support this statement.

Line 252 - "shift in the peak of this curve" - there are no shifts or curves presented in this paper.

Lines 258_. This paragraph could be deleted to make the MS more concise.

Line 374, the WorMS citation is incomplete and add its DOI. See <https://www.marinespecies.org/aphia.php?p=popup&name=citation>

Line 472. Saying datasets are available from somebody is no longer acceptable. Publish the data used on Figshare or back into OBIS or GBIF to enable this study to be reproduced.

Line 532. It is a myth that most data in GBIF and OBIS are from museums and institutions. Most are citizen science and ecological datasets. Please check the source of the datasets used here and then say where they came from.

Line 554. Here Ecoregions are mentioned but they were not in the main text. These are management units but have been aggregated into provinces and realms which are biogeographic. But they are only within EEZ. What about the other 60% of the ocean?

Line 604. Where is the justification for these depth zones used?

Fig 1A is inadequate as the non-blue spots are not clearly visible. I suggest fewer colours with more contrast.

Fig 2A. Richness should not be aggregated by ecoregions or provinces but presented as cells and for the entire ocean where available. The map omits most of the ocean - High Seas.

Fig 2B. As above, what about the rest of the ocean?

Fig 3. What is the scale for the colours in part A?

Extended data

Fig 1 - nice plot, could this be included in main MS as is a key finding?

Fig 3. So here are the environmental data, relegated to SM where few readers will see them and unconnected to the species data.

Table 1. So here I find microscopic text that is the statistics referred to. This is not convincing. Patterns and correlations must be shown to the reader, not hidden away in SM tables.

Similar comments for reset to SM.

*****END*****

Version 2:

Decision Letter:

21st May 2024

Dear Mr Carter,

Your manuscript entitled "Competing processes shape variable deep seafloor diversity between hemispheres" has now been seen again by our 2 reviewers, whose comments are attached. While Reviewer 3 is now satisfied with the study, Reviewer 4 has raised a number of further concerns which will need to be addressed before we can offer publication in Nature Ecology & Evolution.

We therefore invite you to revise your manuscript again taking into account all reviewer comments. Please highlight all changes in the manuscript text file.

* If you have not done so already please begin to revise your manuscript so that it conforms to our Article format instructions at <http://www.nature.com/natecolevol/info/final-submission>. Refer also to any guidelines provided in this letter.

Link Redacted

Nature Ecology & Evolution is committed to improving transparency in authorship. As part of our efforts in this direction, we are now requesting that all authors identified as 'corresponding author' on published papers create and link their Open Researcher and Contributor Identifier (ORCID) with their account on the Manuscript Tracking System (MTS), prior to acceptance. ORCID helps the scientific community achieve unambiguous attribution of all scholarly contributions. You can create and link your ORCID from the home page of the MTS by clicking on 'Modify my Springer Nature account'. For more information please visit www.springernature.com/orcid.

[redacted]

Reviewers' comments:

Reviewer #3 (Remarks to the Author):

The authors have amended the manuscript to account for all my previous comments, and I am happy for this manuscript to proceed to publication.

Reviewer #4 (Remarks to the Author):

Review paper on global biogeography of asteroids

My apology to the authors for this late review. Because I feel it is a worthwhile work to get published and I suspect the first author is a graduate student, I have given more time to provide some detailed suggestions as how to strengthen the paper. That said, the co-authors should have been able to improve parts of the paper before submission even if it is the first authors first paper.

This is an impressive gathering of data for a global scale analysis of a major marine taxonomic group, the starfish and brittlestars. It merits publication following some rethinking of interpretations of the findings, more qualification of the generality of the study, and checking the effects of sampling bias on the results.

The flow and conciseness will be improved if the Results and Discussion are separated, as is normal in primary data analysis papers.

The major weakness of this study is little attempt is made to account for sampling bias. The northern hemisphere peak in sampling bias is well known (see Chaudhary et al. 2016 and 2017 papers in TREE, <http://dx.doi.org/10.1016/j.tree.2017.02.007>). The authors argue there are enough samples to account for this but then also argue more data are needed. There are several ways to do this (see Chaudary et al. 2022, PNAS) using GAM, ES50 and Hill's numbers (rarefaction options). I think it is essential that the authors replot richness using one or more methods to account for richness.

The use of interpolated species ranges is one way to try to account for sampling bias as applied in this study. However, this perpetuates bias because one can only produce polygons for where species have been reported from. Furthermore, this will not account for shifts in species ranges over the last century due to climate change and thus exaggerate equatorial richness (see PNAS paper above – showed thousands marine species shifting away from equator). Better range maps are produced by using species distribution (environmental niche) models although if these use old records they will still exaggerate the ranges (as in the Lin et al. paper cited for fish). It is also true that they can only be run when sufficient data exist, but this is not a reason not to use them, and why not exclude the use of simple interpolation when insufficient data exist?

A key result is the depth gradient in species richness which is relegated to the SM. This should be in the main paper and perhaps broken into subsets for different latitudes and/or oceans.

Length of coastline is a useful indicator of coastline heterogeneity but a better indicator is seabed variability or rugosity (available from BioOracle <https://www.bio-oracle.org/>).

Title – far too broad and should clarify it is only on starfish. It should include asteroids or Asteroidea.

Abstract – last sentence is very weak and can be omitted. This final sentence should focus on the novel implications of the paper.

Introduction

Line 38 – benthos is not a biome; reword. Benthos is an assemblage of species living on the seabed. A biome is widely and long established in ecology as a large area characterised by plants that provide three-dimensional habitat, such as kelp forests. Unfortunately, the term has been used for almost any marine area in the marine literature.

Lines 39 – use primary sources when citing facts as here; i.e., papers which have actually measured the % of the earth that is global.

Line 40-41 – this and elsewhere are “truisms” or statements which are so broad they could be both true and false without more context. One could argue that “large scale patterns .. are WELL understood” as many papers and resources map these patterns (e.g., see AquaMaps website). The simplest solution is just delete such statements as they only add words and not meaning. Other examples:

Line 43-44 all databases have “limited taxon or habitat coverage” (and they always will).

Line 77 – repeats the “more limited” cliché above. It would be better to say how many species were in each study so we know how different they were in scope. Check throughout MS for this phrase and either delete it or make it an objective quantified statement e.g., with only half the number of species than in the present study (see line 301)

Line 53 – delete comprehensive, it is not comprehensive and probably can never be.

Line 170 “difficult to quantify” is a superfluous phrase.

Line 45-47 – one can argue that no two places on Earth are comparable for any number of reasons. Was the Atlantic more glaciated than the North Pacific and Antarctic shelf? I suggest delete this superfluous sentence.

Line 60 – it is not “benthic” diversity, only “asteroid”.

Line 72 – methods should explain what an eco-province is. I assume it is the classification of coastal provinces using expert opinion in the Spalding et al. MEOW classification. The authors may not be aware of the data-derived realms in Costello et al. 2017, Nature Communications. These provide a comparison to MEOW and other systems and are hierarchical. This hierarchy could be compared with the findings in the study for asteroids.

Line 75 – I think there are more studies. E.g. polychaetes in MEPS by Pamungkas et al. The references in this and other papers may provide more examples to enrich the generality of the present study findings.

Line 92 and later refer to bathomes but the papers seems to use different depths for these in different places. A standard definition based on data should be used, not some of the numerous and varying definitions which lack any biological or environmental rationale (as in text books).

Line 104 – ref 23 is a weak one as it is only for one location.

Lines 105-7 – this is true and several studies have suggested this over past century as cited in the Costello and Chaudhary 2017 paper cited. I agree this is an important finding in this study as this fact is generally not understood by scientists who champion exceptional deep-sea diversity.

Line 114-7 – needs a rewrite as unclear. Be especially carefully in claiming peaks when comparing LDG curves when it could be dips.

Lines 122-4 – unclear sentences, “compared with a restriction only”, sentence is too long probably.

Line 125 – state what other taxa are mentioned here, were they also benthic, how comparable are they? Clearly taxa vary in their depth and latitudinal distributions (e.g., auks, penguins, herbivorous to non-herbivorous fish, etc.).

Line 126-7 – yes this has been clearly shown to be due to sampling bias. When this bias was corrected for the latitudinal gradient across all marine species is symmetrical and bimodal (see the 3 Chaudhary et al papers in TREE and PNAS – her PhD may also be online for more details).

Lines 128-31 – I think the Chaudhary et al. PNAS paper accounted for shelf area and coastline length. This aspect needs more thinking through. Area is only important when habitat diversity increases with area up to certain limits – which is why the deep sea has lower richness than coastal waters. If the LDG is symmetrical as found by Chaudhary et al. PNAS then it suggests that simply temperature (or day length) is the primary driver. If not, and sampling bias has been accounted for, then evolutionary history can be shown to be important, especially at family level – as nicely shown for fish in the Lin et al. papers in PeerJ and in the present paper.

Line 134- it would be clearer to be more objective and say how different these biota are in terms of total species and unique (endemic) species.

Line 144-6 – the meaning of these sentences is not clear to me, “bathyal and bathyal and abyssal” and no supporting references cited. This is the first time I heard it claimed that there was a reduction in seafloor due to the Eurasian land mass. Land masses are older than the oceans so changes in sea level and the coming together of the Asia and Australian continents has contributed to high species richness in the IWP.

Line 150 – as mentioned above, this will need rethinking once sampling effort and bias has been considered. The claim that there is no substantive variation in sampling effort across hemispheres does not agree with numerous published studies and the data in OBIS and GBIF.

Line 157 -60 – “topographic heterogeneity of coastline and seafloor” is misleading as only coastline length was analysed here. It would be much better to use “rugosity” (variation in seabed slope per lat-longitude cell), a measure of topographic heterogeneity highly correlated with species richness (see Zhao et al. 2020, Biological Conservation <https://doi.org/10.1016/j.biocon.2020.108536>).

Line 162 – do asteroid and ophiuroids ever compete or occupy similar niches to justify this statement – it may be true but more support would make it more convincing.

Line 167 – habitat diversity is part of biodiversity (includes ecosystem variation) – this sentence could be omitted or revised. Asaad et al. 2017 (Biol Conservation <https://doi.org/10.1016/j.biocon.2018.03.037>) showed high correlation between species and habitat richness for the Coral Triangle.

Lines 171-3 – confusing as no such places are mapped to define boreotropics (a strange term as boreal is north of temperate which is north of tropics, so is boreotropics northern subtropics?). It is best to minimise jargon like this.

Lines 175-182. Needs rethinking. It is a nice theory but is there any evidence that the variation in salinity near these big rivers affects benthic diversity? I think not as the freshwater floats and is rapidly dispersed. This paragraph needs a re-write as the interpretation is not clearly presented and too speculative. Speculation should be supported by stronger examples from this and other studies.

Lines 186-8 – no supporting references cited but see paper by Wei et al. in PLoS One on benthic biomass gradients which, like on land, show a strong temperate relationship.

Line 242-6 – the finding of limited species below 1.5 °C is notable and interesting. Some papers by Daniel Pauly on the GOLT theory and Portner on OCCLT (similar theories) indicate that oxygen becomes limiting at very cold temperatures because it is too cold for the animal to absorb enough oxygen to counter the rate of cold denaturation of proteins. A recent study by Lavin, Pauly and others in *Environmental Biology of Fishes* provided evidence of this in the form that Arctic fish exposed to a degree or two warmer temperature (above 1 or 2 degrees) showed increased growth and size.

Line 249 - unclear meaning, “increasing diatom content” compared to what? “reduction in calcium carbonate.. at the poles” compared to where, and below CCD (see more recent papers by Harris et al. 2023 in *Marine Geology* and Simon-Lledo et al. 2023. Carbonate compensation depth drives abyssal biogeography in the northeast Pacific. *Nat. Ecol. Evol.* <https://doi.org/10.1038/s41559-023-02122-9>). Does the latter paper support the absence of asteroids below the CCD? This would greatly strengthen the proposition in this paragraph.

Line 253 – what is meant by “young” and “but see”. Please elaborate in simpler language. Is it meant that the abyssal fauna is a subset of shallow species as proposed by Rex for molluscs, i.e. the species did not evolve in the abyss (not neoendemics)?

Line 256 – this sentence may mean the opposite – few records generally suggest species to be more endemic not cosmopolitan. Endemic species are geographically rare and more records tend to fill out their distribution to either confirm a restricted range (truly endemic) or not. However, regardless of this clarification this sentence is out of context here and could be omitted or revised to fit the paragraph better.

Line 261 – please include the richness with depth plots in the main MS.

Line 270 – “for the first time in this context” is meaningless because the context is not clear. It is not the first time that low deep sea and polar temperatures have been invoked to explain low species richness in marine taxa, and this is not a surprising finding, while worth noting.

Line 276-80 – seem to be Methods?

Line 281-293 – this merits a re-write. Why are only a few families shown? Are they the most species rich? And as many examples are of only one family then it should be stated which family, not vague phrases like “bathyal families”. This is a nice part of the paper because it shows how different families evolved to fill different geographic and environmental niches. It is another good example of how the latitudinal gradient varies between families but when species are aggregated to Class level there are clear gradients.

Lines 294-5 – a weak sentence and could have been said without any of this research. Revise or omit.

Lines 299-301 – this first statement of the conclusions suggests this is a novel finding. But it is to be expected that patterns in abyssal taxa differ from shallow taxa, right?

Lines 302- this discussion about thermal energy could be clearer by being more explicit about what is meant by “energy” – what is chemical energy if not related to nutrients? Is it oxygen or hydrogen sulphide – but neither were studied here. It may be better to start this paragraph from line 304 (“We show for the first time ...”) and state the temperature threshold which is novel (but the idea of temperature thresholds is not).

Line 311 – I presume nutrient here refers to POC? Better say this and not other nutrient sources (whale and wood falls, hydrogen sulphide vents, etc.). See the important paper by Wei (Wei C-L et al. (2010) *Global Patterns and Predictions of Seafloor Biomass Using Random Forests*. PLoS ONE 5(12): e15323. doi:10.1371/journal.pone.0015323) which is very relevant here. It supports the idea that temperature effects on decomposition and recycling lead to low sediment organic matter in warmer environments (as occurs in forest soils with latitude).

Lines 391 -324 – this is not a conclusion and could be omitted. Conclusions should focus on the implications of the present study.

Lines 324-327 – another truism that does not need to be repeated – of course more sampling will improve knowledge. However, there was no quantification of sampling gaps in this study and there were claims that there was sufficient sampling; both cannot be correct. I encourage the authors to try Chao’s Hill numbers to estimate species richness and indicate where sampling gaps lie.

Line 328 – to say “complete spatial description” is at odds with the previous sentences and cannot be true for either shallow

or deep seas.

Methods

Line 340 – saying the data “will be” available on a institutional website is not acceptable nowadays. The data used should be published with a DOI in a repository such as Figshare or Zenodo. In addition, the new data collated from the literature should be published into GBIF or OBIS (or EMODnet Biology). This would reciprocate the benefits the authors gained from using data from GBIF and OBIS.

Were species only known from fossils excluded? such records occur in GBIF and perhaps OBIS.

Line 344 – each dataset used is like a paper in a journal and if one uses GBIF or OBIS one agrees to cite the datasets (CC-BY licences). Thus, not to do so is a breach of conditions of use. I recommend copying out the citations and placing them in the SM as provided (some will be incomplete).

Line 361 – the authors have greatly misunderstood the iNaturalist dataset. Only “Research grade” data are published into GBIF. These are all verified by independent experts and supported by photographs. In contrast most “scientific” datasets are not independently verified or supported by photographs and are likely to contain more errors. How do the authors know that “experts” identified specimens from research cruises – did they check the metadata of each recorded for who identified the specimens? If so say so but I suspect they did not think of this. Thus, it is likely that some records have been misidentified.

Lines 373-4 – I am surprised that more of these cruise records were not in OBIS. My memory is that considerable data from these research vessels are in OBIS. They would probably be published through the AADC for Australian and NIWA for New Zealand data.

Lines 382-4 – Please make a clearer statement, or a table of the number of records and species from OBIS, GBIF and authors own collation.

Lines 402-3 – the authors should know this exactly from their datasets. Please make a quantified statement of how many species and records with adequate georeferencing were from different kinds of sources.

Line 419- how do trenches create a “disjunct habitat” ? Would not the water mass, temperature, salinity. CCD horizon be the same? If there were too few records below a certain depth for analysis then just say so and not invent debatable reasons.

Line 428 – explain to reader what eco-provinces are (See definition in Spalding et al. source). But note this is only coastal and so omits most of the ocean. Why not use the global classification of marine biogeographic realms instead <https://www.nature.com/articles/s41467-017-01121-2> especially when so much emphasis is on the deep sea in this paper.

Line 445 – another sweeping sentence that can be omitted.

Line 461 – where are the shapefiles from the interpolation available from? E.g. Figshare, Zenodo.

Line 467 – the IWP is not a “basin”. Neither is the Atlantic but it may contain several basins. See Harris et al. 2014 Marine Geology on Geomorphology of the oceans).

Lines 471+ So interpolation was on every species with no requirement for minimum number of records or validation with literature? See previous comments but this method then perpetuates sampling bias. In contrast SDM account for this but have not been used.

Line 489 – this is not true. There is no requirement for an SDM for a minimum number of records per cell. However, the records should be sufficiently geographically spaced to account for the environmental niche of the species.

Line 490-1 – this is also untrue. There are over 25,000 marine species range maps from SDM available from AquaMaps for over ten years, plus expert drawn range maps of 100-1000s species on IUCN Red List Assessment website. Some of these may include asteroids but the authors seem to have been unaware of them.

Line 491 – “many species” – please say how many and/or what proportion.

Lines 496-499 – what is the basis for these depth boundaries? I recommend using one based on environmental data such as the analyses of Ecological Marine Units by Sayre et al. 2017. A three-dimensional mapping of the ocean based on environmental data. *Oceanography* 30(1): 90–103. <https://doi.org/10.5670/oceanog.2017.116> and/or Costello, Sayre et al. 2018. Stratifying ocean sampling globally and with depth to account for environmental variability. (Nature) *Scientific Reports* 8, 11259. DOI:10.1038/s41598-018-29419-1

Line 523 – as noted before, please say exactly what each of these regions of analysis were.

Line 528 – why include silicates when it is then repeatedly stated that they were excluded. Si is only generally important for diatoms; is it important for asteroids?

Line 527 – are phosphates really estimated in this way? No reference is cited. This would not seem to make much sense as

nitrogen is typically limiting in marine waters and varies with seasons and depth. I cannot find any indication that this claim is true e.g., <https://onlinelibrary.wiley.com/doi/10.1111/geb.13813> <https://www.geoplatform.gov/metadata/df6ea7d8-78cc-41f9-8640-93792bcca38b>

The choice of environmental data is curious as the authors seem to have avoided using more well-known and comprehensive sources like BioOracle and GMED.

Line 538 – what “realm boundaries” are being referred to here, Spalding et al., Costello et al.?

Lines 542- the methods seem to redefine the IWP to include the entire west Pacific from the Bering Sea to Australia and New Zealand! At most the IWP covers the Coral Triangle from northern Australia to southern China. Please be more careful with terminology and correct as needed.

Line 547 – no reference cited for claim of Mediterranean being “extremely distinct” (see the EMU and biogeographic sources I have mentioned previously).

Lines 568 and 584 – define SLM and mtry at first mention.

Why was oxygen not included in the analysis? Large mid-depth zones are oxygen deficient and likely to affect asteroids.

Where is the list of species by family used in the study?

References 58 and 59 have author the same as the source name – is that correct?
Check for misspellings.

Ref 60 should be cited by first editors name and then et al. Unfortunately this is hidden behind another click on their webpage but it then gives credit to the editors as is more conventional.

Figures

In all legends just say what the figure is and remove interpretation which is in the main text (and should not repeat what is in the legends). Make all text at least double the size so it is EASY to read. Add in missing parentheses in legends. Replace species diversity with “species richness” or number of species”

Fig 1 – colours of points on map are illegible. The colour scale on the map for sampling effort seems to be duplicated for depth zones.

Fig 2 – Part B does not seem needed and the 3 graphs in Part C should be combined into one plot so it is easier to compare the latitudinal gradients. Part A should be one scale, not three.

Fig 3 – very nice except it excludes most the ocean and all the deep sea that is discussed so much in the paper. It claims to exclude polar regions but does include some of the Arctic, and why exclude them? There are records from Antarctica and the Arctic.

Fig 4 – another nice plot but why only these families and not more? Please add a line plot of total number of species per latitudinal band to should gradient.

Fig 5 – nice plots. Fig 2c should be plotted this way as well. Make plots the same scale for Atlantic and IWP. However, it is not clear what the comparison is. Is it all the Atlantic and West Pacific for the same latitudinal extent? That makes sense but then it is not the IWP. The depths used seem to differ from those in the text and need to be standardised throughout based on some data-driven rationale.

Fig 6 – nice idea but obviously they do not work because there are no scales shown and now different depth zones used than in the last figure. Simple plots of richness per depth, temperature and POC for latitudinal climatological zones and/or oceans may be easier to understand.

Minor comments easily corrected

The text often slips between past and present. All results and methods (eg from line 349) should be in the past tense.

Throughout replace diversity with either richness or number of species. If another measure of diversity is intended then state what it is. E.g., line 240, 291, probably elsewhere as well.

Round % to whole %, at present some are, some to one and some to two decimal places.

Line 83, - nearly always it is best to put “However” at beginning of sentences than break up clauses with it in the middle.

Line 136 – tropico should be tropical.

Line 139 – what is austro-tropical and how can it be in the Atlantic?

Line 140 – here and anywhere else avoid the ambiguous “/” – say and, or, and/or or drop one of the words.

*****END*****

Version 3:

Decision Letter:

18th December 2024

Dear Dr. Carter,

Thank you for submitting your revised manuscript "Competing processes shape variable deep seafloor diversity between hemispheres in Asteroidea" (NATECOLEVOL-220616722C). It has now been seen again by the original reviewers and their comments are below. The reviewers find that the paper has improved in revision, and therefore we'll be happy in principle to publish it in Nature Ecology & Evolution, pending minor revisions to satisfy the reviewers' final requests and to comply with our editorial and formatting guidelines.

We are now performing detailed checks on your paper and will send you a checklist detailing our editorial and formatting requirements early in the new year. Please do not upload the final materials and make any revisions until you receive this additional information from us.

[redacted]

Reviewer #4 (Remarks to the Author):

I thank the authors for the detailed and thorough reply to my review suggestions. I seem to have missed the other reviewer comments so I cannot comment on how they align with mine. The finding that temperature largely explains the patterns advances understanding and supports previous studies. Indeed, it supports the hypothesis that the marine biota are largely distributed according to the environmental conditions rather than dispersal barriers as occurs on land.

The finding of exceptionally low richness below 1.5 oC is also important and may cause the deep-sea super-richness chauvinists food for thought. It also makes sense at many levels (ecological, evolutionary).

However, these results are not clearly shown in the figures. The latitude plots should include temperature (max, min, mean?) and nutrients as well as species richness, and simple X-Y plots of the variables against richness would be better than the 3-D plots with illegible axes. The figures must greatly enlarge the text so it is easily legible.

Reviewers' comments:

Reviewer #1 (Remarks to the Author):

Major comments:

Overall I think this is a great study trying to hypothesise drivers of patterns and processes that shape global patterns of marine biodiversity for one highly novel dataset. I think Nature Ecology and Evolution should consider publishing this paper.

This is an interesting study done on a highly novel global dataset of Asteroids. There clearly has been a huge amount of work put into the gathering of data, developing the manuscript and undertaking analyses. The figures are comprehensive and well done.

My only major concern is the role of sampling effort and the effect that creating range/interpolated maps from biases/patchy data might have on results. I think these concerns could be easily addressed with a few more figures and some more analyses.

- We have re-analysed the full dataset following the reviewer's suggestions. The major points are listed below in brief:
 - Added figure showing global sampling effort and supplementary figures latitudinal, bathymetric and taxa specific uncertainty – ie, where interpolation is being relied upon to a greater or lesser extent: See updated Figure 1 and new Extended Data Figure 7
 - Undertook a new analysis using the global echinoderm sampling effort (~1,350,000 occurrence records from OBIS) to provide a sampling offset for Stationary Linear Models (SLM's) and a sampling weighting for random forest analyses to allow for greater weighting towards regions of higher certainty and to account for sampling effort in the analysis. This approach was used to repeat all statistical analyses across all geographic regions. These analyses test for the effects of patchy sampling effort and all three analyses recover the same major drivers suggesting our results are broadly robust to sampling effort. See Extended Data Table 1 and new Extended Data Table 2, Extended Data Figure 4 and Supplementary Data 2.
 - The text and methods have been amended such that all changes to analytical methods and results have been incorporated see lines: 615-626 and 659-703 in particular.
 - Improved clarity of statistical outputs in supplementary figures to show weighting parameters and correlative associations used by both modelling approaches. See new Extended Data Table 2 and Extended Data Fig 4.

I realise it is very difficult to disentangle observational processes from biodiversity ones. But I would appreciate a little more clarity on some of the datasets, how bias/sampling effort is accounted for in the study and also some reporting of the trends of uncertainty for interpolated/predicted species richness.

- We have improved the clarity of our approach to uncertainty, bias and sampling effort in the following ways:

- Methods have been updated to better clarify the statistical sampling bias offsets and importance of specific parameters used in environmental correlate analysis – See Lines 665-672 and 693-700. New text reads:

Lines 665-672: “The robustness of these analyses, observed trends and statistical correlates and the interpolation approach used was further tested by repeating every analysis across both the interpolated and raw datasets while accounting for a sampling offset (Extended Data Tables 1-2, Extended Data Figure 4, Supplementary Data 2). Sampling offset values were produced by taking a logged value of the total number of unique occurrences of all Echinoderm samples per 1° latitudinal/100m depth cell in the OBIS dataset (~1,350,000 total occurrence records).”

Lines 693-700: “Partial dependence plots for each variable across each geographic and bathymetric split were produced using the R package *plotmo*⁵³ and used to determine whether the importance of a variable was due more to a continual directional impact over the range of values included or to the variable functioning more as an environmental switch with diversity limited above or below a certain value. Pairwise comparisons of variable importance were produced to visualise the impact on diversity of interactions between the most important variables for each geographic and bathymetric split (Extended Data Fig. 4, Supplementary Data 2).”

- Improved clarity of statistical outputs in supplementary figures to show weighting parameters and correlative associations used by both modelling approaches. See new Extended Data Figure 4 and Supplementary Data 2.
- Added figure showing global sampling effort and supplementary figures latitudinal, bathymetric and taxa specific uncertainty – ie, where interpolation is being relied upon to a greater or lesser extent within the latitudinal and bathymetric diversity plots – See Figure 1 and Extended Data Figure 7.

I like the addition of figure one to describe sampling effort, but the mapped figure you cannot see the density of samples per a cell or bioregion. I think it would be reasonable to present this rather than just the spatial locations of Asteroids for shallow, bathyal and abyssal regions. Like Fig. 1B, it would be great to see the sampling effort per cell across latitude/longitude. In fact, it is surprising this has been omitted considering how diligent the authors are on providing latitude/depth maps of sampling effort. The map of spatial occurrences (Fig. 1A) could be moved to the SI. I also think effort needs to be reported on the log scale. This is how it would be used in a model (linear predictor scale) and it would be interesting to see if the sampling effort matches the diversity peak in Fig. 3a. Currently hard to see on the reported count scale because effort is so high in the shallow regions (which is to be expected).

- We have updated Figure 1A in accordance with the reviewer’s comments and reported sampling effort per lat/long degree square on a logarithmic scale as requested. See new Figure 1A.

I am wary of the interpolated ranges approach. I feel it smears much of the nuance when it comes to describing species distributions. I understand that it is commonly used, but if you are then using interpolated ranges to make inference on environmental drivers it seems to me that

this will introduce biases into the inferences, as you are potentially inflating richness where it is low, or unintentionally creating correlations/confounding effects in the interpolated data that did not exist in the raw data.

- As the reviewer notes, interpolated ranges are often used in comparable studies of latitudinal gradients, particularly those incorporating species with small numbers of occurrence records and for which range modelling approaches cannot be used. See e.g:

O'Hara, T.D., Hugall, A.F., Woolley, S.N.C. *et al.* Contrasting processes drive ophiuroid phylodiversity across shallow and deep seafloors. *Nature* **565**, 636–639 (2019).

<https://doi.org/10.1038/s41586-019-0886-z>;

Pyrz, T.W., Willmott, K., Garlacz, R., Boyer, P., Gareca, Y. Latitudinal gradient and spatial covariance in species richness of tropical Lepidoptera in the Andes. *Insect Conservation and Diversity* **7**, 335–364 (2015). <https://doi.org/10.1111/icad.12058>;

Menegotto, A., Rangel, T.F. Mapping knowledge gaps in marine diversity reveals a latitudinal gradient of missing species richness. *Nat Commun* **9**, 4713 (2018).

<https://doi.org/10.1038/s41467-018-07217-7>

- Nevertheless, in accordance with the reviewers concerns, we have tested the robustness of the inferences derived from sampling effort across our dataset by analysing our data in three ways: 1) using the interpolated dataset as in our original manuscript, 2) using interpolated data with a sampling offset as described below and 3) using the raw species data with the same sampling offset. Although there are difference in the absolute values and significance levels, all three approaches crucially identify the same leading drivers of diversity and show the retained role for temperature in shaping diversity in the deep sea, suggesting that these results are broadly robust to sampling inequities. See Extended Data Tables 1-2.
- We have changed the text to better reflect that, regardless of this robustness, the use of interpolation means our results in this instance are estimates and that this approach is vulnerable to potential over inflation of richness: see Lines 59-61 and Lines 579-583.
 - Lines 59-61: New sentence reads: “We compiled interpolated ranges based on the known latitudinal limits for each species within their individual bathymetric ranges to produce **a measure of maximum estimated potential diversity for each degree** of latitude.”
 - Lines 579-583: New sentences read: “This approach greatly reduces sampling bias in our dataset **but may overestimate diversity by overfitting species ranges where habitats or conditions are not continually present.** We therefore analysed our data at the relatively coarse scale of 1° latitudinal intervals **and treated our results as estimated maximum diversity rather than absolute diversity** for each degree of latitude.”
- We have also clarified that manual adjustments were made to species ranges to incorporate known disjunctions or basin specific extents. This approach was taken to

mitigate and minimise potentially smeared nuance in species ranges as also highlighted by the reviewer. We have clarified the text to make this clearer:

- Lines 586-593: New sentence reads “Ranges were not extended over latitudes where the maximum seafloor depth for that latitude was shallower than the bathymetric limits of a species **and where continuous seafloor of the correct depth within a species range was not present. Species known to be anti-tropical or to have significant disjunct portions to their ranges from the published literature were flagged during data validation and their interpolated ranges manually adjusted to encompass the regions of known absence. For analysis of the Atlantic and Indo-West Pacific as separate basins, the ranges of species present in both basins were adjusted to reflect only the limits of the species within that basin.**”

I think this could be easily fixed by using the raw richness and correcting for effort an offset in a statistical model. Sometime like this using a GLM in R (or your software of choice):

species_richness_per_cell ~ 1 + covariates + offset(log(samples_per_cell))

This will effectively correct for richness in areas that are sampled more intensely and remove the need to build metrics/inference on interpolated ranges. Of course this assumes a poisson (log link function) to account for effort in cell/site richness.

- Following the reviewer’s suggestions, we have used this approach across our slm models, both with raw and interpolated data and included a weighting based on the same sampling effort offset in our random forest models.
- Sampling effort for this approach was derived as a logged product of total echinoderm sampling effort in OBIS (~1,350,000 records) summed per lat/depth bin. This approach has been highlighted in the methods (Lines 665-672 and 693-700, new text reads) and all model outputs for interpolated, raw and offset data is presented as data tables in the supplement (See Extended Data Tables 1-2). New Text reads:
 - Lines 665-672: “The robustness of these analyses, observed trends and statistical correlates and the interpolation approach used was further tested by repeating every analysis across both the interpolated and raw datasets while accounting for a sampling offset (Extended Data Tables 1-2, Extended Data Figure 4, Supplementary Data 2). Sampling offset values were produced by taking a logged value of the total number of unique occurrences of all Echinoderm samples per 1° latitudinal/100m depth cell in the OBIS dataset (~1,350,000 total occurrence records).”
 - Lines 693-700: “Partial dependence plots for each variable across each geographic and bathymetric split were produced using the R package *plotmo*⁵³ and used to determine whether the importance of a variable was due more to a continual directional impact over the range of values included or to the variable functioning more as an environmental switch with diversity limited above or below a certain value. Pairwise comparisons of variable importance were produced to visualise the impact on diversity of interactions between the most

important variables for each geographic and bathymetric split (Extended Data Fig. 4, Supplementary Data 2).”

Uncertainty is missing from this analysis. I think it would be prudent to report uncertainty in maps and predictions. As the data is not evenly sampled everywhere I would expect there to be regions which are highly uncertain for the richness estimates where sampling is low or outside of the environmental values observed at survey (raw data) sites. These uncertainties could then be used to identify priorities for future surveys or looking for alternative taxa which could fill these voids.

- We agree with the reviewer that it is important to report some measure of where we are less certain about our estimates. We have highlighted this by comparing our interpolated data to the raw data to show which latitudes and depths rely most heavily on interpolation in new Extended Data Figure 7, and by incorporating sampling effort bias in our analysis as discussed above. The methodology employed here is discussed in a new Section in the Materials and Methods, entitled ‘Uncertainty’, on lines 617-626.
 - Lines 617-626: To illustrate the geographic regions in our analyses where the greatest reliance has been placed on interpolation, we compared raw species richness across the global dataset to the interpolated values. Species were considered continuous within their known bathymetric range at each latitude, but were not considered continuous across latitudes. We then calculated the percentage difference in richness between the raw and interpolated data to give a measure of how much each latitude and depth cell relies on interpolation (Extended Data Fig. 7). We additionally produced a measure of how much each individual species range relies on interpolation by looking at the disjunction of the raw data across the latitudinal range of a species. These data were separately plotted for each of the three major bathomes considered in this analysis, Extended Data Fig. 7 C-E.
- This approach highlights the deep tropics as a region of particularly poor sampling effort and we have amended the text to indicate that this is a region where future sampling efforts would provide the most benefit. See Lines 263-265:
 - Lines 263-265: In particular, we note the vital importance of sampling throughout the understudied tropics to determine the accuracy of our estimates in the face of new discoveries and suggest that this should be a primary target region for future expeditions.

Minor comments:

Line 50: “or risk bias towards common or cosmopolitan species by relying on models” is really an inaccurate comment. This work also included analyses using species richness models that included > 2000 unique species. In addition, the multiple-species Bayesian model was developed to account for sampling bias in the data, but comes at the cost of not being able to include the very rare species in the analysis. This is true for any SDM (species interpolation) that is based on a statistical model.

- Our intention with this sentence was to highlight the general issue with SDM approaches, not as a criticism of this study. Accordingly, we have removed this reference and re-worded the sentence to better reflect the intention of this point.
- Lines 50-52: “Previous studies have focused on limited geographic regions⁵ or at very broad resolutions, risk bias towards common or cosmopolitan species through the use of models rather than collection data, or rely on databases with limited taxon or habitat coverage¹¹

Line 61: abyssal: 2,000-6,000m is not really abyss, more like lower bathyal (2000-3500m) and abyss, yes other authors have used this standard. But many bioregionalisation schemes make this differentiation.

- We have updated the text so that this bathome is now described as “lower bathyal and abyssal” throughout the manuscript and in all appropriate figures.

Lines 158-163: Looks like you have a richness peak in Fig. 3B which is actually a lower bathyal pattern in the northern hemisphere, looks to be correlated with the temperature spike around the same depth and latitude for ED Fig. 4. In addition, some of the richness peaks for truly abyssal species (Fig. 4F) appear to be in areas that contain very few samples (Fig1B). So it is hard to know if this pattern is real or an artefact of using ranges. So I wonder if this equator pattern is biasing the inference on environmental drivers of abyssal diversity.

- The peak in richness identified in 3B is, as the reviewer suggests, probably reflective of the temperature spike in this region and this is discussed in the text as probably linked to the Mediterranean overflow water in the northern Atlantic Lines 208-211
- Lines 208-211: Northern Atlantic bathyal waters are notably warmer and saltier than southern Atlantic waters (Extended Data Fig 10), in part due to the impact of Mediterranean Overflow Water (MOW)³³ and this likely supports higher diversity in the region
- In response to the reviewers concerns about this as a data artefact, our re-analysis of the entire updated global dataset, (see below and new extended data Figures 6-8), suggests that this abyssal pattern is repeated across every hemisphere and region in our study, even in the southern hemisphere and Indo-West Pacific that lack this northern hemisphere temperature spike. The text has been updated throughout to highlight the consistency of this pattern and the regional factors behind its variance: See Lines 153-216.
- Abyssal species have historically been considered cosmopolitan on the basis of very few records and we agree that this may artificially drive our patterns. We have updated the text to reflect this: See Lines 180-183 and included Extended Data Figure 12E to show how much of the abyssal faunal patterns potentially reflect interpolation error.
 - Lines 180-183: “This pattern is replicated to a greater or lesser degree across every lower bathyal and abyssal region studied in our analyses, although these

results may be confounded if a significant number of deep sea cosmopolitan taxa turn out to be species complexes^{30,31}.”

Line 210: It looks like the authors of this paper have closely copied many of the methods used in that paper, but have applied the methods (albeit slightly different) to a new novel dataset. Do you think that this could be a regional bias in taxonomy/sampling? I know that the Woolley et. al., paper was built largely on work done by taxonomists in the southern hemisphere. Perhaps we are seeing a slight bias towards the northern hemisphere?

- We feel that the impact of taxonomic expertise is unlikely to explain the full magnitude of the difference observed. However, we have amended the text to highlight that this may be an important contributing factor.
 - Lines 222-223: “While this may in part reflect taxonomic effort ...”

Line 214: I know the use of emotive language is a winner for Nature, but I wouldn't say these are dramatically different, we are in fact describing pretty similar patterns. If you look at the raw data Fig1C, the patterns are not that different, and it is the choice to use interpolated data which makes the contrast so stark.

- Text has been amended to remove overly emotive language. See new Line 219 and onwards

Methodology:

Datasets:

In addition, the current approach does not account for the fact that the data are largely presence-only observations, therefore it is hard to know if species are absent or just not recorded. This is a common problem in many broadscale biodiversity databases and considerable effort is being undertaken in statistical ecology to account for these biases. I realise the authors have tried to account for this via interpolation of ranges. But it just seems at the very least some kind of effort/sampling offset is needed to correct for the interpolated ranges.

- See responses above regarding interpolation and offsetting. Additionally, we had already made efforts to incorporate known absences in disjunct/anti-tropical ranges prior to initial submission and these are now detailed in full in the methods, see Lines 584-591.
 - Ranges were not extended over latitudes where the maximum seafloor depth for that latitude was outside the bathymetric limits of a species and where continuous seafloor of the correct depth within a species range was not present. Species known to be anti-tropical or to have significant disjunct portions to their ranges from the published literature were flagged during data validation and their interpolated ranges manually adjusted to encompass the regions of known absence. For analysis of the Atlantic and Indo-West Pacific as separate basins, the ranges of species present in both basins were adjusted to reflect only the limits of the species within that basin.

The use of iNaturalist also raises a red flag, as it is largely citizen science data. Though very useful under certain circumstances, it really is a different type of data than that is typically vouchered in museum records, and feels like these two types of data are not really compatible.

There at least needs to be some depiction of the effort between these two data types and also the species, regions and depths they target.

- We have removed all iNaturalist records from the dataset and amended the text and methods accordingly. Removal of this data had no notable impact on the statistical importance of the environmental correlations tested.

The authors need to go into more detail about the “second dataset”, were these records georeferenced? Seems like a very different data type, given a broad regional distribution. I also imagine many of the OBIS records might be duplicated in these literature sources (Assuming OBIS data comes from museums/natural history institutes that voucher many specimens used in historical studies/research). There is also a well-known publishing bias in the northern hemisphere, it really does make you wonder if this data is adding in undue bias into the richness patterns.

- The “second dataset” is not a collection of georeferenced coordinates but a catalogue of literature derived descriptive ranges that were used to verify the OBIS and GBIF coordinate records used in the study, to generate and bound the known bathymetric limits for each species and in the construction of faunal composition lists for the MEOW analysis. (Extended Data Figures 2, Supplementary Data 1). None of this information was used to re-construct additional coordinates for the analysis. We have amended the text to help clarify this concern and no longer use the word ‘dataset’ to refer to the list of references. See Lines 517-529.

Lines 517-529: A catalogue of descriptive geographic ranges for 99.3% (1,903/1,916) of all extant species and known bathymetric limits for 91.7% (1,760/1,916) of all extant species was additionally compiled for use in data validation and faunal assemblage analyses. This catalogue was collated manually from ~750 literature resources spanning 275 years of global research and largely consist of original descriptions, cruise reports and regional faunal monographs. This catalogue comprises full taxonomic hierarchy, maximum and minimum recorded depths and descriptive geographic ranges for all included species. Species for which the bathymetric range was given only as “Intertidal” or “Shallow” were assigned the generic depth range of 0–10 m for inclusion in Extended Data Figure 1 and included in the 0-100m depth cell for all further analysis. Species were further categorised as found in either the Atlantic, Indo-West Pacific, north/east Pacific, Arctic or Antarctic, or to combinations of these regions, to produce broad regional assemblages prior to further analysis.

- The additional 8204 fully georeferenced coordinate records that were added to the dataset following a manual search of the literature/collections at the NHMUK were targeted towards rarer species and regions with poor coverage in OBIS and GBIF specifically to mitigate the northern hemisphere publishing bias and biogeographic gaps. Care was taken to ensure that these additional records were not already present in online repositories. We have amended the text to also clarify this point. See Lines 505-506
 - Lines 505-506 “...and were targeted towards geographic and taxonomic gaps in the GBIF and OBIS datasets.”

I would remove these records and the iNaturalist data and focus just on the OBIS data. Which turns out to be an impressive dataset (196,876 occurrences). Unless a really good reason can be presented why they are complementary.

As above, we removed all iNaturalist records from the dataset following reviewer concerns. In line with a suggestion from reviewer 2, we also incorporated non-iNaturalist records from GBIF, which expanded our final occurrence record total giving a final coordinate dataset of 256,861 validated records.

Line 529: This seems like a very coarse bathymetric map. GEBCO is now available at 15 arc seconds: https://www.gebco.net/data_and_products/gridded_bathymetry_data/#a1. If you are getting the bathymetric location of the missing 33.3% of occurrences wrong, this is going to strongly bias your results. It also requires that the occurrence geocoordinate is accurate. For example, if you sample a seamount, but take a course (average depth) at the geolocation. You end up having a shallower water species appearing to be occupying deeper water.

- We agree with the reviewer that the bathymetric grid we have used to produce depth estimates is coarser than the GEBCO 15 arc second data. However, our data largely lack sufficient geographic precision to warrant more detailed presentation or analyses.
- The potential issue of incorrect depth assignment to species was avoided by using the literature derived bathymetric limits of each species to determine depth ranges, not the bathymetrically estimated values. Grid bathymetric data was only used to compile plots of sampling effort. See Figure 1B.

Environmental Data:

As CARS2009 isn't supported anymore, I'd be using something like WOA13v2. Is more up to date, and is publicly available: <https://www.nodc.noaa.gov/OC5/woa13/>

- We feel that the additional benefit of a slightly more up to date environmental dataset would represent too much work for what are likely to be very marginal changes to our reported results. We additionally feel that the CARS2009 dataset is more useful than the WOA13v2 because values have already been interpolated to the seafloor. However, we are willing to do the extra analyses if the editor feels it is necessary.

Statistical analyses:

Lines 621-643. Did you use a Poisson or Negative binomial model here? As you have a count (richness will be an integer ≥ 0). If you used something like CPUE (catch per unit effort) richness/sampling effort you could maybe get away with a Gaussian model.

- We used a series of *slm*'s fitting the data to a Poisson model and have amended the text to clarify this point. See Lines 681-82.
 - Lines 681-682: "We applied an all-model selection method using the R package *slm* to determine the best models based on AIC scores fitting variables to a Poisson distribution."

Lines 632. I disagree with this statement. By interpolating species ranges you are introducing a very naive/basic species distribution model. So unintentionally you (unknowingly) are potentially

capturing these patterns as effects in SLMs and RF. I think the interpolated ranges approach works well as a pattern describing mechanism, making inference on those patterns becomes more tricky, as you are not modelling observations, but rather inferred data patterns

- As above, we have undertaken new analyses to account for sampling bias, to compare interpolated and non-interpolated data and have amended the text to highlight these changes and remove this statement.

Reviewer #2 (Remarks to the Author):

General: The manuscript uses open-access data and literature data to uncover the patterns of biodiversity in Asteroidea globally and identify the environmental drivers of biodiversity patterns in this group. I acknowledge the effort in mining the data from about 700 papers, as it is very important. I also appreciate the partial data quality control and diverse statistical analyses which were done in the study. However, I highly recommend the authors acknowledge other studies which have been done during the last years on discovering the patterns of species richness against latitude and depth around the world, especially in the NW Pacific area. Against what authors claim is novel “ We also show for the first time that diversity differs enormously between hemispheres and Atlantic and Indo-West Pacific realms and find the temperature, carbon flux, and geographic complexity as main drivers for these differences. We, therefore, present a rare and valuable spatial description of biodiversity across the global oceans and describe patterns unlike those previously reported for shallow-water or terrestrial environments.”; there are a lot of papers published in the last few years uncovering biodiversity patterns in all marine species and across taxa in both shallow and deep-waters, even included even more environmental variables and open database sources such as GBIF which is missing from this paper. They found temperature, carbon flux, and current as important drivers of benthic diversity worldwide, or at a regional scale, as well as reported that the global species richness is bimodal in most taxa, and there are different peaks in bathymetrical species richness which is taxa specific. Some examples are

Jawin, E. R., Walsh, K. J., Barnouin, O. S., McCoy, T. J., Ballouz, R.-L., DellaGiustina, D. N., et al. (2020). Global patterns of recent mass movement on asteroid (101955) Bennu. Journal of Geophysical Research: Planets, 125, e2020JE006475. <https://doi.org/10.1029/2020JE006475>

Arfianti T, Costello MJ (2020) Global biogeography of marine amphipod crustaceans: latitude, regionalization, and beta diversity. Mar Ecol Prog Ser 638:83-94. <https://doi.org/10.3354/meps13272>

Zhao, Q, Basher, Z, Costello, MJ. Mapping near surface global marine ecosystems through cluster analysis of environmental data. Ecological Research. 2020; 35: 327–342. <https://doi.org/10.1111/1440-1703.12060>

Saeedi, H., Costello, M.J., Warren, D. et al. Latitudinal and bathymetrical species richness patterns in the NW Pacific and the adjacent Arctic Ocean. Sci Rep 9, 9303 (2019). <https://doi.org/10.1038/s41598-019-45813-9>

Talmage, S. C. & Gobler, C. J. Effects of Elevated Temperature and Carbon Dioxide on the Growth

and Survival of Larvae and Juveniles of Three Species of Northwest Atlantic Bivalves. Plos One 6, <https://doi.org/10.1371/journal.pone.0026941> (2011)

- We have included additional references including some of these studies as requested. See in particular, Lines 48-51.

As the dataset used here lacks a lot of digital data from OBIS, I would recommend rejecting the paper as stands now and giving the possibility of resubmission if the missing data are integrated and all the analyses are redone. The missing data can definitely change the patterns of diversity across the globe

- We have incorporated the entirety of the GBIF dataset into our study, applied the same rigorous cleaning methods to this data and re-run every analysis on this new data set. See lines 482-497 for a full account of the data validation and incorporation processes used.
 - Lines 482-497: The GBIF dataset comprises 476,834 records assigned to 2,779 names of which 370,208 represent observations that are both georeferenced and identified to species level (last accessed in September 2022) and include fifty-two species not present in the OBIS dataset used in the original manuscript. There is a large overlap of shared records between the GBIF and OBIS datasets although, in general, OBIS has better coverage along the north Atlantic and north Indo-Pacific coastlines and GBIF has better coverage along South African and the western North and South American coastlines. The two datasets are largely complementary over large portions of infrequently sampled deep ocean, particularly in the Pacific where many records are present in only one dataset. Validation was performed in the same way as the OBIS dataset with the exception that records flagged as derived from the iNaturalist citizen science repository were removed. This additional step was taken to ensure that the final dataset comprises only expert identified records, primarily from research cruises and museum collections and to reduce misidentification issues. The post-validated GBIF dataset represents 240,182 records comprising 82.15% (1574/1916) of extant described species and representatives of 94.3% (331/352) extant genera and all 38 extant families.

Ultimately, as can be seen from the updated manuscript, the inclusion of this data had marginal effects on the overall patterns of diversity reported across the global, hemispheric and ocean basin portions of our data (See changes to Figure 3 and Figure 5) but does improve taxon and geographic coverage.

Also, the language of the manuscript needs substantial revisions. I also could not find any patterns or structure in writing the manuscript. What are the knowledge gaps, how do your results address those gaps, what are the highlights of your paper, how the findings could be applied to other disciplines, etc?

- We appreciate the reviewers comments about the structure and presentation of results in study although we are limited in how this is presented by the required structure of

the journal. Nevertheless, we have updated the text in several places to better illustrate the highlights of our paper and how this study fits in the context of recent work where possible. See in particular lines 166-169, lines 178-182, and lines 199-219.

- Lines 166-172: “In lower bathyal and abyssal water, nutrient availability has previously been considered the principal driver of diversity in what is broadly a cold and highly thermally homogenous environment^{2,29}. Conversely, our results suggest that while carbon flux is important in some models, unexpectedly, temperature remains a significant factor shaping diversity at these depths, often to a greater extent than POC flux (Extended Data Tables 1-2, Extended Data Figs 4-5; temperature, $p < 0.001$).”
- Lines 180-195: “This pattern is replicated to a greater or lesser degree across every lower bathyal and abyssal region studied in our analyses, although these results may be confounded where deep sea cosmopolitan taxa turn out to be species complexes^{30,31}. In the bathyal and abyssal southern hemisphere, where POC flux peaks at lower latitudes ($\sim 40^\circ\text{S}$), we further find a clearer and more significant relationship (bathyal: $p < 0.001$, $\text{RF} = 104.46644$; upper bathyal and abyssal: $p < 0.05$, $\text{RF} = 111.19269$) between POC and diversity than in the northern hemisphere where POC flux peaks in arctic waters ($\sim 65^\circ\text{N}$) that are below the thermal diversity limit (bathyal: $p < 0.001$, $\text{RF} = 104.28641$; upper bathyal and abyssal: $p > 0.05$, $\text{RF} = 102.17163$). Deep-water taxa are theorised to be more sensitive to small variations in temperature than coastal species^{29,32} and so we do not rule out the possibility that more localised effects might be observed with a greater sampling effort, but in effect, the coldest high latitude waters prevent the establishment of a species rich community despite relatively elevated levels of chemical energy. Thus, we suggest, for the first time in this context, that this apparent disconnect between nutrient availability and diversity can be explained by there being a thermal limit in bathyal and abyssal waters below which high diversity cannot be sustained even in the presence of elevated nutrient levels.”

Please revise the abstract, the grammar and structure of the sentences are very confusing, some examples, lines 20, 31, 32, ... The methods used, the aim, the gaps, and the highlight of the paper are missing or now well integrated into the Abstract.

- We have altered the abstract where possible to better show the aims and highlights of the paper, see Lines 29-33 and 37-42.
 - Lines 29-33: “Using these data, we have produced a complete comparison of seafloor diversity between hemispheres and across major oceanic biogeographic realms. We reveal highly variable patterns of biodiversity between depth zones and substantial hemispheric divergence, explained in part by geographic, environmental, and taxonomic factors.”
 - Lines 37-42 “Uniquely, we show deep-water biodiversity is restricted above a thermal threshold, with colder waters consistently unable to support high species richness in Asteroidea regardless of other factors across the deep ocean floor. We also show for the first time across an entire clade that diversity differs enormously between basins and find variation in environmental and geographic factors as main drivers for these differences.”

Results and Discussion: There is no clear cut between Introduction, Results, and Discussion. Even if this is a format of the journal, the authors should categorize the text and give a pattern to their findings, also drafting the background studies, the gaps, and how their findings addressed those gaps. After reading the whole text, I was still puzzled about what is new about this manuscript, what were the aims, what were the gaps, and the highlights of the paper.

- We have substantively clarified our text throughout the manuscript and hope that it is now clearer.

Methodology: I see major concerns in the data collection and I believe that the dataset could be majorly improved and the analyses should be done based on the new dataset. I highlight shortly my concerns regarding the dataset. However, I highly respect reviewing about 750 literature by the authors and this is very important. I was not able to see if the data mined from the literature are digitized and will be submitted open access or not.

- The data are digitized and will be available from the lead author on request after publication. This is now clarified in the methods.
 - Line 472: All datasets used are available from H. Carter on request.

The dataset only includes data from OBIS, not the Global Biodiversity Information Facility (GBIF). There are a tremendous amount of data in GBIF which has not been submitted and mobilized to OBIS, thus, the data must be completed by integrating the data from both OBIS and GBIF.

- As above, we have followed the reviewers' suggestions and incorporated GBIF data alongside our original, largely OBIS based, dataset. Many GBIF records lack species level identification or full coordinate references (106,626) so the apparent size difference between the two datasets is smaller than on first appearance. This nevertheless represents a substantial improvement in the taxonomic and geographic coverage of our dataset all-be-it with only small impacts on the overall patterns identified.

What is the resolution of the environmental layers here used? It seems many of the grids have NAs. I recommend extracting the environmental data from BioOracle <https://www.bio-oracle.org/> as the resolution is 5 arcmin.

- See response to Reviewer One above. We believe that the environmental resolution in the CARS2009 dataset, although incomplete on steep continental margins is suitable for the resolution on the dataset presented here and that the amount of time required to make this change would result in only very minor alterations to the final model outputs. If the Editor feels that this re-analysis is essential to the paper, we are happy to utilise this higher resolution dataset.

The data quality control and data cleaning procedure to me lacks fundamental checks. I recommend that authors visit https://manual.obis.org/data_qc.html and reclean third data based on the quality control flags recommended by OBIS to remove the data on lands, out of the bathymetry boundary, data related to fossils, etc.

- Our data has been cleaned above the OBIS standard, with all records on land, out of bathymetric range and from non-extant species removed from the dataset. We have rigorously cleaned every individual species range by comparison of the coordinate points

with the described literature to additionally remove points that would otherwise be missed using the standard OBIS clean-up methods. This is detailed in the methods section lines 531-542:

- Lines 531-542:

Data Validation and Processing

Although OBIS and GBIF data are primarily derived from museum or institutional databases⁴⁰, some records lack species level identification or contain erroneous coordinate placement on land or outside of the known geographic range of a species when compared with our catalogue (described above). Such errors tend to arise either through automated geo-referencing from imprecise locality data or translocation of coordinates⁴⁴. To minimise these error sources, all terrestrial records were removed from the dataset and every record was manually compared against the known range in our catalogue for each species. Records outside these limits were excised from the dataset. A small number of literature derived records were excluded due to inconsistencies between coordinate placement and the published descriptions of the collection locality.

• I am not sure, why the authors used an average for environmental variables per latitude, but not per records per latitude. If you have distribution patches in some areas in one latitude but use an average on 1degree latitude to correlate the species richness with latitude, the results will not be so reliable. One should extract the environmental values from the distribution records of the species, and then make an average of each latitude.

- We thank the reviewer for highlighting this potential source of error in our study. There are several reason we have not done this. Firstly, our use of interpolation allows us to estimate diversity across regions of the seafloor that have been sporadically or never sampled. Using only environmental data from sampled regions would mean that we would be unable to use these estimates (representing around half of all study cells) in our analyses. Secondly, outside of shallow latitudes, sampling is sporadic and incomplete and our estimates for environmental values over these regions would be based on a tiny, potentially unrepresentative dataset. Thirdly, such an approach would massively bias the average for a latitude towards regions with higher sampling effort and therefore not capture the true range of environmental factors across the oceans that the re-sampling grid approach we have used does. The subdivided distributional data used for oceanic basin analyses was additionally only compared against basin specific re-analysis of the environmental dataset pertinent to the appropriate part of the range. Finally, as discussed above, we have reanalysed our data across three different models incorporating sampling offsets and accounting for interpolation to test the robustness of our dataset.

Response to Reviewers

We would like to thank both reviewers for their constructive and thorough feedback on our manuscript. We have listed all specific changes to the manuscript below, but the main body of manuscript text has also been substantively re-written in response particularly to Reviewer 4's issues around legibility and readability and now includes sub-headings that we feel better partition the work. Further to Reviewer 4's comments, we have also increased the number of in-text references to specific figures/data tables such that all claims made in the text are now fully backed up with direct reference to ~~these tables or figures in the text~~, Extended Data (accessible online) or Supplementary Data. Some of reviewer 4's comments around the referencing style and use of ~~E~~extended ~~D~~ata figures relate to formatting styles that are fixed for the journal and so we are unable to change these. It is our understanding that ~~E~~extended ~~D~~ata figures (where we have placed our data tables and some larger figures) should not be considered ~~the same~~ as ~~S~~upplementary ~~figures~~ ~~Data in other publications~~, but would appear directly alongside the online version on this paper and are the correct place for these data according to the journal style.

Reviewer 3:

You use richness and diversity as synonyms throughout the text, but typically in macro-ecological literature 'diversity' differs from 'richness' by including evenness (e.g Shannon Weiner diversity). So the two are not interchangeable. What you have is richness (ie a simple count of species).

We have corrected this throughout the text.

Line 65. Change family to relationships?

~~This change of word has been made~~Done.

~~See Line 63: "...and accounting for taxonomic relationships" ✗~~

Line 82. You don't cite Fig. 2b which appears to justify your division of the global into Atlantic and Indo-Pacific realms.

Fig. 2~~b~~B (renumbered Fig. 3b) is now cited in context on ~~lines 133-134~~ ✗.

~~"Patterns of species richness are also notably different between the Atlantic and the IWP (Fig. 5), which each have highly distinctive faunal compositions (Fig. 3b)."~~

New Sentence reads: ~~"Patterns of species richness are also notably different between the Atlantic and the IWP (Fig 5), which each have highly distinctive fauna (Fig 2B)."~~

~~This~~Figure 2b- is also reference~~d~~s in the methods:

~~Line 538: "Oceanic realm boundaries were based on the limits identified in Figure 3b... Oceanic realm boundaries were based on the limits identified in Figure 2B." ✗~~

Lines 101-2. I don't fully understand what you are trying to say here. You say an increase in temperate richness is "accompanied" by a decrease in deeper tropical richness. But surely temperate deep-sea richness is what has declined. Maybe you mean that deeper richness declines from the tropics. This needs rewording.

We agree that this passage was confusingly written and did not correctly express what we intended. This has now been changed to now read:

~~Lines 114-117~~ ✗: ~~"In the southern hemisphere, below ~100 m, species richness peaks in temperate latitudes whereas species numbers drop off below this depth in the tropics, while there is a more~~

Formatted: Font: Bold

Formatted: Font: Italic

Formatted: Font: Bold

Formatted: Font: Bold

Formatted: Font: Italic

Formatted: Font: Italic

Formatted: Font: Bold

Formatted: Font: Bold

Formatted: Font: Italic

Formatted: Font: Bold

Formatted: Font: Bold

Formatted: Font: Italic

gradual latitudinal shift towards a temperate peak of richness in the northern hemisphere that is only readily apparent below ~750 m (Fig. 2). In the southern hemisphere, below ~100 m species richness peaks in temperate latitudes whereas species numbers drop off below this depth in the tropics, while there is a more gradual latitudinal shift towards a temperate peak of richness in the northern hemisphere that is only readily apparent below ~750 m."

Lines 145-151. Richness around the equator is complex, and in the Atlantic it is affected by the discharge of major sources of fresh water (eg Amazon, Congo, Niger).

We agree with the reviewer that this region is ~~more~~ complex and richness potential affected by more factors than we had previously stated. We have re-written this section with additional data on coastline length across the equator to better support our statements.

~~Lines 169-177X-X: "Marine habitat and spatial heterogeneity has rarely been investigated at scale and is difficult to directly quantify^{33,35,36} but is here roughly estimated as a function of length of available coastline. In the Atlantic, spatial heterogeneity and coastline length in the boreotropics (48,831 km) is more than 2.5 times greater than in the austrotropics (18,185 km), driven principally by the complex basin and island systems in the Caribbean³⁷. This matches patterns of higher asteroid richness north of the equator (Fig 5), although additional factors such as the major freshwater discharges from the Amazon and Congo Rivers in the austrotropics are likely to contribute to the substantial reduction in faunal count observed south of the equator."~~
Marine habitat and spatial heterogeneity has rarely been investigated at scale and is difficult to directly quantify but is here roughly estimated as a function of length of available coastline length. In the Atlantic, spatial heterogeneity and coastline length in the boreotropics (48,831 km) is more than 2.5 times greater than in the austrotropics (18,185 km), driven principally by the complex basin and island systems in the Caribbean³⁷. This matches patterns of higher asteroid richness north of the equator (Fig 5), although additional factors such as the major freshwater discharges from the Amazon and Congo Rivers in the austrotropics are likely to contribute to the substantial reduction in faunal count observed in this region south of the equator."

Formatted: Font: Bold

Formatted: Font: Italic

Formatted: Font: Italic

Formatted: Font: Italic

Formatted: Font: Italic

Formatted: Font: Italic

Line 161. I think it would be good to specify you are using Random Forest and SLM models (with citations) at this point, to understand what has generated the p values.

Change made.

~~Lines 200-204: X-X: "These results were generated using two different statistical model approaches, stationary linear models (SLM's) and Random Forest regression analyses and are robust across multiple iterations accounting for both the effects of sampling effort and potential over-estimation through interpolation (Extended Data Tables 1-2). These results were generated using two different statistical model approaches, stationary linear models (SLM's) and Random Forest regression analyses and are robust across multiple iterations accounting for both the effects of sampling effort and potential over-estimation through interpolation (Extended Data Table 2)."~~
These results were generated using two different statistical model approaches, stationary linear models (SLM's) and Random Forest regression analyses and are robust across multiple iterations accounting for both the effects of sampling effort and potential over-estimation through interpolation (Extended Data Table 2)."

Formatted: Font: Bold

Formatted: Font: Italic

Formatted: Font: Italic

Line 162. Is it temperature 'gradients' (ie spatial changes in temperature) that promote tropical diversity? The latitudinal diversity gradient is often attributed to the higher temperatures alone (e.g. by enhancing mutation) or simply the age of the fauna – stretching back to the Mesozoic (ie less extinction than temperate or polar faunas, see O'Hara et al 2019 Nature).

We agree that there are several major theories attributing higher tropical richness primarily to the higher temperatures and/or greater age of the tropics. This sentence was not intended to directly comment on this and rather reflect that **asteroid species** richness increases as temperature increases across latitudes. To clarify this, we have amended the text as follows:

~~Lines 191-193: X-X: "Higher temperatures at lower latitudes correlate with increased tropical richness in global shallow waters (Extended Data Tables 1-2, p<0.001)."~~
Higher temperatures at lower

Formatted: Font: Bold

Formatted: Font: Italic

latitudes correlate with increased tropical richness in global shallow waters ($p < 0.001$) (Table XXX)....”

Lines 188-193. Polar POC is highly seasonal in nature (summer only) and, thus the annual POC may not be fully available to the fauna.

We have updated the text to better account for additional factors that may be important in producing these results, particularly the seasonality of POC flux at higher latitudes.

Lines 248-253X-X: *“This threshold effect may not be solely linked to thermal effects alone. The seasonality and increasing diatom content of POC flux at temperate and polar latitudes⁴¹ and the reduction in the calcium carbonate-based fraction of marine snow at the poles and below the carbonate compensation depth⁴² may act as further stressors and impose additional ecological constraints on species richness and diversity at polar latitudes and bathyal and abyssal depths. This threshold effect may not be solely linked to thermal effects alone. The seasonality and increasing diatom content of POC flux at temperate and polar latitudes (Gardner et al. 2006, Dipper 2022) and the reduction in the calcium carbonate-based fraction of marine snow at the poles and below the carbonate compensation depth (Boudreau et al. 2010) may act as further stressors and impose additional ecological constraints on species richness and diversity at polar latitudes and bathyal and abyssal depths.”*

Formatted: Font: Bold

Formatted: Font: Bold, Not Highlight

Formatted: Font: Italic

Formatted: Font: Italic

Formatted: Font: Italic

Formatted: Font: Italic

Formatted: Font: Italic

Line 202, 206. I am not sure if I understand this. What is a “thermal cliff”? Do you mean a steep gradient? Are you saying this correlate “drives” the faunal richness models?

We agree that this section was somewhat difficult to interpret. We have added a new main text figure (Fig. 6) to better illustrate this ‘thermal threshold’ in our lower bathyal and abyssal data and amended the text to remove reference to this interaction ‘driving’ faunal richness and instead that it appears to be linked to the faunal richness patterns.

Lines X-X: Lines 240-246: *“A pairwise comparison of the impacts of temperature and POC flux on diversity in this region (Fig. 6, Extended Data Fig. 6) suggests that low species richness is linked not to a linear interaction between the two variables but by the presence of a lower ‘thermal threshold’ around 1–1.5°C below which high species number cannot be maintained. Below this threshold, increasing levels of POC flux only correspond with negligible increases in species richness, but above this threshold they display a strong positive interaction that appears to be independent of additional thermal effects (Fig. 6C).”* *“A pairwise comparison of the impacts of temperature and POC flux on diversity in this region (Extended Data Fig. 6) suggests that low species richness is linked not to a linear interaction between the two variables but by the presence of a lower ‘thermal threshold’ around 1–1.5°C below which high species number cannot be maintained. Below this threshold, increasing levels of POC flux only correspond with negligible increases in species richness, but above this threshold they display a strong positive interaction that appears to be independent of additional thermal effects (Fig. X).”*

Formatted: Font: Bold, Highlight

Formatted: Font: Italic

Formatted: Font: Italic

Formatted: Font: Italic

Line 214. I don’t have this reference (34) at hand, but I suspect they are studying asteroids in estuaries where there are massive changes in salinity. The salinity variation in the oceans is not nearly as strong and I don’t know of a reference that shows echinoderms are biologically affected by these small changes. For me salinity changes across the oceans are a proxy for water mass, a habitat variable that can be important (also incorporating elements of water flow).

We agree that this is not really a relevant reference here and that the changes in salinity analysed in our studies are generally small and not biologically relevant for asteroids. We have amended the text to reflect this and to include reference to the role of water mass boundaries in species range delimitation and that salinity can act as a proxy for this.

Lines 219-227X-X: *“Within these regional analyses, we also find significant correlations with salinity at bathyal and abyssal depths (Atlantic/IWP, p<0.001, Extended Data Tables 1-2), an environmental correlate which is not significant in the global analysis. Although some marine regions, such as major river estuaries or semi-enclosed basins, do show pronounced salinity differences when compared with most of the global oceans, salinity does not vary significantly across the majority of the seafloor (Extended Data Figs 3-5) and not in a way that is likely to be biologically meaningful in restricting asteroid distributions³⁹. Rather, we interpret this apparent relationship in terms of salinity acting as a proxy for major water mass boundaries which may be important delimitators of species ranges within basins⁴⁰.”*
Within these regional analyses, we also find significant correlations with salinity at bathyal and abyssal depths (Atlantic/IWP, p<0.001, Extended Data Tables 1-2), an environmental correlate which is not significant in the global analysis. Although some marine regions, such as major river estuaries or semi-enclosed basins, do show pronounced salinity differences when compared with the majority of most of the global oceans, salinity does not vary significantly across the majority of the seafloor (Extended Data Figs. 4, 9-10) and not in a way that is likely to be biologically meaningful in restricting asteroid distributions [REF]. Rather, we interpret this apparent relationship in terms of salinity acting as a proxy for major water mass boundaries which may be important delimitators of species ranges within basins [REF].”

Formatted: Font: Bold

Formatted: Font: Italic

Formatted: Font: Italic

Formatted: Font: Italic

Formatted: Font: Italic

Formatted: Font: Italic

Line 263-5. It is also important to sample at the poles. Interpolation means that ranges are extended throughout the tropics. But this doesn't happen outside the known edge of species ranges. So marginal habitats need more intensive sampling. There was a lot of discussion about the mid-domain artefacts in early diversity studies.

We agree and have amended the text to reflect this.

Lines X-X/ Lines 324-328: *“In particular, we note the vital importance of sampling throughout the understudied tropics to determine the accuracy of our estimates of species numbers in the face of new discoveries, and at the poles to establish the true extents of species ranges in marginal habitats. We suggest that both regions should be primary targets for future scientific expeditions.”*
In particular, we note the vital importance of sampling throughout the understudied tropics to determine the accuracy of our estimates of species numbers in the face of new discoveries, and at the poles to establish the true extents of species ranges in marginal habitats. We suggest that both regions should be primary targets for future scientific expeditions.”

Formatted: Font: Bold, Highlight

Formatted: Font: Bold

Formatted: Font: Italic

Fig. 2 & 3. Add that these use the interpolated dataset.

Done for Figure- 3. Figure- 2. does not use interpolated data but does incorporate range data from literature sources where coordinate points were not available

Line 549. Change “abyss” to “hadal”.

Text changed to

Lines 418-420: Line X: *“...and because the trenches that characterise this bathome create a disjunct habitat and have been extremely variably sampled⁶⁴.”*
“...and because the trenches that characterise this bathome create a disjunct habitat and have been extremely variably sampled.”

Formatted: Font: Bold, Not Italic

Formatted: Font: Italic

Formatted: Line spacing: single

Formatted: Font: Italic

Formatted: Font: Italic

Line 656-7. Can you discuss this in relation to Figure 2b, which shows separate NW Pacific and Aust/NZ clusters?

We have added some text to clarify why, despite a certain faunal distinctiveness for these two regions, we included them for our IWP analysis. Temperate Australia and New Zealand in particular do cluster most closely with the rest of the IWP, but were highlighted to indicate a certain faunal uniqueness.

Formatted: Font: Not Bold

Lines 539-543: "Although the temperate regions of Australia and New Zealand and the north-west Pacific form clusters distinct from the remainder of the Indo-west Pacific, we have included these within the IWP basin analyses for simplicity and to allow these basins to cover an equivalent latitudinal extent to the Atlantic."

Formatted: Font: Italic

Formatted: Font: Italic

Line 681. Need a reference for the SLM method.

Reference added

Extended Fig. 1. Change "Abyssal" on the figure to "Lower Bathyal and Abyssal".

Figure changed as requested

Extended Fig. 3. The legend for the salinity plot is incorrect (it shows the temperature gradient).

Figure changed as requested

Line 868. Do you mean "lower bathyal" rather than "upper abyssal"?

We do, and this has been changed.

Reviewer #4 (Remarks to the Author):

My apologies for the delay in providing my review.

Like the previous referees, I see merit in this work. I also agree with the findings in principle in that there are no surprises, and the data may provide rare support for several biogeographic hypotheses. The compilation of data is commendable.

We thank the reviewer for their constructive comments on our manuscript and interest in our research.

A large part of the delay is that I found the paper heavy reading and hard to align the declared findings with the data presented. The mixing of Results and Discussion and use of numbering for references was partly to blame (and partly the fault of the journal style). The SM lacked sufficient legends for me to understand what all the figures were.

We have taken a number of steps to improve the readability of the manuscript.

- We have broken up the text into sections that group our key findings into more interpretable and readable sections, each with an appropriate heading to help guide the reader
- We have also added more references throughout the manuscript to our figures and results tables to make it easier to follow and to ensure that all results quoted in the manuscript can be traced back to the underlying data.

- We have also ensured that figure legends adequately explain figures and tables and have added extra detail where appropriate.
- We have substantively re-written large portions of the manuscript to better clarify our points and to increase the clarity of our points.

-The main conclusions relate to the role of temperature and particle flux to the deep sea. However, no data, graphs or maps of these variables and their relationship to asteroid species composition (regions of endemism) or richness are provided. This means I cannot trace the main findings to the data. The text muddles these findings by citing other studies so I am not sure whether the statements refer to the present or other studies. It does quote some statistics but these are not convincing without some visualization of the results.

We have added more references to our figures and results tables, ensuring that all major statements are now backed up with the corresponding data to make it easier to know when we are referring to our own data. These data were present in the original manuscript primarily as extended data figures/tables but these should not be considered the same as supplementary data in other journals.

It is our understanding that Extended Data figures (which we have largely retained for our data tables and some larger figures) should not be considered the same as Supplementary Data in other publications but would appear directly alongside the online version on this paper and are the correct place for these data according to the journal style.

Formatted: Space After: 8 pt

~~The main conclusions relate to the role of temperature and particle flux to the deep sea. However, no data, graphs or maps of these variables and their relationship to asteroid species composition (regions of endemism) or richness are provided. This means I cannot trace the main findings to the data. The text muddles these findings by citing other studies so I am not sure whether the statements refer to the present or other studies. It does quote some statistics but these are not convincing without some visualization of the results.~~

Formatted: Space After: 0 pt

It is commonplace in biogeography to interpolate between species' reported locations to estimate their range. However, the authors should include other ways to estimate richness that account for sampling effort, such as ES50, Hill Numbers (sense Chao), and fitting GAM models (as in the paper by Chaudhary et al cited). They should also distinguish alpha and gamma richness because these can be quite different. For example, the total number of species in a latitudinal band (gamma) may be higher in the southern hemisphere if there are many endemic species in Southern America, South Africa, Australia and New Zealand, even though the number of species in each may be similar to that in equivalent northern latitudes.

I note previous referees also recommend better ways to estimate richness and the authors say they did it but buried the results in SM. They also refer to the poor structure and writing of the paper, specifically the blending of methods, results and discussion. This is still problematic.

Specific details of these changes are listed below and described in the methods, but we have accounted for sampling bias across both models used in this study by weighting/controlling for sampling effort.

-Sampling offset values were ultimately derived from the log of all echinoderm samples per 1° latitudinal/100 m depth cell in the OBIS dataset (~1,350,000 total occurrence records) so that true variation in asteroid species richness were better accounted for. We agree with the original reviewer

and reviewer 4 that our approach of just presenting interpolated data without reference to sampling effort was flawed. We have and have presented the full output of all of our random forest and SLM model analyses across different degrees of sampling offset and interpolation in Fig. 6, the extended data tables—Extended Data Tables 1-2; Extended Data Fig. 6 and Supplementary Data 2. As noted above, we feel these are the most appropriate places for these data according to the journal style and will be directly available alongside the online version of this article.

We have not attempted to use the other methods for estimating species richness such as those listed above by the reviewer because we ultimately feel that interpolation (with appropriate weighting/sampling bias control) is the best metric to use approach given the particular constraints of our dataset. Although our dataset is global in scope, only the shallow bathome has a consistent sampling effort, great consistently high enough to allow for us to calculate es50 or Hill numbers across all cells. Below this bathome, there are many 100 m latitudinal cells that do not contain enough samples to adequately use such methods, and which would therefore have to be dropped from our analysis. Our concern is that this would then potentially greatly bias our results towards only those regions which are better sampled and that this could lead to erroneous correlations. We deliberately chose not to include modelled species data, such as those produced using GAM's because of the risk that estimating species richness using environmental data would lead to circularity when then trying to show which environmental variables correlate with species richness.

It is true that the iNaturalist data are collected by citizens but only verified research-grade data are submitted to GBIF. These are at least as good as scientist identified (many scientists are not good at taxonomy).

iNaturalist Data were removed at the request of previous Reviewers.

We thank the reviewer for their constructive comments on our manuscript. As partially acknowledged, some of the issues with our manuscript are imposed by the journal style, particularly the use of numbered references and the use of extended data figures alongside main text figures. We do however agree that the overall readability of the study was low and that it was difficult to trace some of the results quoted to the appropriate figures or data tables.

— We have made significant changes to the structure and clarity of our manuscript which we hope help to break up the text and group our key findings into more interpretable and readable sections. We also hope that the greater references to our figures and results tables which we have included in the text back up all of the major statements we have made and make it far easier to interpret our results.

We have also amended the text and the methodology to make it clearer what changes we made to our ways of estimating species richness as a response to the previous referees. Specific details of these are listed below and described in the methods, but we have accounted for sampling bias across both types of model used in this study by weighting/controlling for sampling effort. Sampling offset values were ultimately derived from the log of all echinoderm samples per 1° latitudinal/100 m depth cell in the OBIS dataset (~1,350,000 total occurrence records) so that true variation in asteroid species richness were better accounted for. We agree with the original reviewer that our approach of just presenting interpolated data without reference to sampling effort was flawed and have presented the full output of all of our model analyses across different degrees of sampling offset and interpolation in the extended data tables. We have not attempted to use other methods for estimating species richness such as those listed above because we ultimately feel that interpolation is the best metric to use given the particular constraints of our dataset. Although it is global in scope, only the shallow bathome has the consistent sampling effort great enough to allow

Commented [SW1]: This needs tweaking to better respond to the reviewer

Formatted: List Paragraph, Bulleted + Level: 1 + Aligned at: 0.72 cm + Indent at: 1.36 cm

Commented [SW2]: This needs expanding. Do not rely on this reviewer to read your replies to the other Reviewer. Spell everything you have done out again - cut and paste if necessary.

Commented [SW3]: What do you mean here by model

Commented [SW4]: See note below

Commented [SW5]: Where? By this reviewer? I would spell out the actual methods

us to calculate es50 or Hill numbers across all cells. Below this bathome, there are many 100m latitudinal cells that do not contain enough samples to adequately use such methods and which would therefore have to be dropped from our analysis. Our concern is that this would then potentially greatly bias our results towards only those regions which are better sampled and that this could lead to erroneous correlations. We deliberately chose not to include modelled species data, such as those produced using GAM's because of the risk that estimating species richness using environmental data would lead to circularity when then trying to show which environmental variables correlate with species richness.

Minor comments

Line 20 - this line could be omitted as it is circular. Depending on how one defines an ecosystem it can be true or false.

Line removed. The whole abstract has also been re-written and shortened. This line now reads

~~Lines 18-19: Line X: "The occurrence, strength, shape and drivers of global distributional trends in species richness throughout the deep sea are broadly inconclusive and poorly explored at scale^{1,2}. The occurrence, strength, shape and drivers of global distributional trends in species richness throughout the deep sea are broadly inconclusive and poorly explored at scale³⁻²"~~

- Formatted: Font: Bold, Not Highlight
- Formatted: Font: Italic
- Formatted: Font: (Default) +Body (Calibri), Not Bold, Italic
- Formatted: Font: (Default) +Body (Calibri), Not Bold, Italic

Line 47 - now the deep sea is no longer an ecosystem but called a habitat. I suggest it is neither but an environment defined by depth. To be a habitat one needs to specify for what species or assemblage. To be an ecosystem it should have defined boundaries of energy exchange (maybe research will find it does, eg a net sink).

We agree that our definition of ecosystem/habitat was too vague and used interchangeably. We have amended this definition to refer to the deep sea as a 'biome'.

Line 38X: "The deep benthos is by far the largest single **biome** on Earth..."

- Formatted: Font: Bold
- Formatted: Not Highlight

Line 49 - A primary reference for the 90% and 70% would be desirable.

We agree and have added in-text references here.

See Refs 3-4 (Lines 604-607):

3. Taylor, M. L. & Roterman, C. N. Invertebrate population genetics across Earth's largest habitat: The deep-sea floor. *Molecular Ecology*, **26**, 4872-4896 (2017).

4. Ramirez-Llodra, E. et al. Deep, diverse and definitely different: unique attributes of the world's largest ecosystem. *Biogeosciences*, **7**, (2010).

ADD REF

- Formatted: Font: Bold
- Formatted: Font: Bold
- Formatted: Font: 11 pt, Italic
- Formatted: Font: 11 pt, Italic
- Formatted: Justified
- Formatted: Font: 11 pt
- Formatted: Font: 11 pt, Italic
- Formatted: Font: 11 pt
- Formatted: Font: 11 pt, Italic
- Formatted: Font: 11 pt
- Formatted: Font: 11 pt, Italic
- Formatted: Space After: 0 pt

Line 55 and throughout - the word diversity is used which can mean many different things (richness, relative abundance indices, beta - alpa - gamma diversity). I recommend being more specific. e.g, say richness if that is what is meant.

We agree with the R_eviewer₁ and this is a point also raised by R_eviewer 3. We have amended the text throughout to specify-clarify that we are talking about species richness throughout.

Line 79 - here again, I am not sure how a paper on limpets on one island can support the prior general statement.

Again, we agree with the R_eviewer that a more suitable reference should be used and have added a more wide-ranging study which found the same result to the text here.

See Ref.19 (Lines 641-643)

19. Sanciango, J. C., Carpenter, K. E., Etnoyer, P. J. & Moretzsohn, F. Habitat availability and heterogeneity and the Indo-Pacific Warm Pool as predictors of marine species richness in the Tropical Indo-Pacific. PLOS ONE 8, e56245 (2013).

ADD REF

Line 90 - please replace amiguous / with whatever is meant in words (and., or, and or, etc.). Check elsewhere in MS for this. See line 236.

We have checked the manuscript and replaced this where noticed. We could not find the / referred to on line 90 but have replaced that found on line on line 236

Line X: Lines 289-290: "Conversely, we identified a small number of shallow and upper bathyal families endemic to Antarctica. Conversely, we identified a small number of shallow and upper bathyal families endemic to Antarctica..."

Line 84. this is a new paragraph. What part of the previous paragraph does "this" refer to?

We agree that this section was confusingly written and unclear. We have substantively re-written this portion of text to bring this sentence within the main body of the paragraph for clarity.

Lines 72-77: X-X "Grouping shallow species ranges by marine eco-province¹⁴, identified the central Indo-West Pacific (IWP) and to a lesser extent the Atlantic Caribbean Basin as hotspots of high tropical diversity (Fig. 3a). While this global pattern is similar to that seen in other terrestrial and shallow marine groups¹⁵⁻¹⁷, it is in contrast to a previous study of shallow water asteroids which proposed low tropical species richness, based on a more limited taxon and habitat sample dataset¹⁰. Grouping shallow species ranges by marine eco-province¹⁰, identified the central Indo-West Pacific (IWP) and to a lesser extent the Atlantic Caribbean Basin as hotspots of high tropical diversity (Fig. 2a). While this global pattern is similar to that seen in other terrestrial and shallow marine groups¹¹⁻¹³ this it is in contrast to a previous study of shallow water asteroids, which proposed low tropical species richness, based on a more limited taxon and habitat sample dataset⁹"

Line 96-97 - this is a key finding, supporting the hypothesis proposed in Costello and Chaudhary 2017 (ref 26). But Fig 4 does not show this as explicitly as a graph of a simpler plot of species range against depth. I recommend such an x-y plot is added to or replaces Figure 4.

Prompted by the R_eviewer we re-analysed our data of species range size against depth. This did not reveal any significant trend in range size with increasing depth at the species level, and a A positive trend was however only observed at the family level, which is illustrated in figure 4 and Supp Data 3. Although some species do have large range sizes at depth, so too do some shallow species and there are a large number of rare deep-water species with very small observed range sizes. While it is

Formatted: Font: Bold, Font color: Custom Color(RGB(36,36,36)), Ligatures: None

Formatted: Font: (Default) +Body (Calibri), 11 pt, Italic

Formatted: Font: (Default) +Body (Calibri), 11 pt

Formatted: Font: (Default) +Body (Calibri), 11 pt, Italic

Formatted: Highlight

Formatted: Font: Italic

Formatted: Font: Bold, Not Italic

Formatted: Font: Italic

Formatted: Font: Italic

Formatted: Font: Italic

Formatted: Font: Italic

Formatted: Font: Italic

Formatted: Font: Italic

Formatted: Font: Italic

possible that ~~these latter~~ ~~these deep-sea species ranges~~ will be shown to be much larger with greater sampling effort, we cannot say this directly from our dataset alone. Indeed, our study suggests that species richness is more evenly distributed ~~in~~ ~~across~~ the deep-sea, rather than species having larger ranges *per se*. As such, we have amended this sentence to better reflect that we only observed this effect at higher taxonomic levels.

Formatted: Font: Italic

Formatted: Font: Italic

~~Lines 101X-107X: "While species richness in the Lower Bathyal and Abyssal bathome remains highest outside of the tropics, the relative differences in faunal number between latitudes are greatly reduced (Fig. 2e). Such a homogenization of species richness has been moted in other taxa^{2,23} at these depths and reflects both the decrease in sampling effort and observed species numbers at these extreme depths (Fig. 1c). The apparent tendency for deep-water clades to occupy more cosmopolitan ranges across larger and less variable habitats (Fig. 4) may also play a role in the homogeneity of species distribution at these depths^{24,25}"~~ ~~While species richness in the Lower Bathyal and Abyssal bathome remains highest outside of the tropics, the relative differences in faunal number between latitudes are greatly reduced (Fig. 3E). Such a homogenization of species richness has been suggested noted in other taxa^{2,22} at these depths and reflects both the decrease in sampling effort and observed species numbers at these extreme depths (Fig 1c). The apparent tendency for deep-water clades to occupy more cosmopolitan ranges across larger and less variable habitats (Fig 4) may also play a role in the homogeneity of species distribution at these depths^{2,2}"~~

Formatted: Font: Bold

Formatted: Font: Italic

Formatted: Font: Italic

Formatted: Font: Italic, Font color: Auto, Not Highlight

Formatted: Font: Italic

Formatted: Font: Italic, Not Highlight

Formatted: Font: Italic, Not Highlight

Formatted: Font: Italic

Formatted: Font color: Black

Line 105, very recently some papers by Lin et al. in PeerJ also show geographic patterns with depth in global oceans for all fish and could add support to the Discussion.

We thank the reviewer for alerting us to this useful study which we have included in our discussion.

~~Lines 93-94: Lines X: "Such bimodal temperate peaks of deep-water richness have also been observed in other taxa^{2,22}"~~ ~~"Bimodal temperate peaks of deep-water richness have also been observed in other taxa^{2,2}"~~

Formatted: Font: Bold, Not Highlight

Formatted: Font: Italic

Formatted: Font: Italic

Formatted: Font: Italic

Line 123 why "intriguingly" - the difference between the Atlantic and IWP is well known and to be expected.

We agree that this is a not an unusual finding and have removed this word.

~~Lines X-X: Lines 133-134: "Patterns of species richness are also notably different between the Atlantic and the IWP (Fig. 5), which each have highly distinctive faunal compositions (Fig. 3b)"; "Patterns of species richness are also notably different between the Atlantic and the IWP (Fig 5), which each have highly distinctive faunal compositions (Fig 2B)"~~

Formatted: Highlight

Formatted: Font: Italic

Lines 134, 135. The term "shift" is not appropriate here as the species or richness has not moved (as they may for example due to climate change).

We agree that this could be a misleading word and have amended the text.

~~Lines 136-139: Lines X-X: "This tropico-temperate peak persists into the bathyal and abyssal zone, gradually becoming more boreo-temperate with increasing depth in a similar, but less pronounced, trend to that observed in the global dataset (Fig. 2)." "This tropico-temperate peak persists into the bathyal and abyssal zone, gradually becoming more boreo-temperate with increasing depth in a similar, but less pronounced, trend to that observed in the global dataset (Fig. 3)." "~~

Formatted: Font: Italic

Lines 136-138, this is a truism as depending on what one means by "patterns of diversity" it is always or never true. I suggest deleting the sentence.

We agree that this rather sweeping general statement is always a truism and not particularly relevant to our work here. We have removed the sentence accordingly.

Line 145 - where is the evidence or data that "In the Atlantic, spatial heterogeneity is much higher north of the equator"?

The sentence was lacking data to support the claim and we have added values on coastline length as a rough proxy for spatial heterogeneity. Defining 'complexity' is not a straightforward process and so we have used this simple measure only as a rough indication of potential habitat area. Specifically:

Lines 169-182; Lines X-X: *"Marine habitat and spatial heterogeneity has rarely been investigated at scale and is difficult to directly quantify^{33,35,36} but is here roughly estimated as a function of length of available coastline. In the Atlantic, spatial heterogeneity and coastline length in the boreotropics (48,831 km) is more than 2.5 times greater than in the austrotropics (18,185 km), driven principally by the complex basin and island systems in the Caribbean³⁷. This matches patterns of higher asteroid richness north of the equator (Fig 5), although additional factors such as the major freshwater discharges from the Amazon and Congo Rivers in the austrotropics are likely to contribute to the substantial reduction in faunal count observed south of the equator. The tropical IWP is contrastingly far more even in spatial complexity and coastline extent, with the boreotropical coastline (122,730 km) only 1.25 times longer than the austrotropical coastline (97,727 km). This evenness is reflected in the more equal distribution of species across the tropical IWP (Fig. 5) and these patterns in both oceans support an important role for spatial factors in the promotion of species richness."* ~~Marine habitat and spatial heterogeneity has rarely been investigated at scale and is difficult to directly quantify^{33,35,36} but is here roughly estimated as a function of length of available coastline. In the Atlantic, spatial heterogeneity and coastline length in the boreotropics (48,831 km) is more than 2.5 times greater than in the austrotropics (18,185 km)...~~

Formatted: Not Highlight

Formatted: Font: Italic

Formatted: Font: Italic

Formatted: Font: Italic

Formatted: Font: Italic

Formatted: Font: Italic

~~And throughout the remainder of this paragraph.~~ We have also added a sentence to the methods to explain how this coastline length was calculated.

Commented [SW6]: Not clear

Lines 550-552X-X: *"We further used coastline length to establish a measure of habitat complexity for the tropics across both the IWP and Atlantic. This was calculated in GIS software using the natural earth 10 m coastline shapefile trimmed to each region of interest"* ~~We further used coastline length to establish a measure of habitat complexity for the tropics across both the IWP and Atlantic. This was calculated in GIS software using the natural earth 10 m coastline shapefile trimmed to each region of interest."~~

Formatted: Font: Bold

Formatted: Font: Bold, Not Highlight

Formatted: Font: Bold, Italic

Formatted: Font: Italic

Line 149 - consequently this paper does not show what it says it does, no habitat complexity data are provided or correlated.

Formatted: Font: Font color: Auto, Ligatures: Standard + Contextual

Please see response above covering Lines 169-182.

Formatted: Font: Bold

Line 158 - It says "we show that the principal sources of energy at the seafloor" but I cannot find these data in the paper. Maybe something is in the SM but it must be in the main paper as by definition SM are supplementary.

~~All these data are available as various figures and data tables in the extended data figures. The Associate Editor has confirmed that the extended data figures in which we placed these data are in this journal, it is our understanding that these are not equivalent to the Supplementary Data and will in fact appear alongside the article online.~~

Line 161 - how were point data (for fluxes) correlated with species range data in this paper?
Where and what are the models referred to here?

We agree that it was unclear what models were being referred to here and have added details to the text. This was also brought to our attention by Reviewer 3.

Lines 200-204X-X: "These results were generated using two different statistical model approaches, stationary linear models (SLM's) and Random Forest regression analyses and are robust across multiple iterations accounting for both the effects of sampling effort and potential over-estimation through interpolation (Extended Data Tables 1-2)."~~"These results were generated using two different statistical model approaches, stationary linear models (SLM's) and Random Forest regression analyses and are robust across multiple iterations accounting for both the effects of sampling effort and potential over-estimation through interpolation (Extended Data Table 2)."~~

Formatted: Font: Bold

Formatted: Font: Bold, Not Highlight

Formatted: Font: Italic

Line 174-5. Again where is the data to show these differences in temperature?

All these data are available ~~as various figures and data tables in the extended data figures. In this journal, it is our understanding that these are not equivalent to the supplementary data and appear alongside the article online.~~
in graphical form in Extended data figure 3-5.

Line 177-8. This is very interesting but again, where is the data showing this 1.5oC threshold?

~~All these data are available as various figures and data tables in the extended data figures. In this journal, it is our understanding that these are not equivalent to the supplementary data and appear alongside the article online. We do however agree that this specific data was not easy to find even in the extended data and have added a new main text figure (Figure 6X) to better illustrate this for the global dataset; A new extended data fig (Extended Data Fig. 6) to show this for all abyssal regions studied, and included all interaction plots for every bathome and basin tested to, along with Supplementary Data 2XX.-~~

Formatted: Font: Not Bold

Formatted: Font: Bold

Formatted: Font: Bold

Line 182-3. Why is it assumed that deep-sea cosmopolitan taxa are species complexes when the data here support wider ranges for deep-sea species? There seems no need to perpetuate thinking that there are no such thing as a cosmopolitan species' because they clearly exist.

We have removed reference to species complexes, but as detailed above, do not agree that our study supports wider ranges specifically for individual deep-sea species. We do not dispute that cosmopolitan species exist, as has been shown in several studies, ~~but suggest that further studies may reveal cryptic species in some under-investigated taxa, but do disagree that all species that have historically been considered cosmopolitan actually are so.~~ We have, amended the text so that this statement is not in isolation ~~and as part of a larger discussion of additional factors which could have some bearing on our results.~~

Lines X-X:248-257: "This threshold effect may not be solely linked to thermal effects alone. The seasonality and increasing diatom content of POC flux at temperate and polar latitudes⁴¹ and the reduction in the calcium carbonate-based fraction of marine snow at the poles and below the carbonate compensation depth⁴² may act as further stressors and impose additional ecological constraints on species richness and diversity at polar latitudes and bathyal and abyssal depths. The abyssal fauna is also probably relatively young⁴³; but see⁴⁴ and so may have had less time to develop the complex community structures characterising older assemblages. Further, the historical tendency for abyssal species to be

Formatted: Font: Bold

Formatted: Font: Italic

Formatted: Space Before: 0 pt

Formatted: Font: Italic

Formatted: Font: Italic

Formatted: Font: Italic

Formatted: Font: Italic

Formatted: Font: Italic

Formatted: Font: Italic

Formatted: Font: Italic

Formatted: Font: Italic

considered cosmopolitan based on a small number of records⁴⁵⁻⁴⁸ may impact the results presented here. This threshold effect may not be solely linked to thermal effects alone. The seasonality and increasing diatom content of POC flux at temperate and polar latitudes⁴ and the reduction in the calcium carbonate based fraction of marine snow at the poles and below the carbonate compensation depth³ may act as further stressors and impose additional ecological constraints on species richness and diversity at polar latitudes and bathyal and abyssal depths. The abyssal fauna is also probably relatively young²; (but see¹) and so may have had less time to develop the complex community structures characterising older assemblages. Further, the historical tendency for abyssal species to be considered cosmopolitan based on a small number of records¹⁻³ may impact the results presented here.”

Formatted: Font: Italic

Formatted: Font: Italic

Line 189 - did the cited paper just speculate or did they show deep-sea species are more sensitive to environmental variation than shallow-water species?

Formatted: Font: 12 pt, Not Italic, Font color: Auto, (Asian) Chinese (Simplified, Mainland China), Ligatures: None

~~This paper was largely speculative.~~ We have removed this section as we felt it was irrelevant to the larger findings of our study.

Line 197. What realms - those defined by Spalding et al. (MEOW) or Costello et al 2017 biogeographic realms?

Our revision of the text has removed this section, but the realms defined here are derived from the Spalding et al. regions as referred to in the methods:

Lines 538-540: ~~Lines X-Y: “Oceanic realm boundaries were based on the limits identified in Figure 3b but limited to 50°S as an approximate margin for the Southern Ocean which has a distinct and complex fauna largely shared across its Atlantic and Indo-West Pacific sectors...” Oceanic realm boundaries were based on the limits identified in Figure 2B, but limited to 50° S as an approximate margin for the Southern Ocean which has a distinct and complex fauna largely shared across its Atlantic and Indo-West Pacific sectors.”~~

Formatted: Font: Italic

Formatted: Font: +Body (Calibri), 11 pt, Italic

Formatted: Font: Italic

Line 213 - asteroids should be lowercase; only formal names are capitalised.

We have corrected this

Lines 215-8 - this sentence is always true and can be omitted.

We agree ~~that this statement was rather generic~~ and ~~it this sentence~~ has been omitted. ~~as it does not contribute to the findings of this study~~

Line 220. See also recent paper by Victoreo e tal. 2023 in Ecography on ophiuroid biogeography.

We have added this important reference

Line 244 - there is no comparison with terrestrial environments here to support this statement.

We agree that this study does not directly refer to any study of terrestrial environments other than as brief mention in references and have accordingly removed the word terrestrial from this sentence. Line now reads:

Lines 299-300: ~~Lines X-X: "In conclusion, we describe patterns of deep bathyal and abyssal richness in asteroids that differ substantially from those found in most shallow marine taxa⁵³⁻⁵⁵ in conclusion, we describe patterns of deep bathyal and abyssal richness in asteroids that differ substantially from those found in most shallow marine taxa¹⁻³,"~~

Formatted: Not Highlight

Formatted: Font: Italic

Formatted: Font: +Body (Calibri), 11 pt, Italic

Formatted: Font: +Body (Calibri), 11 pt, Italic

Line 252 - "shift in the peak of this curve" - there are no shifts or curves presented in this paper.

We agree that this turn of phrase was potentially misleading and have removed reference to this in the text. We have re-written this section to better clarify our findings.

Lines 308-313: ~~Lines X-X: "Under our model, peaks in latitudinal species richness become broader and species richness more evenly distributed with increasing depth, broadly through decreased environmental variation. The bimodal peaks of temperate richness apparent at bathyal depths are closely linked to elevated nutrient availability on either side of the tropics. At abyssal depths diversity is relatively evenly distributed at lower latitudes but is ultimately restricted towards the poles by a deep-water thermal threshold. Under our model, peaks in latitudinal species richness become broader and more even with increasing depth, broadly as a result of decreased environmental variation. The bimodal peaks of temperate richness apparent at bathyal depths are closely linked to elevated nutrient availability on either side of the tropics. At abyssal depths diversity is relatively evenly distributed at lower latitudes but is ultimately restricted towards the poles by a deep-water thermal threshold."~~

Formatted: Font: +Body (Calibri), 11 pt, Italic

Lines 258_. This paragraph could be deleted to make the MS more concise.

We have retained this paragraph because we feel it is useful as a conclusion to the study ~~and because we are not close to the word limit for article length.~~

Line 374, the WorMS citation is incomplete and add its DOI.

See <https://www.marinespecies.org/aphia.php?p=popup&name=citation>

We have updated this reference

Line 472. Saying datasets are available from somebody is no longer acceptable. Publish the data used on Figshare or back into OBIS or GBIF to enable this study to be reproduced.

We agree that datasets should be easily available on publication. Our dataset will be available from <https://data.nhm.ac.uk/dataset/> on publication. All new coordinate data from collections data used in this study but not previously published are fed directly into the appropriate GBIF datasets from the NHM collections portal.

Formatted: Font: Not Bold

Formatted: Font: +Body (Calibri), 11 pt

Formatted: Font: +Body (Calibri), 11 pt

Formatted: Font: +Body (Calibri), 11 pt

Formatted: Font: +Body (Calibri), 11 pt

Field Code Changed

Formatted: Font: +Body (Calibri), 11 pt

Formatted: Font: +Body (Calibri), 11 pt

Formatted: Font: +Body (Calibri), 11 pt

Formatted: Font: +Body (Calibri), 11 pt

Line 532. It is a myth that most data in GBIF and OBIS are from museums and institutions. Most are citizen science and ecological datasets. Please check the source of the datasets used here and then say where they came from.

We thank the reviewer for drawing this to our attention. Although several of the largest contributors to OBIS and GBIF for the Asteroidea are museum or institutional, more are ecological survey datasets and we have amended the text to reflect this.

Lines 402-403X-X: *“OBIS and GBIF data are primarily derived from museum or institutional databases or the results of ecological surveys⁵⁸, but some records lack OBIS and GBIF data are primarily derived from museum or institutional databases or the results of ecological surveys³³, but some records lack...”*

- Formatted: Font: Bold
- Formatted: Font: Bold, Not Highlight
- Formatted: Font: +Body (Calibri), 11 pt, Italic
- Formatted: Font: +Body (Calibri), 11 pt, Italic
- Formatted: Font: +Body (Calibri), 11 pt, Italic

Line 554. Here Ecoregions are mentioned but they were not in the main text. These are management units but have been aggregated into provinces and realms which are biogeographic. But they are only within EEZ. What about the other 60% of the ocean?

The part of the study referred to here is specifically only discussing shallow water richness patterns as grouped by marine eco-province. We were not investigating deep-sea diversity patterns in this particular part of our analysis. *This is now made clearer by the use of by specifying that this is shallow water richness only.*

Line 72: *“Grouping shallow species ranges by marine eco-province¹⁴...”*

a section heading entitled, “XXXXX”. While species were initially assigned to eco-region as a first step, we ultimately aggregated these by eco-province for biogeographic comparison between regions. This is ~~fully~~ described in the methods.

- Formatted: Font: Italic
- Formatted: Font: +Body (Calibri), 11 pt, Italic
- Formatted: Font: +Body (Calibri), 11 pt, Italic
- Formatted: Font: +Body (Calibri), 11 pt, Italic
- Formatted: Font: Bold

Lines 423-431 Lines X-X: *“To investigate the shallow water faunal compositions, we allocated each species to one or more of 232 marine ecoregions of the world¹⁴. Species were allocated either on the basis of a coordinate record falling within the limits of the ecoregion or if the descriptive ranges in our catalogue placed them within the region. Coordinate records were allocated to biogeographic regions using the R package meowR¹⁴ and manually verified using GIS software. We ultimately grouped these results by marine eco-province and used these to compare faunal composition and patterns of β -diversity as this maximised faunal information between regions and minimised variability based on inconsistent sampling effort (Figs. 1, 3). To investigate shallow water faunal compositions, we allocated each species to one or more of 232 marine ecoregions of the world¹⁰. Species were allocated either on the basis of a coordinate record falling within the limits of the ecoregion or if the descriptive ranges in our catalogue placed them within the region. Coordinate records were allocated to biogeographic regions using the R package meowR¹⁰ and manually verified using GIS software. We ultimately grouped these results by marine eco-province and used these to compare faunal composition and patterns of β -diversity as this maximised faunal information between regions and minimised variability based on inconsistent sampling effort (See Fig. 1).”*

- Formatted: Not Highlight
- Formatted: Font: Italic
- Formatted: Font: Italic
- Formatted: Font: Italic
- Formatted: Font: Italic
- Formatted: Font: Italic

Line 604. Where is the justification for these depth zones used?

We have added a reference to the study that these bathome limits was based on.

- Formatted: Font: Not Bold, Font color: Auto, Pattern: Clear, Ligatures: Standard + Contextual

Lines X-X: Lines 495-496: *“Interpolated ranges were summed per degree of latitude then split bathymetrically into three major bathomes based on those defined in² (Shallow: 0–200 m, Upper Bathyal: 200–2,000 m, Lower Bathyal and Abyssal: 2,000–6,000 m).” Interpolated ranges were summed per degree of latitude then split bathymetrically into three major bathomes based on those defined in³ (Shallow: 0–200 m, Upper Bathyal: 200–2,000 m, Lower Bathyal and Abyssal: 2,000–6,000 m).”*

- Formatted: Font: Bold, Highlight
- Formatted: Font: Italic
- Formatted: Font: Italic
- Formatted: Font: Italic

Fig 1A is inadequate as the non-blue spots are not clearly visible. I suggest fewer colours with more contrast.

We have amended the figure as requested and reduced the number of colours in our scale to better enable visualisation of the redder 'highly sampled' cells.~~ADD~~

Fig 2A. Richness should not be aggregated by ecoregions or provinces but presented as cells and for the entire ocean where available. The map omits most of the ocean - High Seas.

Fig 2B. As above, what about the rest of the ocean?

Richness is here aggregate by eco-province because this particular part of the study incorporates both coordinate records and literature derived range extents and is *only* focused on patterns of diversity in the shallow oceans. Literature derived ranges can often allow species to be allocated to a marine province without a direct coordinate record and so allow for a more complete picture of regional species richness than coordinates alone. As above, we have explained this in the methodology.

Lines 422-430: "To investigate ~~the shallow water faunal compositions~~, we allocated each species to one or more of 232 marine ecoregions of the world¹⁴. Species were allocated either on the basis of a coordinate record falling within the limits of the ecoregion or if the descriptive ranges in our catalogue placed them within the region. Coordinate records were allocated to biogeographic regions using the R package meowR¹⁴ and manually verified using GIS software. We ultimately grouped these results by marine eco-province and used these to compare faunal composition and patterns of \$\beta\$ -diversity as this maximised faunal information between regions and minimised variability based on inconsistent sampling effort (Figs. 1,3)."

Formatted: Font: Bold, Italic

Formatted: Font: Italic

Formatted: Line spacing: single

Formatted: Font: Italic

Formatted: Font: Italic

Formatted: Font: Italic

Formatted: Font: Italic

Fig 3. What is the scale for the colours in part A?

We have added explanation of this to the figure legend. The scale is simply raw interpolated species number.

~~Lines X-X~~ Lines 806-807: "Blue colours indicate fewer species, red colours more species, scale bar gives total number of species present in a given cell"

Formatted: Font: Bold, Not Italic, Highlight

Formatted: Font: Not Italic

Formatted: Font: Not Italic

Extended Data ~~Extended data~~

Fig 1 - nice plot, could this be included in main MS as is a key finding?

We feel that although an interesting finding, this is not as key as the other findings that receive main text figures. It is present in the ~~E~~extended ~~F~~figures where it will be presented next to the text in the online version of this study.

Fig 3. So here are the environmental data, relegated to SM where few readers will see them and unconnected to the species data.

Table 1. So here I find microscopic text that is the statistics referred to. This is not convincing. Patterns and correlations must be shown to the reader, not hidden away in SM tables.

Similar comments for reset to SM.

As described above, it is the journal style to include most of the findings in the ~~E~~extended ~~F~~figures with only a select number of main text figures for the most key findings, primarily due to constraints

of space. ~~These are not the same as supplementary figures and should not be treated as such nor should the data be considered 'hidden away'.~~ The Extended Figures will be presented alongside the manuscript in the online version of this study. Our increased reference to figures and tables throughout the text will also help readers to find the appropriate data and make our study clearer.

Dear Reviewers

Please find attached our revised manuscript, “Competing processes shape variable deep seafloor diversity between hemispheres in Asteroidea”. We have revised our manuscript substantively in light of the Reviewers’ comments, and our responses to each comment are detailed below. We thank the Reviewers and Editors and hope that the paper is now acceptable for publication in *Nature Ecology & Evolution*.

In particular, we have addressed the major concern of Reviewer 4 on the use of different methods for estimating species richness and accounting for sampling bias across our dataset. We have spent considerable time re-analysing our data to calculate Hill’s Numbers across each degree of latitude for our full dataset and each major depth strata discussed in the study, as requested. We show that this produces a very noisy estimate of species richness between latitudes, with the magnitude of the confidence interval heavily influenced by sampling effort. Moreover, sampling effort in some of the deepest waters, particularly at southern tropical latitudes is too poor to calculate any values using this method.

The interpolated curve of species richness we use in this study, accounts for sampling discrepancies by collapsing all records for a species within a degree of latitude to a single datapoint and then expanding ranges between occurrences where there are gaps. We modified the usual approach to interpolation, using additional data to produce ‘informed interpolation’. This was a bespoke approach where each species was considered independently. We used primary literature for each species to define depth ranges and maximum and minimum latitudinal ranges (see our Catalogue). We also took into account habitat data for each species (where known) and coastline disjunctions to avoid interpolation across impossible or improbable latitudes and depths, thus minimizing interpolatory overfitting. This method was highly time intensive and involved study of museum collections and primary literature and took a total of ~6 months for data processing alone.

Our ‘informed interpolation’ results are generally at the upper end of, but within, the confidence interval produced by Hill’s numbers, with this relationship becoming tighter when the dataset is split by bathymetry. We have included the Hill’s Number analyses as plots in Supplementary Data 4. Given that the interpolated data used in our study has such a close relationship with the probability envelope for the Hill’s number data we are confident in retaining it here, with appropriate caveats about the assumptions underlying this in the methods. We moreover show that the resolution of Hill’s number data is insufficient to investigate trends in total oceanic species richness at scale.

The other methods suggested by Reviewer 4 are not appropriate for our dataset given the fact that our data should only be treated as presence/absence data due to the variation in reporting (e.g. some records provide only presence/absence data, whereas other report abundances of each species) and sample collecting method. (e.g. some were hand collected, others trawled etc.). This violates the assumptions of ES50 based methods which require all samples to have been collected in the same way and leads to considerable caveats with random draw methods like Hill’s numbers. We are also not able to calculate GAM based ranges across unknown sampling because of their reliance on environmental data which would preclude our analyses of environmental drivers through circularity.

Specific comments are listed below.

Reviewer #4 (Remarks to the Author):

Review paper on global biogeography of asteroids

My apology to the authors for this late review. Because I feel it is a worthwhile work to get published and I suspect the first author is a graduate student, I have given more time to provide some detailed suggestions as how to strengthen the paper. That said, the co-authors should have been able to improve parts of the paper before submission even if it is the first authors first paper.

This is an impressive gathering of data for a global scale analysis of a major marine taxonomic group, the starfish and brittlestars. It merits publication following some rethinking of interpretations of the findings, more qualification of the generality of the study, and checking the effects of sampling bias on the results.

We thank the review for his comments about the scale of our analysis. As laid out below we have taken several steps to address these concerns, including substantive re-writing of regions of the text which required a greater degree of qualification, particularly where generalities about asteroid distributional patterns could be read as applying to the entire benthic fauna (e.g. Line 60). We had already applied corrections for sampling bias to our models, see Lines 588-598, but have further addressed the reviewers concerns about methodology by calculating Hill Number values across our various datasets. These are now included in Supplementary Data 4.

The flow and conciseness will be improved if the Results and Discussion are separated, as is normal in primary data analysis papers.

We are willing to change this if the Editor agrees with this suggestion but feel that the current text is within the acceptable style parameters for this journal. We have, however, made substantial changes to the text to address the flow of this work in response to the Reviewer's concerns.

The major weakness of this study is little attempt is made to account for sampling bias.

We have already accounted for sampling bias across our dataset, in response primarily to comments by Reviewer 1. This includes using sampling offsets and weighting in all statistical analyses (See Statistical Analyses, Lines 588-632 in the methods) such that sampling bias is accounted for across any predictions or correlations and including Figure 6 and a section in the Methodology on uncertainty (Lines 537-548) so that sample incompleteness can be better visualised. These have already been accepted by Reviewers 1 and 3.

The interpolated curve of species richness we use in this study, accounts for sampling discrepancies by collapsing all records for a species within a degree of latitude to a single datapoint and then expanding ranges between occurrences where there are gaps. We modified the usual approach to interpolation, using additional data in our Catalogue to produce 'informed interpolation'. This was a bespoke approach where each species was considered independently. We used primary literature for each species to define depth ranges and

maximum and minimum latitudinal ranges (see our Catalogue for details). We also took into account habitat data for each species (where known) and coastline disjunctions to avoid interpolation across impossible or improbable latitudes and depths, thus minimizing interpolatory overfitting.

The northern hemisphere peak in sampling bias is well known (see Chaudhary et al. 2016 and 2017 papers in TREE, <http://dx.doi.org/10.1016/j.tree.2017.02.007>). The authors argue there are enough samples to account for this but then also argue more data are needed. There are several ways to do this (see Chaudary et al. 2022, PNAS) using GAM, ES50 and Hill's numbers (rarefaction options). I think it is essential that the authors replot richness using one or more methods to account for richness.

We do not disagree that there is a well-known peak in northern hemisphere sampling bias, and that the methods listed by the reviewer can be used to correct this. However, not all these methods can be applied to our dataset for the reasons listed below.

- ES50 (of the Hurlbert Index) is primarily an abundance-based metric, calculating the expected number of species from a random subsample of 50 samples within a larger dataset. While our data, compiled from GBIF, OBIS and historical collections, is partially recorded as abundance data, this is not consistently true and even where it is, abundance data is not qualified in the same way between different datasets. I.e., some records give every specimen in a particular trawl, whereas others list every unique species encountered. The dataset thus presented can therefore only be used as a presence/absence dataset for any further analyses.
- A second issue with ES50 calculations, and one that can be also applied to Hill Number analyses, is that they are based on the assumption that samples have all been collected using the same sampling method. This is clearly not true in a global dataset which incorporates multiple sampling methods such as trawling, box coring, tangle nets, targeted ROV collection and hand collecting.
- GAM calculations are a fundamental tool in species range modelling, in effect predicting species range sizes based on the environmental (and other) conditions which occur for the known sampling sites for the species. However, given that the crux of our study is to investigate which potential environmental drivers may be linked to patterns of asteroid distributions, to calculate these from ranges derived from the same values would introduce issues of circularity into our analyses.

We have, however, taken steps to calculate Hill's Number values across our major datasets as a comparison with the interpolated results we present here. This has required a full re-analysis of our dataset, with all results now presented in Supplementary Data 4. We were only able to calculate presence/absence derived Hill's Numbers here for the reasons listed above, and also caveat that we have serious concerns about using such analyses across datasets with variable sampling method.

- Hill number values were calculated across all sampling points for each degree of latitude for the full, shallow, bathyal and lower bathyal and abyssal datasets.
- Several latitudes in the lower bathyal and abyssal dataset, particularly in the southern tropics had too few species or sampling sites for values to be calculated and are not included in our analyses.

- In general, the confidence interval for each Hill number value (given as the gap between the upper and lower estimates at each latitude) is much larger where sampling effort is low, and highly variable across the dataset, as can be seen in Supplementary Figure 4A-D.
- Comparison of the new Hill's Number results with our interpolated dataset shows that across all analyses, the interpolated results are generally (although not exclusively) towards the upper end of, but within the Hill Number confidence intervals. The only major exception to this is for some southern tropical latitudes across the full and shallow bathome datasets where our estimate is slightly higher than that of the Hill Number values.
- This relationship between the two analyses is closer when the dataset is split into bathymetric zones and with increasing depth.

Ultimately, we show that our interpolated results do not present a wildly different estimate of latitudinal diversity than do the values calculated using Hill Number extrapolations, and that these are often in close accord with each other. However, Hill Number estimates cannot give the same resolution as interpolation across this dataset because of the lack of sufficient samples across most of the deep ocean, and particularly in the deep tropics. Given the similarity between the Hill Number calculations and our interpolated values, we feel confident in continuing to use the latter for our analyses.

The use of interpolated species ranges is one way to try to account for sampling bias as applied in this study. However, this perpetuates bias because one can only produce polygons for where species have been reported from.

This is a bias that applies to all species richness estimates except for environmental niche (GAM) modelling which we cannot apply here.

Furthermore, this will not account for shifts in species ranges over the last century due to climate change and thus exaggerate equatorial richness (see PNAS paper above – showed thousands marine species shifting away from equator). Better range maps are produced by using species distribution (environmental niche) models although if these use old records they will still exaggerate the ranges (as in the Lin et al. paper cited for fish). It is also true that they can only be run when sufficient data exist, but this is not a reason not to use them, and why not exclude the use of simple interpolation when insufficient data exist?

- Species distribution (environmental niche) models are an excellent way of showing estimated species ranges, especially as they are able to predict occurrences where they have not yet been sampled. However, as discussed above, we were unable to rely on these for our analyses because they introduce circularity when trying to investigate potential environmental drivers.
- It may also be true that species are currently shifting away from the equator, and that current global diversity may not be fully reflective of historical diversity. However, this is not intended to be a study looking at recent range shifts, nor do we believe that the data are available to do so here. Sample collection data is not known for every data point in our dataset, nor has repeated sampling occurred at many tropical datapoints. To interrogate this phenomenon here would require us to exclude enormous amounts of data which may not have modern sampling analogues.

A key result is the depth gradient in species richness which is relegated to the SM. This should be in the main paper and perhaps broken into subsets for different latitudes and/or oceans.

Done see amended figure 2B

Length of coastline is a useful indicator of coastline heterogeneity but a better indicator is seabed variability or rugosity (available from BioOracle <https://www.bio-oracle.org/>).

We have included reference to the rugosity data given in Zhao et al., 2020 <https://doi.org/10.1016/j.biocon.2020.108536> which supports our coastline length data.

Lines 163-165: *Rather, we suggest that these patterns are at least partially linked to differences in the extent of the topographic heterogeneity of coastline and seafloor that exist between these regions³⁴ and to differences in the relative impacts of important environmental variables*

Title – far too broad and should clarify it is only on starfish. It should include asteroids or Asteroidea.

Done

Abstract – last sentence is very weak and can be omitted. This final sentence should focus on the novel implications of the paper.

We have reordered our abstract to end on a stronger point. See Lines 19-21 and 31-33

Introduction

Line 38 – benthos is not a biome; reword. Benthos is an assemblage of species living on the seabed. A biome is widely and long established in ecology as a large area characterised by plants that provide three-dimensional habitat, such as kelp forests. Unfortunately, the term has been used for almost any marine area in the marine literature.

We agree that these terms are often conflated in the ecological marine literature. We have changed this sentence as below:

Line 37: *The deep benthos is by far the largest single **habitable space** on Earth*

Lines 39 – use primary sources when citing facts as here; i.e., papers which have actually measured the % of the earth that is global.

We agree with the reviewer and recognise that the first source we cite (for the 90% value) does not directly calculate these values, while the second source (for the 70% value) does directly state the values used in both area calculations. We have therefore used the latter reference for both numbers cited.

Lines 37-38: *...comprising around 90% of the ocean seafloor – itself 70% of the global surface³*

Line 40-41 – this and elsewhere are “truisms” or statements which are so broad they could be both true and false without more context. One could argue that “large scale patterns .. are WELL understood” as many papers and resources map these patterns (e.g., see AquaMaps website). The simplest solution is just delete such statements as they only add words and not meaning. Other examples:

We agree that this is a general statement and have changed the sentence to clarify that we mean that the drivers behind large scale patterns remain poorly understood. We would however argue that the proliferation of maps for these patterns show that they are increasingly well described not necessarily that they are well understood.

Lines 39-40: ... ***the drivers shaping large scale patterns of diversity and species richness remain poorly understood***⁷⁻⁹

Line 43-44 all databases have “limited taxon or habitat coverage” (and they always will).

We agree that this is a further truism and have amended this passage to clarify that we refer here to the potential limitations of this incompleteness on such analyses

Lines 40-44: *Many previous studies have focused on limited geographic regions⁴, use very broad resolution data, or risk potential bias towards common or cosmopolitan species through reliance on modelled results rather than collection data. **Despite the increasing size of many available datasets, taxonomic or habitat coverage incompleteness have also hindered such analyses***¹⁰

Line 77 – repeats the “more limited” cliché above. It would be better to say how many species were in each study so we know how different they were in scope. Check throughout MS for this phrase and either delete it or make it an objective quantified statement e.g., with only half the number of species than in the present study (see line 301)

We agree that this general statement does not allow the reader to understand the difference between the sample sizes in the study and have amended as requested.

Lines 74-77: ...*this in contrast to a previous study of shallow water asteroids which proposed low tropical species richness, **based on data from reef and rocky substrate species representing only ~10% of the taxa included in the present study***¹⁰.

Line 53 – delete comprehensive, it is not comprehensive and probably can never be.

This statement was only intended to refer to the comprehensive nature of the taxonomic component of this database and we agree that it's use here, where it could also refer to the geographic completeness of the dataset is incorrect.

Line 52: *A large, **near taxonomically** comprehensive*

Line 170 “difficult to quantify” is a superfluous phrase.

We have deleted this phrase

Lines 177-178: *Marine habitat and spatial heterogeneity has rarely been investigated at scale^{33,35,36} but is here roughly estimated as a function of length of available coastline.*

Line 45-47 – one can argue that no two places on Earth are comparable for any number of reasons. Was the Atlantic more glaciated than the North Pacific and Antarctic shelf? I suggest delete this superfluous sentence.

We included reference to this more extensive glaciation at the request of the third reviewer, but agree that it is true that no two parts of the world can ever be directly comparable. We have removed reference to glaciation but have retained part of this sentence because we feel it is important to include some comment on the fact that the Atlantic cannot be used as a proxy for wider global oceans despite the tendency in some studies to describe global patterns based on the Atlantic alone.

Lines 44-47: *For instance, most **putatively global** studies have focused only on the Atlantic (although see²), **which should not necessarily be treated as representative of other oceans which have their own geographic and evolutionary histories** impacting faunal compositions¹¹⁻¹³.*

Line 60 – it is not “benthic” diversity, only “asteroid”.

We have amended this to clarify that we are only talking about asteroid diversity

Lines 59-60: *...and to depict the changing shape of benthic diversity **in this clade** across three major depth strata in the ocean*

Line 72 – methods should explain what an eco-province is. I assume it is the classification of coastal provinces using expert opinion in the Spalding et al. MEOW classification. The authors may not be aware of the data-derived realms in Costello et al. 2017, Nature Communications. These provide a comparison to MEOW and other systems and are hierarchical. This hierarchy could be compared with the findings in the study for asteroids.

The Spalding et al. MEOW dataset shows a biogeographic classification of the world's coastal and continental shelf waters, following a nested hierarchy of realms, provinces and ecoregions. It describes 232 ecoregions, which lie within 62 provinces and 12 large realms. The regions aim to capture generic patterns of biodiversity across habitats and taxa. We have added a small phrase to clarify this in the methods.

Lines 443-444: *We ultimately grouped these results by marine eco-province, **a nested hierarchical level of 62 regions encompassing one or more eco-region.***

Line 75 – I think there are more studies. E.g. polychaetes in MEPS by Pamungkas et al. The references in this and other papers may provide more examples to enrich the generality of the present study findings.

We have included additional references selected from those cited in Pamungkas et al .

Line 92 and later refer to bathomes but the papers seems to use different depths for these

in different places. A standard definition based on data should be used, not some of the numerous and varying definitions which lack any biological or environmental rationale (as in text books).

We have changed these to ‘major depth strata’, ‘major depth zones’ or ‘depth zones’ throughout the text to remove any confusion. These depth strata have ultimately been chosen so that the results in this study can be easily compared with the other works which have used similar levels. See for example: Wooley et al., 2016; Smith et al., 2008; Brown & Thatje, 2013

Line 104 – ref 23 is a weak one as it is only for one location and Lines 105-7 – this is true and several studies have suggested this over past century as cited in the Costello and Chaudhary 2017 paper cited. I agree this is an important finding in this study as this fact is generally not understood by scientists who champion exceptional deep-sea diversity.

We agree, and have removed this reference, which is only for a small portion of the deep North Atlantic off Delaware and rephrased the paragraph to better reflect the findings presented.

Lines 100-107: *Species richness in the Lower Bathyal and Abyssal zone remains highest outside of the tropics in both hemispheres. This unimodal per hemisphere distributional pattern has been noted in other taxa^{2,23}, although we note that the magnitude of these peaks are much lower than those seen in shallow waters (Fig. 2e). This relative evenness of species richness across the deep sea when compared to shallow waters reflects both the decrease in sampling effort and observed species numbers at these bathymetric extremes (Fig. 1c), but also supports the apparent tendency for deep-water clades to occupy more cosmopolitan ranges across larger and less variable habitats (Fig. 4F) at these depths^{24,25}*

Line 114-7 – needs a rewrite as unclear. Be especially carefully in claiming peaks when comparing LDG curves when it could be dips.

We have rewritten the text to make it clearer:

Lines 114-117: *Below ~100m species richness is higher in southern temperate waters than in southern tropical waters. This is not true in the northern hemisphere where species richness is higher in northern temperate waters compared to northern tropical waters only below ~750 m (Fig.2).*

Lines 122-4 – unclear sentences, “compared with a restriction only”, sentence is too long probably.

Sentence has been rewritten; new text is below

Lines 121-125: *At abyssal depths the difference between polar regions is partly due to the Arctic being shallower (Figs 2a-b), only exceeding 4000m in the Amundsen Basin and in a small region known as the Malloy Hole, and thus having less available deep water seafloor area compared to the much more extensive deep water in the Southern Ocean (5 deeps ref). This does not, however explain the observed patterns at shallower depths and outside of polar waters.*

Line 125 – state what other taxa are mentioned here, were they also benthic, how

comparable are they? Clearly taxa vary in their depth and latitudinal distributions (e.g., auks, penguins, herbivorous to non-herbivorous fish, etc.).

We have clarified our statement to include the taxa being compared.

Line 127: ...been noted previously in **benthic molluscs and crustaceans**^{26–28}

Line 126-7 – yes this has been clearly shown to be due to sampling bias. When this bias was corrected for the latitudinal gradient across all marine species is symmetrical and bimodal (see the 3 Chaudhary et al papers in TREE and PNAS – her PhD may also be online for more details).

We have added reference to these papers.

Lines 128-31 – I think the Chaudhary et al. PNAS paper accounted for shelf area and coastline length. This aspect needs more thinking through. Area is only important when habitat diversity increases with area up to certain limits – which is why the deep sea has lower richness than coastal waters. If the LDG is symmetrical as found by Chaudhary et al. PNAS then it suggests that simply temperature (or day length) is the primary driver. If not, and sampling bias has been accounted for then evolutionary history can be shown to be important, especially at family level – as nicely shown for fish in the Lin et al. papers in PeerJ and in the present paper.

We agree with the reviewer that area is not everything when considering habitat diversity and species richness and this is especially not true in the deep sea. We do argue that it may play some role in shallower waters where there is a greater diversity of habitats and where species richness is more likely to be affected. We further agree with the reviewer that this is more likely to be apparent at lower taxonomic levels.

Lines 129-134: *This pattern in shallower waters may reflect the relative lack of continental shelf area in the tropical and temperate zones south of the equator³⁰, although this may be a non-trivial relationship to untangle requiring further investigation and likely plays a larger role when considering diversity between genera or families (Fig. 4). Further potential explanations include this being an artefact of uneven sampling and taxonomic effort across latitudes^{30,32}.*

Line 134- it would be clearer to be more objective and say how different these biota are in terms of total species and unique (endemic) species.

We have included these numbers

Lines 137-138: *...between the Atlantic (439 species, 352 endemics) and the IWP (1029 species, 944 endemics)...*

Line 144-6 – the meaning of these sentences is not clear to me, “bathyal and bathyal and abyssal” and no supporting references cited. This is the first time I heard it claimed that there was a reduction in seafloor due to the Eurasian land mass. Land masses are older than the oceans so changes in sea level and the coming together of the Asia and Australian continents has contributed to high species richness in the IWP.

We agree that again this sentence is unclear. We intended this to refer only to the fact that in the modern oceans, the Eurasian land mass covers a much larger proportion of the northern temperate zone than do landmasses in the southern temperate zone and that this leads to there being less seafloor area in the northern hemisphere.

Lines 152-154: *Lower levels of species richness across most of the northern temperate IWP are partially explained by their being less available seafloor area, since more of the total area at these latitudes is land when compared with the southern temperate regions.*

Line 150 – as mentioned above, this will need rethinking once sampling effort and bias has been considered. The claim that there is no substantive variation in sampling effort across hemispheres does not agree with numerous published studies and the data in OBIS and GBIF.

We agree with the reviewer that there is clearly a massive sampling bias towards the northern hemisphere across total sampling effort and that this has been shown in numerous studies. We have clarified this statement because we intended only to refer to sampling effort across the tropics – where sampling effort for asteroids is relatively even north and south of the equator and not to temperate and polar regions where there is a much greater difference in sampling effort between the hemispheres.

Lines 160-164: *...but across the tropics where the major hemispheric asymmetry in species richness shown in this study occurs (Fig. 2b), there is no substantive variation in sampling effort, in contrast to higher latitudes where sampling effort is heavily weighted towards the northern hemisphere as has been found in other studies^{25,30}*

Line 157 -60 – “topographic heterogeneity of coastline and seafloor” is misleading as only coastline length was analysed here. It would be much better to use “rugosity” (variation in seabed slope per lat-longitude cell), a measure of topographic heterogeneity highly correlated with species richness (see Zhao et al. 2020, Biological Conservation <https://doi.org/10.1016/j.biocon.2020.108536>).

We agree with the reviewer and have added reference to the Zhao paper whose findings support our statement

Lines 164-166: *Rather, we suggest that these patterns are at least partially linked to differences in the extent of the topographic heterogeneity of coastline and seafloor that exist between these regions³⁴ and to differences in the relative impacts of important environmental variables.*

Line 162 – do asteroid and ophiuroids ever compete or occupy similar niches to justify this statement – it may be true but more support would make it more convincing.

Empirical data for this hypothesis are lacking, but there is some evidence for resource stratification between the clades, including in the cited study from the Aviles Canyon system.

Lines 169-171: *This may reflect effects of evolutionary history, possible niche competition or differential habitat preference but this requires further investigation³⁶.*

Line 167 – habitat diversity is part of biodiversity (includes ecosystem variation) – this sentence could be omitted or revised. Asaad et al. 2017 (Biol

Conservation <https://doi.org/10.1016/j.biocon.2018.03.037>) showed high correlation between species and habitat richness for the Coral Tringle.

We have revised the sentence.

Lines 175-177: **Species diversity and habitat diversity are closely linked³³ and geographically complex regions with greater habitat heterogeneity and more barriers to dispersal might be expected to harbour greater species richness^{25,34}.**

Lines 171-3 – confusing as no such places are mapped to define boreotropics (a strange term as boreal is north of temperate which is north of tropics, so is boreotropics northern subtropics?). It is best to minimise jargon like this.

We have replaced the term 'boreotropics' and the related term 'austrotropics' here and throughout the text. See. Eg.

Lines 179-181: *In the Atlantic, spatial heterogeneity and coastline length in the **northern** tropics (48,831 km) is more than 2.5 times greater than in the **southern** tropics (18,185 km), driven principally by the complex basin and island systems in the Caribbean³⁷.*

Lines 175-182. Needs rethinking. It is a nice theory but is there any evidence that the variation in salinity near these big rivers affects benthic diversity? I think not as the freshwater floats and is rapidly dispersed. This paragraph need a re-write as the interpretation is not clearly presented and too speculative. Speculation should be supported by stronger examples from this and other studies.

Our intention was not to indicate that it is the salinity variation caused by freshwater discharges that affects benthic diversity as this is likely to have effects only on small or local scales, as indicated by the rapid horizon over which salinity effects were observed in Van Diggelen and Montagna (2016), although one might expect there to be a greater effect at larger river outflows. Indeed, later in the paper we state that:

Although some marine regions, such as major river estuaries or semi-enclosed basins, do show pronounced salinity differences when compared with most of the global oceans, salinity does not vary significantly across the majority of the seafloor (Extended Data Figs 3-5) and not in a way that is likely to be biologically meaningful in restricting asteroid distributions³⁹.

Rather, our intention was to refer to additional effects of river discharge on benthic diversity, primarily those linked to variable and increased sediment outflow and seafloor turbidity, which have been shown to have major impacts on benthic communities (particularly shallow ones)

We have rephrased our sentence to clarify that we are referring only to the deposition and turbidity effects caused by river discharges and the impacts these may have on benthic diversity.

Lines 182-186: *This matches patterns of higher asteroid richness north of the equator (Fig 5), although additional factors such as the major freshwater discharges from the Amazon and Congo Rivers in the **southern** tropics, **which vastly increase turbidity and seafloor sediment deposition**, are likely to contribute to the lower faunal count observed south of the equator^{42,43}.*

Lines 186-8 – no supporting references cited but see paper by Wei et al. in PLoS One on

benthic biomass gradients which, like on land, show a strong temperate relationship.

We have added the requested citation

Line 242-6 – the finding of limited species below 1.5 oC is notable and interesting. Some papers by Daniel Pauly on the GOLT theory and Portner on OCCLT (similar theories) indicate that oxygen becomes limiting at very cold temperatures because it is too cold for the animal to absorb enough oxygen to counter the rate of cold denaturation of proteins. A recent study by Lavin, Pauly and others in Environmental Biology of Fishes provided evidence of this in the form that Arctic fish exposed to a degree or two warmer temperature (above 1 or 2 degrees) showed increased growth and size.

And

Line 270 – “for the first time in this context” is meaningless because the context is not clear. It is not the first time than low deep sea and polar temperatures have been invoked to explain low species richness in marine taxa, and this is not a surprising finding, while worth noting.

We thank the reviewer for their interest in this finding and for indicating the above studies on temperature mediated oxygen limitation in fish which provide support for our theory. We have rephrased this, and later passages to highlight these references although we note that a physiological mechanism for this thermal effect has yet to be determined in echinoderms. We have also omitted the reference to ‘for the first time in this context’.

Lines 275 – 283: This thermal threshold effect may operate in similar way to the mechanisms proposed in the gill-oxygen limitation⁵³ and oxygen and capacity-limited thermal tolerance⁵⁴ (theories whereby at very low temperatures, body size is limited. These theories have received recent empirical evidence in polar fish²², but any such physiological mechanism remains to be identified in echinoderms. While seasonal POC flux variability may play a role in shaping deep sea species richness, with the magnitude of this yet to be tested, this process appears to be principally governed by this thermal threshold effect. Thus, low seafloor temperatures prevent the establishment of species rich asteroid communities regardless of the levels of chemical energy entering the system⁴⁵.

Line 249 - unclear meaning, “increasing diatom content” compared to what? “reduction in calcium carbonate.. at the poles” compared to where, and below CCD (see more recent papers by Harris et al. 2023 in Marine Geology and Simon-Lledo et al. 2023. Carbonate compensation depth drives abyssal biogeography in the northeast Pacific. Nat. Ecol. Evol. <https://doi.org/10.1038/s41559-023-02122-9>). Does the latter paper support the absence of asteroids below the CCD? This would greatly strengthen the proposition in this paragraph.

We have clarified our statement and also included reference to the Simon-Lledo et al paper which shows a rough halving of the relative abundance of asteroids below the CCD in the CCZ and does therefore add support to this proposition. We thank the reviewer for bringing this study to our attention.

Lines 256-262: *The seasonality and increasing diatom content of POC flux at temperate and polar latitudes⁴¹ **compared with the tropics** and the reduction in the calcium carbonate-based fraction of marine snow at the poles and below the carbonate compensation depth⁴² **when compared with the rest of the global oceans**, may act as further stressors and impose additional ecological constraints on species richness and diversity at polar latitudes and bathyal and abyssal depths **as has recently been shown in the Clarion-Clipperton Zone**⁵⁰.*

Line 253 – what is meant by “young” and “but see”. Please elaborate in simpler language. Is it meant that the abyssal fauna is a subset of shallow species as proposed by Rex for molluscs, i.e. the species did not evolve in the abyss (not neoendemics)?

“Young” is meant to indicate the view suggested by Rex and also proposed for echinoids that the abyssal radiation is relatively recent and derived from shallower species and “but see” is intended to indicate that alternative views on the origins of the echinoderm abyssal fauna exist. We have amended this sentence to clarify this. We have retained “but see” for word count reasons and because this phrase is commonly used to indicate this point.

Lines 262-264: *The abyssal echinoderm fauna is also probably relatively **recently derived from shallower taxa**⁴³ (but see⁴⁴) and so may have had less time to develop the complex community structures characterising older assemblages.*

Line 256 – this sentence may mean the opposite – few records generally suggest species to be more endemic not cosmopolitan. Endemic species are geographically rare and more records tend to fill out their distribution to either confirm a restricted range (truly endemic) or not. However, regardless of this clarification this sentence is out of context here and could be omitted or revised to fit the paragraph better.

We agree with the reviewer that in general few records can be an indicator of endemism, but intended this point to refer to what we perceive as a potential over-eager view to ascribe cosmopolitanism to abyssal species which have been sampled sporadically and may have been mis-attributed this status due to the lack of morphological divergency. Cosmopolitanism certainly does exist for some species, we wished to highlight that it may not be completely true for all or even most deep-sea taxa. Nevertheless, we have omitted this sentence as requested.

Line 261 – please include the richness with depth plots in the main MS.

Done see amended figure 2b

Line 276-80 – seem to be Methods?

We prefer to retain this text here as an introduction and link to this section.

Line 281-293 – this merits a re-write. Why are only a few families shown? Are they the most species rich? And as many examples are of only one family then it should be stated which family, not vague phrases like “bathyal families”. This is a nice part of the paper because it shows how different families evolved to fill different geographic and environmental niches.

It is another good example of how the latitudinal gradient varies between families but when species are aggregated to Class level there are clear gradients.

And

Lines 294-5 – a weak sentence and could have been said without any of this research. Revise or omit.

We have only shown these six families in the main text as indicative examples of the biogeographical points we wanted to highlight, although A, B and E are amongst the most species rich of all asteroid families. We have not shown all families because of space limitations in the figure but every currently recognised asteroid family is plotted in Supplementary Data 3 and we have added additional references to these data in the text. We have also revised this section to better emphasize the point that latitudinal gradients are often scale dependent. We have further added detail to the methods to indicate that only exemplar families are shown in Figure 4. This is further highlighted in the figure legend for Figure 4, where we explain that only 6 families are shown, and other families can be found in Supp Data 3.

Lines 290-307: *Examination of species richness at family level provides the opportunity to investigate the effects of phylogeny on bathymetric diversity and to investigate the impacts of taxonomic scale. A large proportion (11/38) of families, including several of the most species rich, are restricted primarily to the shallow tropics (e.g. Fig. 4a for one exemplar and Supplementary Data 3 for more examples), contributing to the high diversity observed in this region, whereas only a small number of shallow families are anti-tropical (e.g. Fig. 4b, Supplementary Data 3). Bathyal families tend to encompass broader latitudinal ranges as environmental conditions become less variable and tend to either be centred on the tropics (e.g. Fig. 4d, Supplementary Data 3) or have distinct southern tropical minima (e.g. Fig. 4e, Supplementary Data 3). Primarily abyssal families occupy practically all available deep-sea latitudes (Fig. 4f; Supplementary Data 3), absent only from sub-zero polar waters. Conversely, we identified a small number of shallow and upper bathyal families endemic to Antarctica (e.g. Fig. 4c, Supplementary Data 3) and note that overall taxonomic diversity at all levels is much higher at high southern latitudes than the equivalent northern latitudes, in line with previous findings for echinoderms⁵². **It is clear, therefore, that the strength, direction or presence of latitudinal gradients vary greatly with taxonomic scale, often only becoming apparent when aggregated to Class level.***

Lines 299-301 – this first statement of the conclusions suggests this is a novel finding. But it is to be expected that patterns in abyssal taxa differ from shallow taxa, right?

We agree with the reviewer that a pattern of differing abyssal diversity when compared with shallower waters has been shown in several taxa, although we believe it is still worth noting.

Lines 302- this discussion about thermal energy could be clearer by being more explicit about what is meant by “energy” – what is chemical energy if not related to nutrients? Is it oxygen or hydrogen sulphide – but neither were studied here. It may be better to start this paragraph from line 304 (“We show for the first time ...) and state the temperature threshold which is novel (but the idea of temperature thresholds is not).

And

Line 311 – I presume nutrient here refers to POC? Better say this and not other nutrient sources (whale and wood falls, hydrogen sulphide vents, etc.). See the important paper by

Wei (Wei C-L et al. (2010) Global Patterns and Predictions of Seafloor Biomass Using Random Forests. PLoS ONE 5(12): e15323. doi:10.1371/journal.pone.0015323) which is very relevant here. It supports the idea that temperature effects on decomposition and recycling lead to low sediment organic matter in warmer environments (as occurs in forest soils with latitude).

We agree that one of the more interesting findings of this study is linked to the thermal threshold effects and variation in LDGs with taxonomic scale and have accordingly edited this passage following the reviewer suggestions to lead with the threshold effect. We have further clarified where we intended to refer to POC flux rather than using nutrient availability as a convenient synonym and have added reference to the Wei paper within the text.

Lines 314-329: *In conclusion, we show for the first time that benthic species richness is strongly limited by a minimum temperature threshold, around 1.5 °C, observed across all bathyal and abyssal regions in this study. Overall, the bathymetric profile of starfish biodiversity supports a species-energy framework, driven at all depths above the thermal threshold by variation in thermal energy, with nutrient availability (POC flux) of increased importance in deeper, more thermally homogenous waters. The bimodal peaks of temperate richness apparent at bathyal depths are closely linked to elevated POC flux on either side of the tropics, but at abyssal depths, the relatively even distribution of species at lower latitudes is restricted near the poles by a deep-water thermal threshold. The magnitude of this hard thermal boundary suggests that such threshold effects may play a greater role in determining deep-sea richness patterns than has been previously recognised. Our results, encompassing sampling across all major basins and bathymetric and phylogenetic levels for the first time for a single Class of marine invertebrates, also show that while broad patterns of diversity are shaped by widely recognised general trends, regionally specific evolutionary factors, including geographic complexity and clade niche specificity are of substantive importance at different scales.*

Lines 207-209: *In lower bathyal and abyssal water, nutrient availability has previously been considered the principal driver of diversity in what is broadly a cold and highly thermally homogenous environment^{2,38}*

Lines 391 -324 – this is not a conclusion and could be omitted. Conclusions should focus on the implications of the present study.

We prefer to retain this sentence here.

Lines 324-327 – another truism that does not need to be repeated – of course more sampling will improve knowledge. However, there was no quantification of sampling gaps in this study and there were claims that there was sufficient sampling; both cannot be correct. I encourage the authors to try Chao's Hill numbers to estimate species richness and indicate where sampling gaps lie.

Please see comment above referring to Chao's Hill numbers.

Line 328 – to say “complete spatial description” is at odds with the previous sentences and cannot be true for either shallow or deep seas.

We agree and have rephrased this passage

Lines 340-341: *Large scale spatial and bathymetric* descriptions of global and regional diversity for an entire Class of benthic marine species *are* a rare resource

Methods

Line 340 – saying the data “will be” available on a institutional website is not acceptable nowadays. The data used should be published with a DOI in a repository such as Figshare or Zenodo. In addition, the new data collated from the literature should be published into GBIF or OBIS (or EMODnet Biology). This would reciprocate the benefits the authors gained from using data from GBIF and OBIS.

It is an institutional requirement for the data to be hosted this way and only to be available after publication. All of the new coordinate data, which are principally those derived from museum specimens, is being periodically fed directly into GBIF via the NHM data portal as these records are updated in the NHM catalogue and can be found at <https://www.gbif.org/dataset/7e380070-f762-11e1-a439-00145eb45e9a>

Were species only known from fossils excluded? such records occur in GBIF and perhaps OBIS.

Yes, only extant taxa were included.

Line 344 – each dataset used is like a paper in a journal and if one uses GBIF or OBIS one agrees to cite the datasets (CC-BY licences). Thus, not to do so is a breach of conditions of use. I recommend copying out the citations and placing them in the SM as provided (some will be incomplete).

Done, See Supplementary Data 5

Line 361 – the authors have greatly misunderstood the iNaturalist dataset. Only “Research grade” data are published into GBIF. These are all verified by independent experts and supported by photographs. In contrast most “scientific” datasets are not independently verified or supported by photographs and are likely to contain more errors. How do the authors know that “experts” identified specimens from research cruises – did they check the metadata of each recorded for who identified the specimens? If so say so but I suspect they did not think of this. Thus, it is likely that some records have been misidentified.

We removed the records from iNaturalist at the request of another reviewer. The requirements for “Research grade” data as flagged by iNaturalist are not that they are verified by independent experts but that they meet five criteria – they have an attached image, a locality, time and date information, are from a named individual and finally have 2/3rds of all identifications in agreement. In practice, this means that once something has been identified by two people who agree with each other, it is classed “Research grade”. This can be done by ANY member of the public, not just independent experts, and indeed few of the people who use this site would be categorized as such. We have observed that species are frequently mis-identified, particularly marine invertebrates. We do, however, agree that just because something was collected on a research cruise or is at a museum it does not necessarily follow that it is expert identified, but it

is more likely to have been treated by a taxonomic authority than iNaturalist data. Mistakes in identification will be present in both kinds of datasets and this is why we introduced a second step to compare all the localities for each species with our descriptive Catalogue to remove species from localities where they are not known to exist in the published literature. This additional step, the comparison of observational data points with species ranges derived by hand from peer-reviewed literature reports for every single species in the study, greatly strengthens our study. Details are given in the methods from Lines 399-414

Lines 373-4 – I am surprised that more of these cruise records were not in OBIS. My memory is that considerable data from these research vessels are in OBIS. They would probably be published through the AADC for Australian and NIWA for New Zealand data.

The asteroid data from these expeditions is not, to the authors knowledge, present in OBIS. For John Murray, Challenger, Discovery and BANZARE, practically all the asteroid material is held in the NHM (London), where all authors are based, except for a small subset of duplicate specimens retained in other institutions. These, and some instances where some specimens from this material have been included in later publication datasets may be the records seen by the reviewer. These data are only present in GBIF where they have been uploaded through the NHM Data portal which links directly with the GBIF database, but where present, almost all are listed simply under a station number and do not have coordinates attached. Indeed, the geolocations for all of the Discovery specimens are placed in central Uganda while the Challenger material is listed only as a single lot indicating that 1096 specimens were received from the expedition. Full georeferenced data from these expeditions will be uploaded to GBIF as part of the data release from this publication.

Lines 382-4 – Please make a clearer statement, or a table of the number of records and species from OBIS, GBIF and authors own collation.

These numbers are in the text, Lines 354-396; and we have further included a table in our methodology section to highlight this (Table 3).

Lines 402-3 – the authors should know this exactly from their datasets. Please make a quantified statement of how many species and records with adequate georeferencing were from different kinds of sources.

These were already included in the text but are also now included in the table mentioned above.

Line 419- how do trenches create a “disjunct habitat” ? Would not the water mass, temperature, salinity. CCD horizon be the same? If there were too few records below a certain depth for analysis then just say so and not invent debatable reasons.

We have removed this comment and revised the text in line with the reviewer's point.

Lines 433-436: *Records of asteroids from hadal depths (>6,000m) were removed as taxonomic diversity was low, data were sparsely distributed **and number of total records low**, and because the trenches that characterise this depth zone have been extremely variably sampled⁶⁴.*

Line 428 – explain to reader what eco-provinces are (See definition in Spalding et al.

source). But note this is only coastal and so omits most of the ocean. Why not use the global classification of marine biogeographic realms instead <https://www.nature.com/articles/s41467-017-01121-2> especially when so much emphasis is on the deep sea in this paper.

The analysis presented in figure 3 is focused on shallow water only and while most of the interesting findings of this study do focus on the deep sea, this is a separate analysis.

Line 445 – another sweeping sentence that can be omitted.

We were asked to include a qualifying statement of this kind at the request of another reviewer

Line 461 – where are the shapefiles from the interpolation available from? E.g. Figshare, Zenodo.

These data are intended for publication in a separate work which will be submitted before the end of the year. Data were generated in batches and summed to give the final values in our analyses. A map of these data included FOR REVIEW PURPOSES ONLY, is included here.

Line 467 – the IWP is not a “basin”. Neither is the Atlantic but it may contain several basins. See Harris et al. 2014 Marine Geology on Geomorphology of the oceans).

We agree that we have used the wrong terminology here and have rephrased this sentence

Lines 496-498: *For analysis of the Atlantic and Indo-West Pacific as separate **regions**, the ranges of species present in both **oceanic regions** were manually adjusted to reflect only their range limits within that **region**.*

Lines 471+ So interpolation was on every species with no requirement for minimum number of records or validation with literature? See previous comments but this method then perpetuates sampling bias. In contrast SDM account for this but have not be used.

We did not set a minimum number of records per species, but all ranges generated were manually and extensively validated with literature records, and only on datapoints which has likewise been literature validated. We appreciate that this may lead to over-estimation of range sizes in some cases but have noted this potential source of error in the text.

Lines 467-468: *...but may overestimate diversity by overfitting species range where habitats and conditions are not continuously present.*

Line 489 – this is not true. There is no requirement for an SDM for a minimum number of records per cell. However, the records should be sufficiently geographically spaced to account for the environmental niche of the species.

And

Line 490-1 – this is also untrue. There are over 25,000 marine species range maps from SDM available from AquaMaps for over ten years, plus expert drawn range maps of 100-1000s species on IUCN Red List Assessment website. Some of these may include asteroids but the authors seem to have been unaware of them.

We thank the reviewer for bringing this error to our attention, we had not meant to include ‘per cell’ as there is no minimum requirement for SDMs in this way. We had also neglected to refer to **AquaMaps** which does include 386 Asteroidea range maps (although only 3 have been expert verified), but do not feel that these would be useful in our analysis due to the way they are generated. Only one starfish species, *Pycnopodia helianthoides* is currently on the IUCN Red List. We have revised these sentences to remove these claims.

Lines 515-518: *Such species range models typically require a minimum number of records to **adequately capture species niches** and are only considered robust with much larger sample sizes, with most previous work having been done on terrestrial and not marine fauna⁶⁷⁻⁷⁰.*

Line 518 – “many species” – please say how many and/or what proportion.

We have added this information.

Lines 520-522: **A total of 22.6% of species in our analysis are represented by fewer than ten records and would need to be discarded from further analyses using these methods.**

Lines 496-499 – what is the basis for these depth boundaries? I recommend using one based on environmental data such as the analyses of Ecological Marine Units by Sayre et al. 2017. A three-dimensional mapping of the ocean based on environmental data. Oceanography 30(1): 90–103. <https://doi.org/10.5670/oceanog.2017.116> and/or Costello, Sayre et al. 2018. Stratifying ocean sampling globally and with depth to account for environmental variability. (Nature) Scientific Reports 8, 11259. DOI:10.1038/s41598-018-29419-1

These depth boundaries were selected to be directly comparable to those used in Wooley et al (2016) and are based on the existing biogeographical boundaries of ophiuroids, which are likely to be a decent proxy for those in asteroids given their close phylogenetic relationship, similar habitat and trophic habits

Line 523 – as noted before, please say exactly what each of these regions of analysis were.

We have included this information here, and elsewhere in the text where appropriate.

Lines 528-532: *These data were used to produce bathymetric diversity plots across the global extent of latitude (Fig. 2a), for the Atlantic and ‘extended’ Indo-West Pacific (encompassing temperate Australia and New Zealand north to Japan) assemblages separately (Fig. 5) and for each family of the Asteroidea (Fig. 4, Supplementary Data 3).*

Line 528 – why include silicates when it is then repeatedly stated that they were excluded. Si is only generally important for diatoms; is it important for asteroids?

Silicates were only excluded from our analyses of the Atlantic and extended IWP, but were included in the total and hemisphere analysis. While asteroids may not be directly influenced by Si levels, it is possible, especially below the CCD, that diatom based (and therefore silicate dependent) marine snow may form a portion of their diets. It is clear from our results that there is no substantive relationship between asteroid richness and silicate levels, but we felt that there was no point in excluding a variable without testing it. We have amended this section to clarify that Silicates were only excluded from some analyses.

Lines 558-560: *Silicates were not included in our separate analyses of the Atlantic and expanded IWP due to co-linearity issues with nitrates.*

Line 527 – are phosphates really estimated in this way? No reference is cited. This would not seem to make much sense as nitrogen is typically limiting in marine waters and varies with seasons and depth. I cannot find any indication that this claim is true

e.g., <https://onlinelibrary.wiley.com/doi/10.1111/geb.13813> <https://www.geoplatform.gov/metadata/df6ea7d8-78cc-41f9-8640-93792bcca38b>

The choice of environmental data is curious as the authors seem to have avoided using more well-known and comprehensive sources like BioOracle and GMED.

We have checked this statement and identified that this was made due to a misinterpretation of the environmental data calculations involved. We have removed this statement accordingly.

Line 538 – what “realm boundaries” are being referred to here, Spalding et al., Costello et al.?

These are based on the Spalding regions but determined by the shallow water clusters shown in figure 3b

Lines 568-570: *Environmental data for regional analyses were separately averaged within the boundaries of each oceanic realm. Oceanic realm boundaries were based on the **shallow water clusters identified** in figure 3b...*

Lines 542- the methods seem to redefine the IWP to include the entire west Pacific from the Bering Sea to Australia and New Zealand! At most the IWP covers the Coral Triangle from northern Australia to southern China. Please be more careful with terminology and correct as needed.

The INDO-west Pacific also includes the Indian Ocean and a many of the oceanic islands to the East of the Coral Triangle. Our rationale for including Australia/New Zealand and expanding north to the Bering Sea is given in the methods and we now use the term “extended IWP” to indicate the region. Being discussed is larger than the area normally considered.

Lines 579-581: *The ‘extended’ IWP was extended northwards to incorporate the Cold Temperate Northwest Pacific as the Bering Sea provided a more structured geographic boundary and allows for direct latitudinal comparisons with the Atlantic.*

We have amended the text throughout to clarify our use of ‘Extended’ IWP

Lines 572-575: *Although the temperate regions of Australia and New Zealand and the north-west Pacific form clusters distinct from the remainder of the Indo-West Pacific, we have included these within an extended IWP regional analyses for simplicity and to allow these basins to cover an equivalent latitudinal extent to the Atlantic.*

Line 547 – no reference cited for claim of Mediterranean being “extremely distinct” (see the EMU and biogeographic sources I have mentioned previously).

Our environmental data clearly show the Mediterranean to be significantly warmer and saltier than the neighbouring Atlantic. We have, however included references for this statement as it was unreferenced in the current text.

Lines 568 and 584 – define SLM and mtry at first mention.

Lines 600 ...*each depth section for SLM (stationary linear model) based analysis*

And

Line 616-617: ...*using a mtry (number of values to sample at each node) value of three.*

Why was oxygen not included in the analysis? Large mid-depth zones are oxygen deficient and likely to affect asteroids.

Oxygen was included in all analyses. See statistical results tables (Extended Data Tables 1-2)

Where is the list of species by family used in the study?

We have included this in Supplementary Data 6

References 58 and 59 have author the same as the source name – is that correct? Check for misspellings.

We have checked all references and confirmed that this has been amended.

Ref 60 should be cited by first editors name and then et al. Unfortunately this is hidden behind another click on their webpage but it then gives credit to the editors as is more conventional.

We have not altered this citation as we are following the citation guidelines on the WoRMS webpage. https://www.marinespecies.org/about.php#cite_worms

Figures

In all legends just say what the figure is and remove interpretation which is in the main text (and should not repeat what is in the legends). Make all text at least double the size so it is EASY to read. Add in missing parentheses in legends. Replace species diversity with “species richness” or number of species”

Fig 1 – colours of points on map are illegible. The colour scale on the map for sampling effort seems to be duplicated for depth zones.

We have already changed the colour scheme at the reviewer's request. The colour points on the map are harder to see because of the single degree square resolution used, but the figure file allows for zooming in.

Fig 2 – Part B does not seem needed and the 3 graphs in Part C should be combined into one plot so it is easier to compare the latitudinal gradients. Part A should be one scale, not three.

We have replaced Part B with the plot of starfish richness with depth.

Fig 3 – very nice except it excludes most the ocean and all the deep sea that is discussed so much in the paper. It claims to exclude polar regions but does include some of the Arctic, and why exclude them? There are records from Antarctica and the Arctic.

This part of the study is focused only on shallow water taxa. We chose not to include the Antarctic and Arctic Oceans because their level of circumpolar connectivity precluded them from informative inclusion in our UPGMA analysis.

Fig 4 – another nice plot but why only these families and not more? Please add a line plot of total number of species per latitudinal band to show gradient.

As mentioned in the figure legend at text, these are illustrative families for the major distribution patterns shown across the Asteroidea. We have included all other families in Supplementary Data 2

Fig 5 – nice plots. Fig 2c should be plotted this way as well. Make plots the same scale for Atlantic and IWP. However, it is not clear what the comparison is. Is it all the Atlantic and West Pacific for the same latitudinal extent? That makes sense but then it is not the IWP. The depths used seem to differ from those in the text and need to be standardised throughout based on some data-driven rationale.

We have clarified that this figure shows our ‘extended’ IWP, incorporating Japan and temperate Australia/New Zealand such that the Atlantic and West Pacific regions cover the same latitudinal extents.

Fig 6 – nice idea but obviously they do not work because there are no scales shown and now different depth zones used than in the last figure. Simple plots of richness per depth,

temperature and POC for latitudinal climatological zones and/or oceans may be easier to understand.

We have amended these plots to include scales and better legends to increase the interpretability of the figures.

Minor comments easily corrected

The text often slips between past and present. All results and methods (eg from line 349) should be in the past tense.

We have corrected this throughout the text

Throughout replace diversity with either richness or number of species. If another measure of diversity is intended then state what it is. E.g., line 240, 291, probably elsewhere as well.

We have corrected this throughout the text where present

Round % to whole %, at present some are, some to one and some to two decimal places.

We have rounded figures reported from other studies to whole percentages and presented our data to one decimal place.

Line 83, - nearly always it is best to put “However” at beginning of sentences than break up clauses with it in the middle.

Changed.

Line 136 – tropico should be tropical.

Changed.

Line 139 – what is austro-tropical and how can it be in the Atlantic?

This was meant to refer to the southern tropics but is clearly ambiguous and has been corrected.

Line 140 – here and anywhere else avoid the ambiguous “/” – say and, or, and/or or drop one of the words.

Changed throughout

Reviewer #4:

Remarks to the Author:

I thank the authors for the detailed and thorough reply to my review suggestions. I seem to have missed the other reviewer comments so I cannot comment on how they align with mine. The finding that temperature largely explains the patterns advances understanding and supports previous studies. Indeed, it supports the hypothesis that the marine biota are largely distributed according to the environmental conditions rather than dispersal barriers as occurs on land.

The finding of exceptionally low richness below 1.5 °C is also important and may cause the deep-sea super-richness chauvinists food for thought. It also makes sense at many levels (ecological, evolutionary).

However, these results are not clearly shown in the figures. The latitude plots should include temperature (max, min, mean?) and nutrients as well as species richness, and simple X-Y plots of the variables against richness would be better than the 3-D plots with illegible axes. The figures must greatly enlarge the text so it is easily legible.

We thank reviewer 4 for his comments on our findings, especially those around the low thermal threshold.

We have included latitudinal plots of environmental conditions (including Temperature and Particulate Organic Carbon Flux) in Extended Data Figures 2-4. We have also modified Main Figure 6 so that XY plots of richness against Temperature and Particulate Organic Carbon Flux are presented alongside our 3D plots as an aide to interpretation. We have ensured that the text size for these figures meets the NEE standards and is legible.